# MultiViewPano: Training-Free 360° Panorama Generation via Multi-View Diffusion and Pose-Aware Stitching

**(A) Our method (single- & multi-view input)**

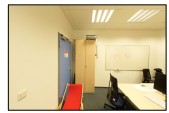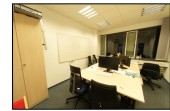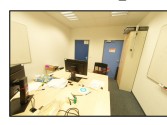

*Inputs — Multiple arbitrarily posed images*

**(B) Existing methods**

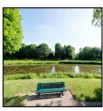

*Input — Fixed 90° FOV image*

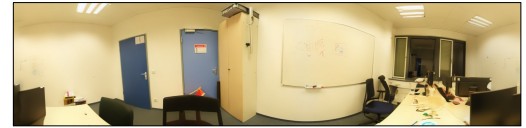

*Generated 360° panorama*

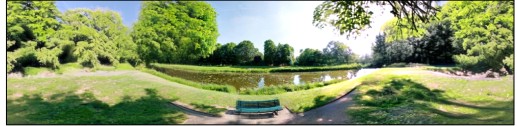

*Generated 360° panorama*

Figure 1: Existing methods focus on generating panoramas from a single image, assumed to have a 90° FoV. Our method extends to multiple arbitrarily posed inputs and FoV. In example A, the input images are from the ScanNet++ dataset Yeshwanth et al. (2023). Example B uses a perspective crop from the SUN360 dataset Xiao et al. (2012). Both panoramas shown here were generated using our method.

## Abstract

We propose MultiViewPano, a training-free framework for generating 360° panoramas from one or more arbitrarily positioned input images with varying fields of view. Existing panorama generation methods are limited by requirements for fixed 90° field-of-view inputs, single viewpoint assumptions, or extensive task-specific fine-tuning. Our approach addresses these limitations by leveraging a pre-trained multi-view diffusion model (SEVA) to synthesize overlapping novel views along strategically sampled camera trajectories, followed by a novel pose-aware stitching algorithm that exploits known camera geometry for seamless fusion. Unlike traditional feature-based stitching, our pose-aware approach directly utilizes camera poses to determine optimal seams and blending regions. Experiments on standard benchmarks demonstrate that our method achieves competitive performance with state-of-the-art approaches (FID of 12.82 on Laval Indoor, 22.99 on SUN360) while supporting arbitrary input configurations without requiring retraining or fine-tuning. Anonymous Repository link.

## 1 Introduction

Panoramic images are essential for applications including virtual reality, architectural visualization, and immersive content creation da Silveira et al. (2024). Unlike standard rectilinear images, panoramas capture complete horizontal coverage from a viewpoint, providing comprehensive scene visualization. However, generating coherent panoramas requires addressing several technical challenges: inferring occluded scene content, maintaining geometric consistency across viewpoints, and seamlessly blending views with varying exposures and perspectives.

Panorama generation from real-world images presents fundamental challenges that existing methods struggle to address. Current approaches typically assume restrictive input constraints, such as fixed 90° field-of-view or single viewpoint captures, that limit their practical applicability. Meanwhile, the limited size and diversity of panoramic datasets compared to standard vision datasets makes it difficult to train robust, generalizable models.

Existing approaches suffer from three key limitations that restrict their real-world utility. First, most methods require task-specific fine-tuning on limited panoramic datasets, constraining their ability to generalize beyond training distributions. For instance, CubeDiff produces high-quality results but requires inputs with exactly 90° field-of-view Kalischek et al. (2025), failing when this assumption is violated. Second, outpainting-based methods like PanoDiffusion Wu et al. (2024) assume all input crops originate from a single camera center, making them impractical for multi-view captures where cameras are spatially distributed. Third, existing methods struggle with arbitrary camera poses and varying intrinsics, limiting their applicability to real-world image collections.

We observe that multi-view diffusion models, trained on diverse video and multi-view datasets, naturally learn spatial priors including viewpoint coherence, parallax relationships, and object permanence, properties essential for panorama synthesis but absent in single-view models. Building on this insight, we propose MultiViewPano, a training-free framework that leverages pretrained multi-view diffusion for flexible panorama generation.

Our approach operates by sampling strategic camera trajectories around a scene center, synthesizing overlapping views using a multi-view diffusion model, and fusing them through a novel pose-aware stitching algorithm. Unlike traditional feature-based stitching methods that can fail with repetitive textures or insufficient overlap, our pose-aware approach directly exploits known camera geometry to determine optimal seam placement and blending weights. This design enables panorama generation from arbitrary multi-view inputs without requiring retraining or fine-tuning.

Our key contributions are:

1. We introduce MultiViewPano, a training-free pipeline that generates 360° panoramas from one or more arbitrarily posed input images, supporting flexible camera configurations that existing methods cannot handle.

2. We develop a novel pose-aware stitching algorithm that leverages camera geometry for robust view fusion, outperforming traditional feature-based approaches especially in challenging scenarios with repetitive textures or limited visual overlap.

3. We demonstrate competitive performance on standard benchmarks (FID of 12.82 on Laval Indoor, 22.99 on SUN360) while uniquely supporting multi-view inputs with arbitrary poses and field-of-view constraints.

The remainder of this paper is organized as follows. Section 2 reviews related work in panorama generation, novel view synthesis, and image stitching techniques. Section 3 details our MultiViewPano framework, including trajectory design, pose-aware stitching, and multi-view extensions. Section 4 presents comprehensive experimental evaluation on standard benchmarks, including quantitative comparisons, ablation studies, and qualitative analysis. Finally, Section 5 discusses limitations and future research directions.

## 2 RELATED WORK

Our work connects to multiple areas. We begin with panorama generation methods, which form our main point of comparison. Since our approach combines multi-view diffusion with pose-aware stitching, we also discuss related work on multi-view diffusion models, image stitching, and neural radiance methods.

### 2.1 PANORAMA GENERATION

Panorama generation can be broadly divided into *Text-to-Panorama* Chen et al. (2023); Feng et al. (2023); Wang et al. (2023); Zhang et al. (2024) and *Image-to-Panorama* Dastjerdi et al. (2022); Wu et al. (2024); Kalischek et al. (2025); Zheng et al. (2025). Most existing methods fine-tune

Stable Diffusion to predict an equirectangular panorama via progressive outpainting or multi-view generation, but this strategy often produces noticeable geometric distortions Zheng et al. (2025). CubeDiff Kalischek et al. (2025) alleviates these artifacts by generating the six perspective faces of a cubemap, better matching the inductive biases of the pretrained model. Likewise, MVDiffusion Tang et al. (2024) generates eight perspective views using a correspondence-aware attention architecture. The views can then be stitched into a full 360° panorama, albeit with a restricted vertical FoV.

## 2.2 Multi-View Diffusion Models

Stable Virtual Camera (SEVA) stands apart as a generalist diffusion model; it accepts any number of input images with unrestricted camera intrinsics and extrinsics, and directly samples novel views at user-specified poses. SEVA achieves state-of-the-art consistency and fidelity across diverse benchmarks Jensen et al. (2025). Other notable contributions are GEN3C Ren et al. (2025) and CAT3D Gao et al. (2024). In this paper we use SEVA, but this component of the proposed pipeline is interchangeable with any multi-view conditioned model.

## 2.3 Image stitching

Image stitching refers to the process of combining multiple overlapping images into a single seamless representation. Brown and Lowe's seminal work Brown & Lowe (2007) established the foundation for modern image-stitching techniques by introducing an algorithm for the automatic alignment and blending of overlapping images. The proposed pipeline relies on Scale-Invariant Feature Transform (SIFT) Lowe (2004) for matching keypoints between images and Random Sample Consensus (RANSAC) Fischler & Bolles (1981) for estimating homographies.

Once image positions on the stitching canvas are determined, subsequent processing involves gain compensation to equalize exposure and color discrepancies, followed by blending. A widely adopted technique is multi-band blending Burt & Adelson (1983), in which each input image is decomposed into multiple spatial-frequency bands and then recombined using spatially varying weights.

Recent approaches have proposed neural image stitching methods Kim et al. (2024); Song et al. (2022), which train end-to-end networks to align and blend image pairs. However, their pairwise training paradigm limits scalability to large image sets, making them unsuitable for constructing full equirectangular panoramas.

## 2.4 Neural Radiance methods

NeRF Mildenhall et al. (2020) introduces a neural radiance field approach that learns a continuous 3D scene representation via a neural network. From this representation, full equirectangular panoramas can be rendered directly. When camera poses are known, NeRF-style methods offer an alternative to classic stitching, acting as interpolators to create seamless equirectangular projections.

Building on classical light field approaches Levoy & Hanrahan (1996); Gortler et al. (1996) and recent advances in neural light field representations Attal et al. (2022); Suhail et al. (2022), Neural Light Spheres (NLS) Chugunov et al. (2024) propose a compact spherical representation that implicitly stitches and re-renders panoramic frame sequences. The NLS can re-render a sequence of frames and also generate higher FoV images of the scene. As the authors denote in their paper, this method constitutes an implicit image-stitching approach.

# 3 MultiViewPano

Existing panorama generation methods are constrained by rigid assumptions such as fixed field-of-view inputs, single camera centers, and the need for task-specific fine-tuning. These limitations reduce flexibility and make deployment impractical for real-world scenarios where images come from arbitrary viewpoints with diverse camera parameters. We introduce *MultiViewPano*, a training-free framework that addresses these constraints by combining multi-view diffusion synthesis with pose-aware stitching to generate 360° panoramas from one or more arbitrarily posed input images.

**Framework Overview**   Our approach operates in three stages, as illustrated in Figure 2. First, we design camera trajectories around a computed scene center to ensure comprehensive horizontal coverage while respecting the consistency constraints of multi-view diffusion models. Second, we use SEVA to synthesize overlapping novel views along these trajectories, leveraging any available input images as geometric anchors. Finally, our pose-aware stitching algorithm exploits known camera geometry to seamlessly fuse the generated views into a coherent 360° panorama, followed by optional quality enhancement.

This design enables flexible panorama generation from diverse input configurations—from single images with arbitrary field-of-view to multiple spatially distributed views, without requiring retraining or task-specific optimization.

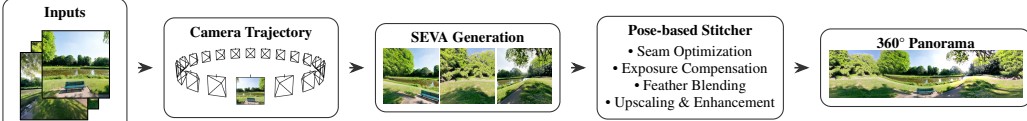

Figure 2: **MultiViewPano** overview: from one or multiple arbitrarily posed input images, we generate a virtual camera trajectory around the scene center, synthesize multiple views with SEVA, and stitch them into a complete 360° panorama with an upscaling step.

### 3.1   MULTI-VIEW SYNTHESIS WITH SEVA

To achieve comprehensive 360° coverage, we design camera trajectories that balance scene coverage with SEVA's inherent consistency constraints. We adopt SEVA's `img2img` mode, generating $21-N$ novel views from $N$ input images to remain within the model's context window limit.

**Trajectory Design.**   We evaluate two trajectory strategies optimized for horizontal panorama generation. The *pure panoramic rotation* samples camera poses at a fixed spatial location with varying azimuth angles, providing complete 360° coverage without introducing parallax. The *panorama circle* places cameras on a small circle around the scene center, introducing modest parallax that matches SEVA's training on translated viewpoints, while maintaining sufficient view overlap for consistent stitching.

**Scene Center Estimation.**   For single-view inputs, we define the scene center using the input camera's position. For multi-view inputs, we compute the scene center as the centroid of input camera positions, providing a natural reference point for trajectory generation.

Our trajectory design addresses SEVA's observed biases: the model exhibits stronger multi-view consistency for horizontal camera motions compared to vertical translations, and performance improves substantially when multiple input views provide geometric anchoring. While this design choice constrains our method to primarily horizontal panoramas, it ensures reliable generation quality across diverse indoor and outdoor scenes, as shown in Figure 3.

### 3.2   POSE-AWARE STITCHING

Our stitching approach leverages known camera geometry to achieve robust view fusion where traditional feature-based methods often fail. By directly utilizing pose information rather than relying on feature matching, our method handles challenging scenarios with repetitive textures, limited visual overlap, or insufficient keypoints.

**Spherical Projection.**   We map each rectilinear frame to a common equirectangular surface using calibrated camera parameters. For pixel coordinates $(x, y)$ with intrinsics matrix $K$ and camera-to-world rotation $R$:

$$\mathbf{d}_{\mathrm{cam}} = \frac{K^{-1}[x,\, y,\, 1]^{\mathsf{T}}}{\|K^{-1}[x,\, y,\, 1]^{\mathsf{T}}\|_2}, \qquad \mathbf{d}_{\mathrm{world}} = R\,\mathbf{d}_{\mathrm{cam}} \qquad (1)$$

Figure 3: SEVA generations under the *pure panoramic rotation* trajectory. Each row corresponds to one example, showing the input view (left) followed by sampled frames rendered along the trajectory (right).

The world ray $\mathbf{d}_{\text{world}} = (d_x, d_y, d_z)^{\mathsf{T}}$ is converted to spherical coordinates and mapped to equirectangular pixel coordinates:

$$\theta = \text{atan2}(d_x, d_z), \quad \varphi = \arcsin(d_y), \quad u = \frac{\pi + \theta}{2\pi} W, \quad v = \frac{\varphi + \pi/2}{\pi} H \quad (2)$$

This direct geometric warping eliminates the uncertainty and potential failures inherent in feature-based alignment approaches.

**Optimal Seam Selection.** In overlapping regions between projected frames, we compute pixel-wise $L_2$ color costs and extract minimum-cost vertical seams using dynamic programming. This optimization ensures that blending boundaries follow natural image gradients, minimizing visible discontinuities.

**Pose-Aware Blending.** Selected seams partition the panorama canvas into disjoint regions. We apply symmetric feather blending in a $\pm w$-pixel band around each seam:

$$I(u, v) = \alpha(u, v) I_i(u, v) + (1 - \alpha(u, v)) I_j(u, v), \quad \alpha = \frac{1}{2} + \frac{d_i - d_j}{2w} \quad (3)$$

where $d_i$ and $d_j$ represent distances to the seam boundary. Prior to blending, we apply per-channel gain compensation to handle exposure variations between frames. This pose-aware approach remains robust under modest parallax while maintaining computational efficiency. We take inspiration from methods described in Szeliski (2006).

### 3.3 MULTI-VIEW EXTENSION

MultiViewPano naturally accommodates multiple input views with known camera poses, providing several advantages over single-view generation. Multiple inputs serve as geometric anchors during SEVA synthesis, significantly improving multi-view consistency and reducing hallucination artifacts in generated regions. The pose-aware stitcher handles arbitrary numbers of input views by incorporating them directly into the spherical projection and seam optimization process.

This multi-view capability distinguishes our approach from existing panorama generation methods that typically assume single-viewpoint captures or fixed camera configurations. By supporting arbitrary camera positions and orientations, our method enables panorama generation from real-world image collections where cameras are spatially distributed, a common scenario in applications such as architectural documentation, virtual tours, and multi-device captures.

The framework scales naturally: as more input views become available, both geometric consistency and coverage improve, leading to higher-quality panoramas with fewer synthesis artifacts.

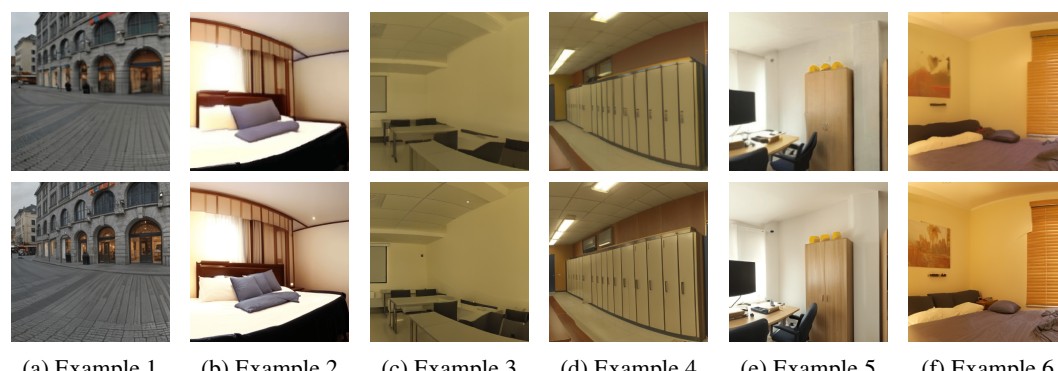

(a) Example 1    (b) Example 2    (c) Example 3    (d) Example 4    (e) Example 5    (f) Example 6

Figure 4: Effect of panorama enhancing on cropped regions. For each example, the top row shows the raw SEVA generation and the bottom row shows the enhanced output. The upscaler sharpens textures and reduces stitching artifacts.

### 3.4 QUALITY ENHANCEMENT

To address inherent limitations in diffusion-based synthesis, particularly reduced sharpness and low-frequency blur, we apply targeted post-processing using the Clarity AI Upscaler (Fainsin (2024)) framework. This combines super-resolution capabilities (ESRGAN Wang et al. (2018)), structure preservation (ControlNet Zhang & Agrawala (2023)), and seamless tiling (MultiDiffusion Bar-Tal et al. (2023)) to improve visual fidelity while maintaining geometric consistency across the panorama. This enhancement serves dual purposes: improving texture detail and mitigating residual stitching artifacts at seam boundaries. As demonstrated in Figure 4, this step provides substantial improvements in perceived quality, particularly for fine-grained textures and architectural details.

In summary, MultiViewPano provides a unified, training-free framework for flexible panorama generation that accommodates diverse input configurations while maintaining high visual quality through principled pose-aware stitching and targeted enhancement.

## 4 EXPERIMENTS

We evaluate MultiViewPano primarily in the single image to panorama setting due to the absence of established multi-view to panorama benchmarks in the literature. While our method is specifically designed to handle multiple arbitrarily posed inputs, a key advantage over existing approaches, current evaluation protocols and datasets only support single-image input scenarios. The single-image case also represents the most challenging evaluation scenario, requiring the model to infer extensive unseen regions from minimal visual context, thereby providing a rigorous test of our training-free framework's capabilities. This evaluation approach enables direct comparison with existing panorama generation methods while demonstrating that our multi-view approach remains competitive even when constrained to single-view inputs.

### 4.1 EXPERIMENTAL SETUP

**Datasets.** We evaluate on two standard panorama benchmarks: Laval Indoor Gardner et al. (2017), containing 2,100 indoor panoramas with diverse architectural styles and lighting conditions, and SUN360 Xiao et al. (2012), a more challenging dataset with 67,583 panoramas spanning indoor environments, outdoor landscapes, and complex scenes. Following prior work Kalischek et al. (2025), we use the full Laval Indoor dataset and sample 2,000 panoramas from SUN360 for robust statistical evaluation.

**Evaluation Protocol.** We adopt the evaluation protocol established by CubeDiff Kalischek et al. (2025): extracting 10 random 90° field-of-view rectilinear crops from both generated and ground-truth panoramas. This approach accommodates the fact that standard perceptual metrics (FID, KID) are designed for perspective images rather than equirectangular projections. We compute Fréchet

Inception Distance (FID) Heusel et al. (2017), Kernel Inception Distance (KID) Bińkowski et al. (2018), and CLIP-FID for semantic evaluation.

**Baselines.** We compare against three state-of-the-art panorama generation methods: CubeDiff Kalischek et al. (2025), which generates cubemap faces and achieves current best performance; PanoDiffusion Wu et al. (2024), an outpainting-based approach; and OmniDreamer Akimoto et al. (2022), which uses specialized panorama training. All baseline results are taken from the literature.

**Implementation Details.** We use SEVA with classifier-free guidance weight of 5.0 and camera scale of 0.1, determined through systematic ablation (Section 4.4). For trajectory sampling, we generate 20 novel views using the panorama circle configuration. Post-processing enhancement is applied using Clarity AI Upscaler with default settings.

## 4.2 QUANTITATIVE EVALUATION

Table 1 presents comprehensive quantitative results on both benchmarks. Our method significantly outperforms outpainting-based approaches (OmniDreamer , PanoDiffusion ) and achieves competitive performance with CubeDiff despite being training-free. Notably, MultiViewPano achieves superior performance on SUN360, demonstrating better generalization to diverse scene types.

The performance gap between methods is particularly pronounced on SUN360, which contains more challenging outdoor scenes. We attribute our method's strong performance to the multi-view diffusion backbone, which provides better geometric understanding and scene completion compared to models trained exclusively on limited panorama datasets.

When enhanced through post-processing, our method achieves the best overall performance on SUN360 (FID: 22.99, KID: 0.93), surpassing all baselines while maintaining competitive results on Laval Indoor.

| | LAVAL Indoor | | | SUN360 | | |
|---|---|---|---|---|---|---|
| | FID ↓ | KID ($\times 10^2$)↓ | CLIP-FID ↓ | FID ↓ | KID ($\times 10^2$)↓ | CLIP-FID ↓ |
| *Baselines* | | | | | | |
| OmniDreamer | 71.0 | 5.17 | 23.9 | 92.3 | 8.89 | 51.7 |
| PanoDiffusion | 58.6 | 4.08 | 26.6 | 52.9 | 3.51 | 28.9 |
| CubeDiff | 11.7 | 0.47 | 4.4 | 27.4 | 1.35 | 11.5 |
| *Our method* | | | | | | |
| **MultiViewPano** | 12.82 | 0.60 | 6.67 | 27.92 | 1.25 | 17.18 |
| **MultiViewPano (upscaled)** | 13.94 | 0.74 | 7.14 | **22.99** | **0.93** | 12.17 |

Table 1: Quantitative comparison on Laval Indoor and SUN360 in the single–image to panorama setting. Our method clearly outperforms OmniDreamer and PanoDiffusion, and achieves competitive results with CubeDiff Kalischek et al. (2025). Notably, *MultiViewPano* achieves the best performance on SUN360, surpassing CubeDiff despite being entirely training-free. Baseline results are reproduced from CubeDiff Kalischek et al. (2025).

## 4.3 QUALITATIVE ANALYSIS

Figure 5 presents qualitative comparisons highlighting key advantages of our approach. Compared to baseline methods, MultiViewPano maintains better geometric consistency, produces more coherent scene completions, and preserves input image fidelity. The pose-aware stitching approach eliminates the geometric distortions often visible in methods that learn panorama projections implicitly.

## 4.4 ABLATION STUDIES

We conduct systematic ablation studies to understand the contribution of key components and hyperparameters. Table 2 shows results across different trajectory types, camera scales, and classifier-free guidance settings.

Our pose-aware stitcher consistently outperforms traditional feature-based approaches (Hugin, OpenCV). Feature-based methods frequently fail in challenging scenarios with repetitive textures

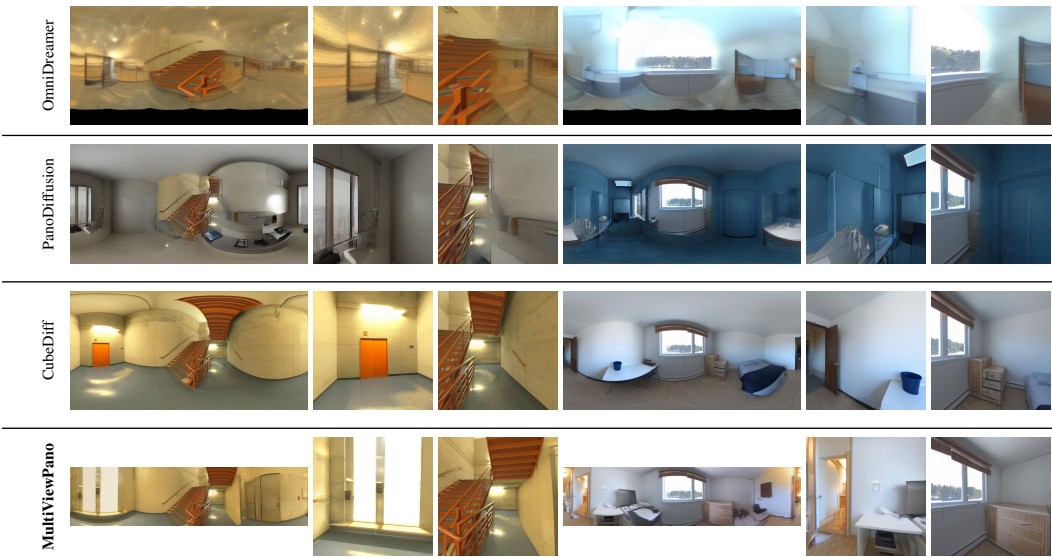

Figure 5: Qualitative comparison in the single–image to panorama setting on two example scenes. For each scene, we show the generated panorama (with the central 90° region corresponding to the input image) and two additional rectilinear views of synthesized areas. OmniDreamer and PanoDiffusion exhibit poor visual and geometrical coherence, while *MultiViewPano* preserves both consistency with the input and correct panoramic geometry. Baseline results are reproduced from CubeDiff Kalischek et al. (2025).

or insufficient visual overlap, common issues when stitching AI-generated views. Our pose-aware approach eliminates these failure modes by directly exploiting known camera geometry.

The panorama circle trajectory (cameras positioned on a small circle around scene center) generally outperforms pure rotation, particularly with smaller camera scales. This modest parallax appears to provide beneficial geometric cues without exceeding SEVA's consistency constraints.

Camera scale significantly impacts performance: smaller scales (0.1) produce more consistent results than larger values (0.5), as excessive parallax complicates multi-view consistency. Classifier-free guidance shows optimal performance at CFG=5.0, balancing generation quality with input fidelity.

| Hyperparameters | | | SUN360 | | | | | | Laval Indoor | | | | | |
| --- | --- | --- | --- | --- | --- | --- | --- | --- | --- | --- | --- | --- | --- | --- |
| | | | Hugin | | OpenCV | | Custom | | Hugin | | OpenCV | | Custom | |
| Trajectory | Camera scale | CFG | KID | FID | KID | FID | KID | FID | KID | FID | KID | FID | KID | FID |
| pure panoramic rotation | 0.5 | 2 | 1.67 | 85 | 2.16 | 156 | 1.90 | 75 | 3.03 | 103 | 2.28 | 84 | 1.65 | 72 |
| panorama circle | 0.5 | 2 | 1.76 | 88 | 1.51 | 77 | 1.89 | 74 | – | – | 2.08 | 90 | 2.05 | 77 |
| panorama circle | 0.5 | 5 | 1.13 | 81 | 1.25 | 77 | 1.21 | 66 | 1.84 | 95 | 1.60 | 81 | 0.89 | 60 |
| panorama circle | 0.5 | 8 | 1.49 | 86 | 1.80 | 83 | 1.35 | 71 | 1.89 | 84 | 1.52 | 81 | 0.91 | 60 |
| pure panoramic rotation | 0.1 | 5 | – | – | – | – | 0.97 | 64 | – | – | – | – | 0.74 | 60 |
| panorama circle | 0.1 | 5 | 0.83 | 81 | 1.00 | 69 | 1.13 | 65 | 2.06 | 98 | 1.05 | 67 | 0.82 | 59 |

Table 2: Ablation results for **MultiViewPano** across three stitching methods: Hugin, OpenCV, and our custom pose-based stitcher (Sec. 3.2). Hyperparameters include the trajectory (*pure panoramic rotation* or *panorama circle*), camera scale, and CFG. FID and KID are reported for SUN360 and Laval Indoor, with the lowest value in each metric column highlighted.

## 4.5 LIMITATIONS AND FAILURE CASES

Our method has several limitations worth noting. First, the horizontal trajectory design constrains vertical field-of-view, limiting panorama coverage near polar regions. Second, SEVA's consistency can degrade with extreme viewpoint changes, occasionally producing visible artifacts. Finally, the method inherits SEVA's limitations in handling highly reflective surfaces or transparent materials.

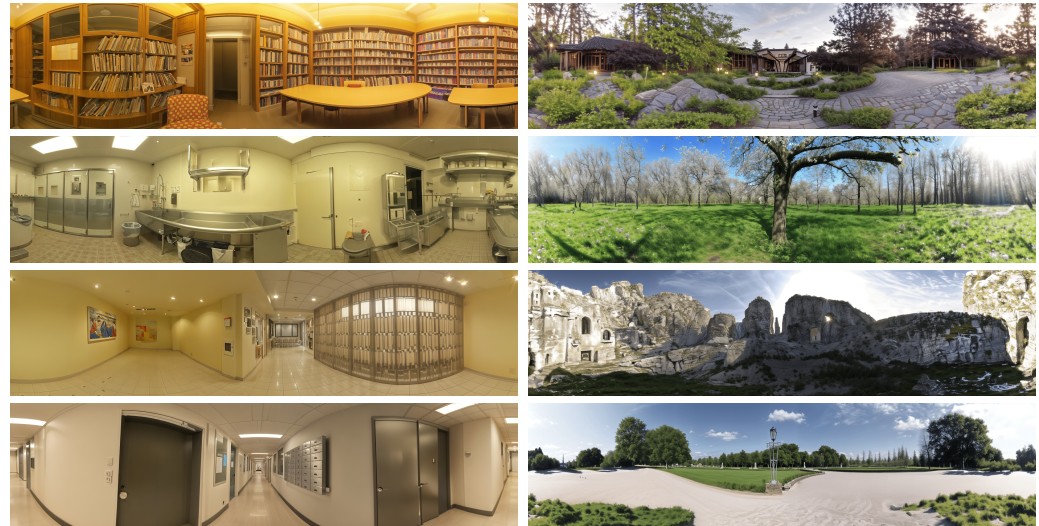

Figure 6: Examples of generated 360° panoramas in the single-image-to-panorama regime. Left: Laval Indoor Gardner et al. (2017); right: SUN360 Xiao et al. (2012). These examples highlight the diversity of our method across different indoor and outdoor scenes. Additional qualitative results, including SEVA-generated frames, stitched panoramas, and their upscaled versions for each scene, are provided in the supplementary material.

Despite these constraints, our approach represents a significant step toward flexible, training-free panorama generation that can accommodate diverse real-world capture scenarios.

## 5 CONCLUSION

We present MultiViewPano, a training-free framework for 360° panorama generation that supports arbitrary input camera poses and field-of-view constraints, capabilities absent in prior work. By combining multi-view diffusion synthesis with pose-aware stitching, our method achieves competitive performance on standard benchmarks without requiring task-specific training. Our approach leverages the spatial priors naturally encoded in multi-view diffusion models: viewpoint coherence, parallax relationships, and geometric consistency. Through strategic camera trajectory design and principled geometric stitching, we demonstrate that training-free panorama generation can match specialized approaches while offering greater flexibility.

MultiViewPano is designed as a modular framework where the multi-view synthesis component can be readily replaced as better models emerge, whether through architectural improvements, mixture-of-experts approaches, or specialized training. This modularity ensures the framework can evolve alongside advances in multi-view synthesis while maintaining its core advantages. Experimental results show state-of-the-art performance on SUN360 and competitive results on Laval Indoor, with particular strength on challenging outdoor scenes. Our pose-aware stitching consistently outperforms traditional feature-based methods, especially in scenarios with repetitive textures or limited visual overlap.

**Limitations and Future Work.** Current limitations include restricted vertical field-of-view due to SEVA's horizontal bias and the lack of multi-view to panorama benchmarks for comprehensive evaluation. Future work should focus on improving multi-view diffusion consistency across diverse camera trajectories and establishing evaluation protocols for multi-view panorama generation. This work demonstrates that combining general-purpose multi-view synthesis with principled geometric reasoning provides a promising path toward flexible panorama generation systems that can accommodate real-world capture scenarios where camera constraints cannot be controlled.

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
