# Supplementary Material: MultiViewPano Pipeline

**Abstract**

This PDF has been shrunk in size for upload purposes. Image quality may be reduced compared to the original files. All results were generated directly with our MultiViewPano pipeline (SEVA frame synthesis + pose-aware stitcher + a lightweight upscaling step).

# Contents

# 1 Dataset Overview

This supplementary document provides full qualitative examples for both benchmarks:

- **SUN360** (20 samples): perspective input crops, SEVA-generated frames, stitched panoramas, and upscaled panoramas.

- **Laval Indoor** (20 samples): identical structure with dataset sample IDs included in captions.

# 2 Qualitative Examples

## 2.1 SUN360 Benchmark

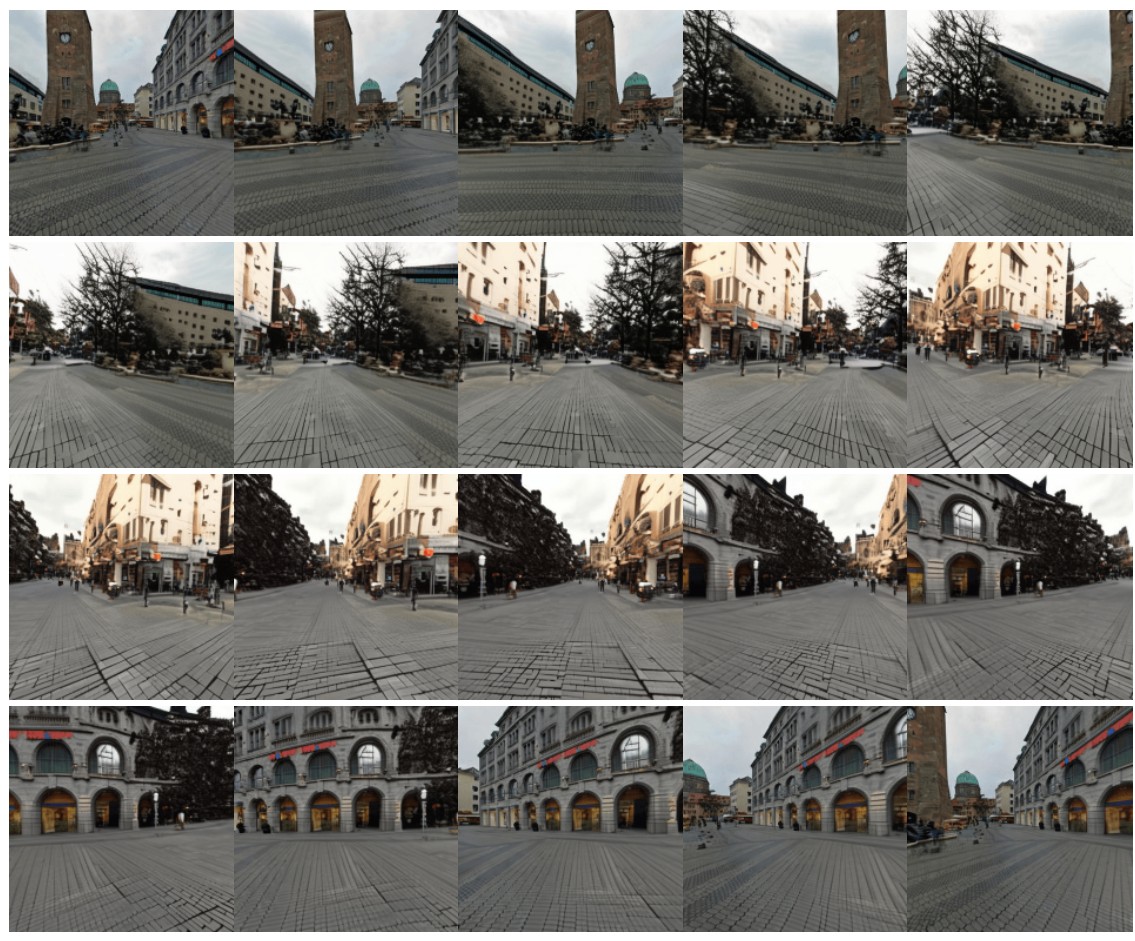

Figure 1: SUN360 Example #1 — Input Views, Stitch, and Upscale

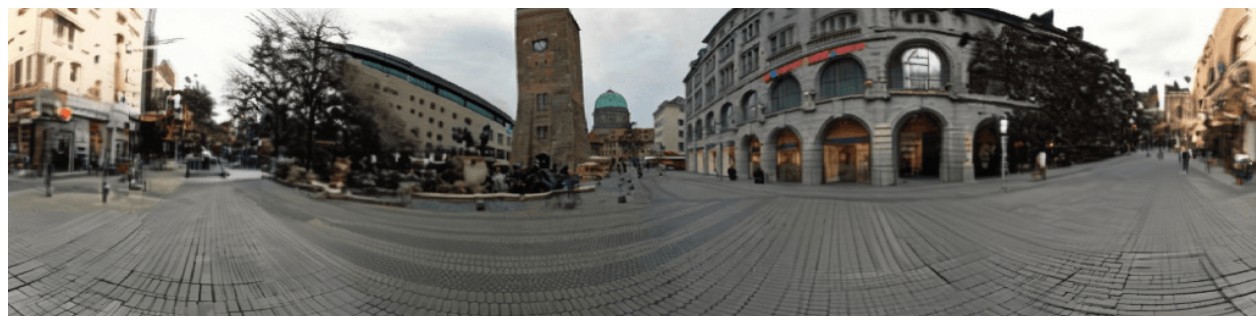

Stitched panorama

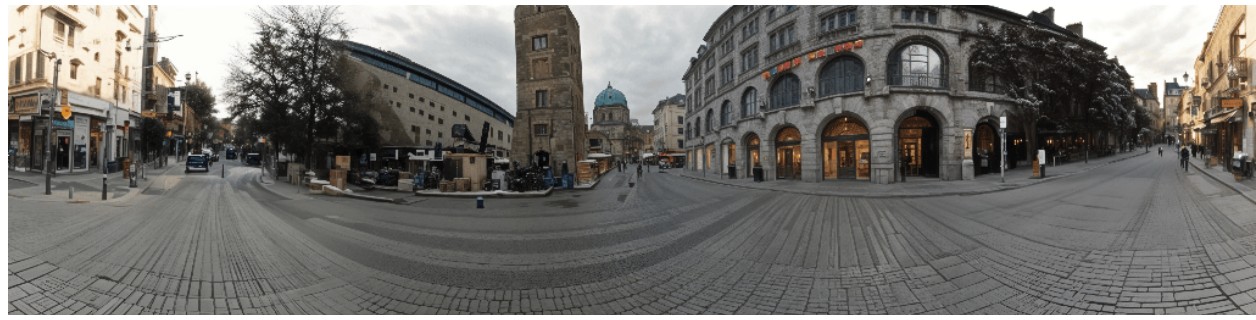

Upscaled final panorama

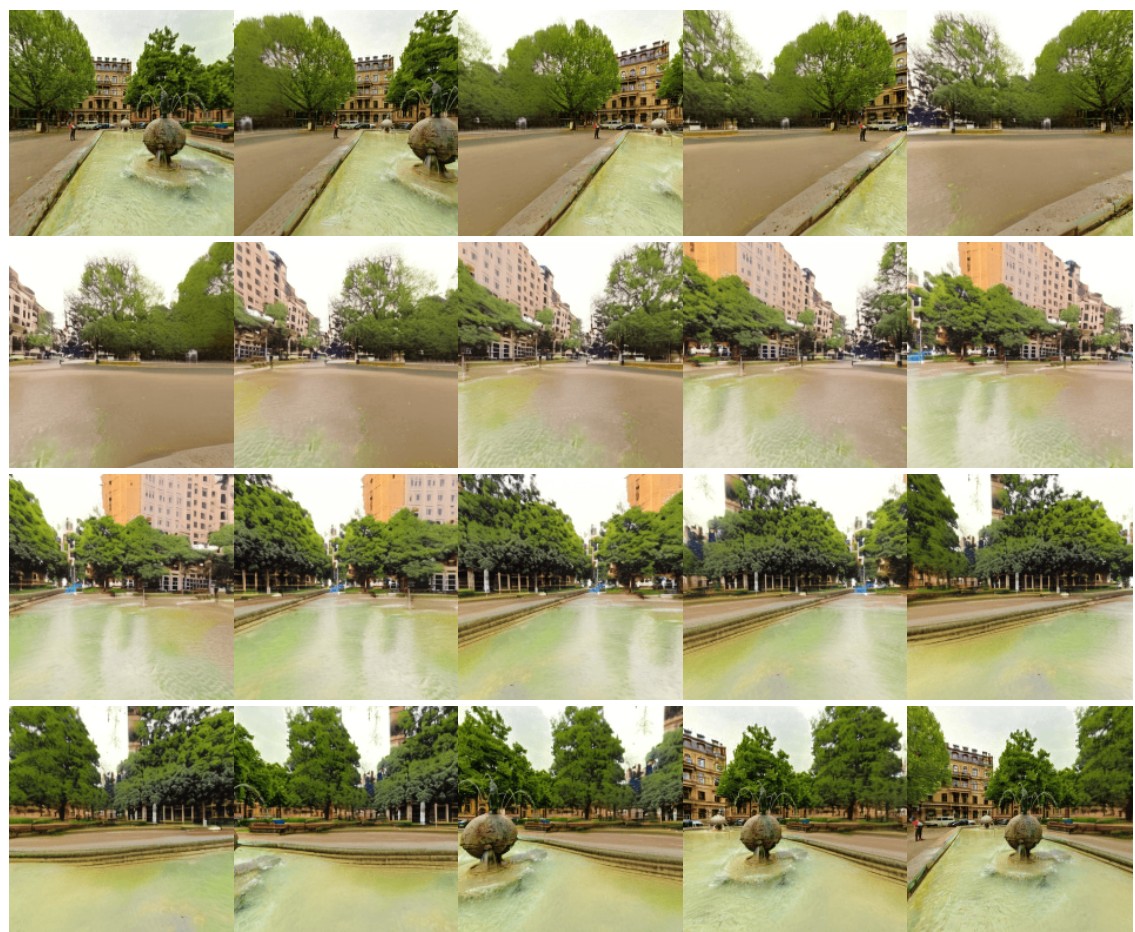

Figure 2: SUN360 Example #2 — Input Views, Stitch, and Upscale

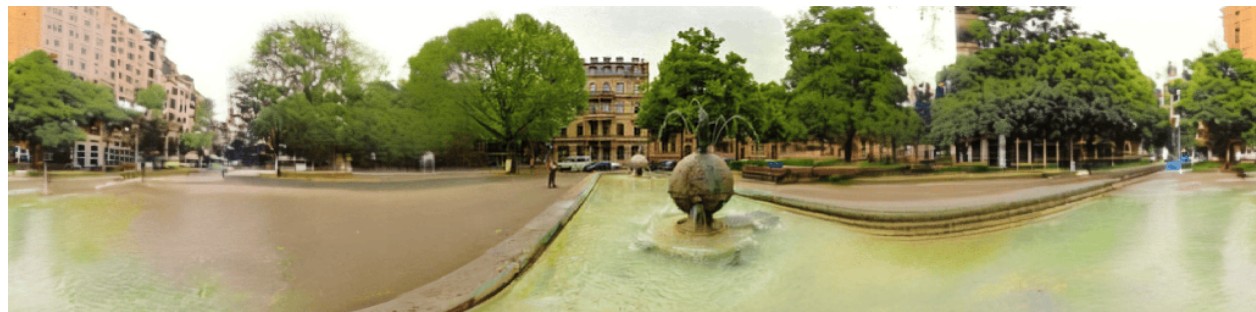

Stitched panorama

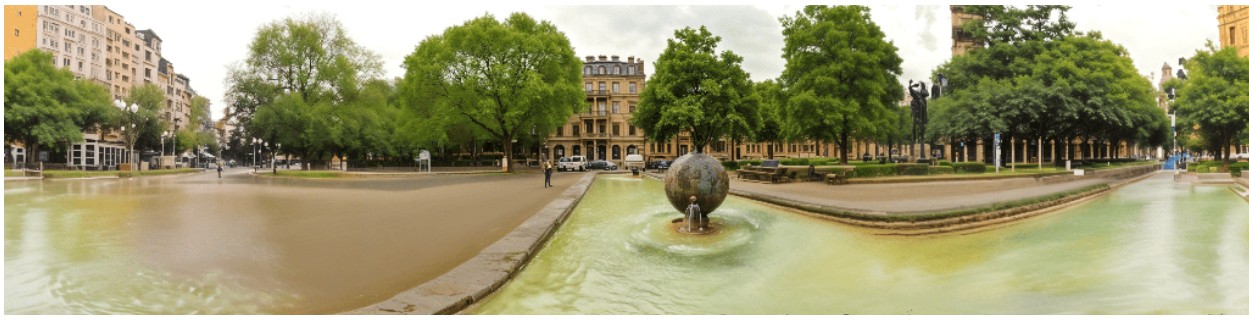

Upscaled final panorama

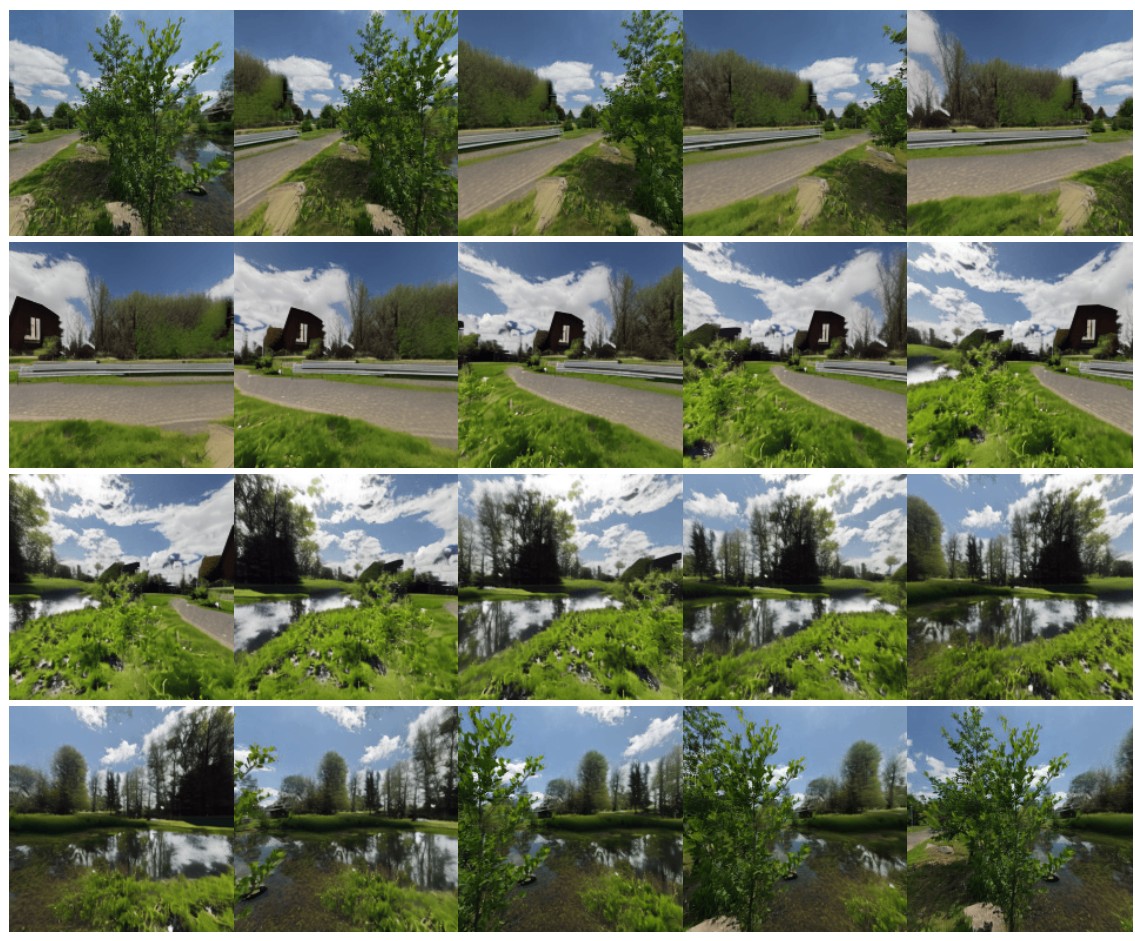

Figure 3: SUN360 Example #3 — Input Views, Stitch, and Upscale

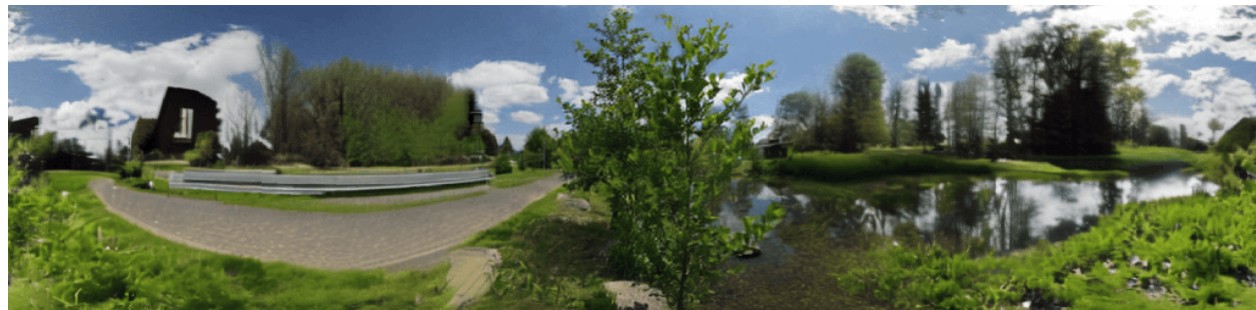

Stitched panorama

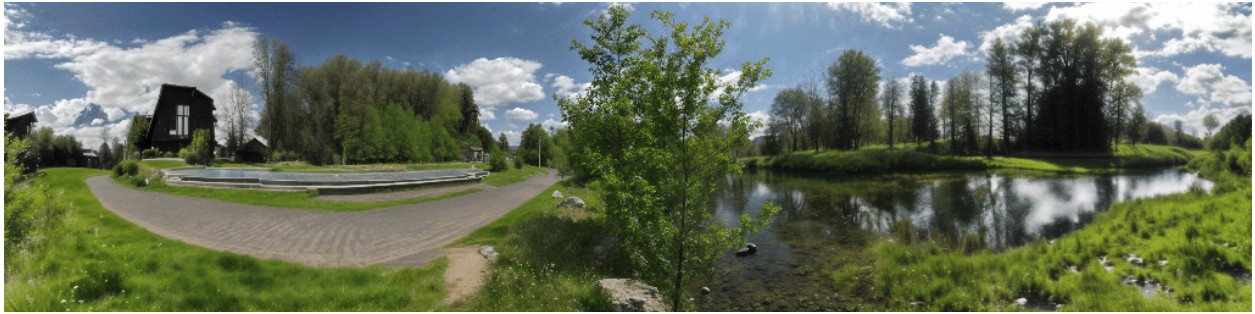

Upscaled final panorama

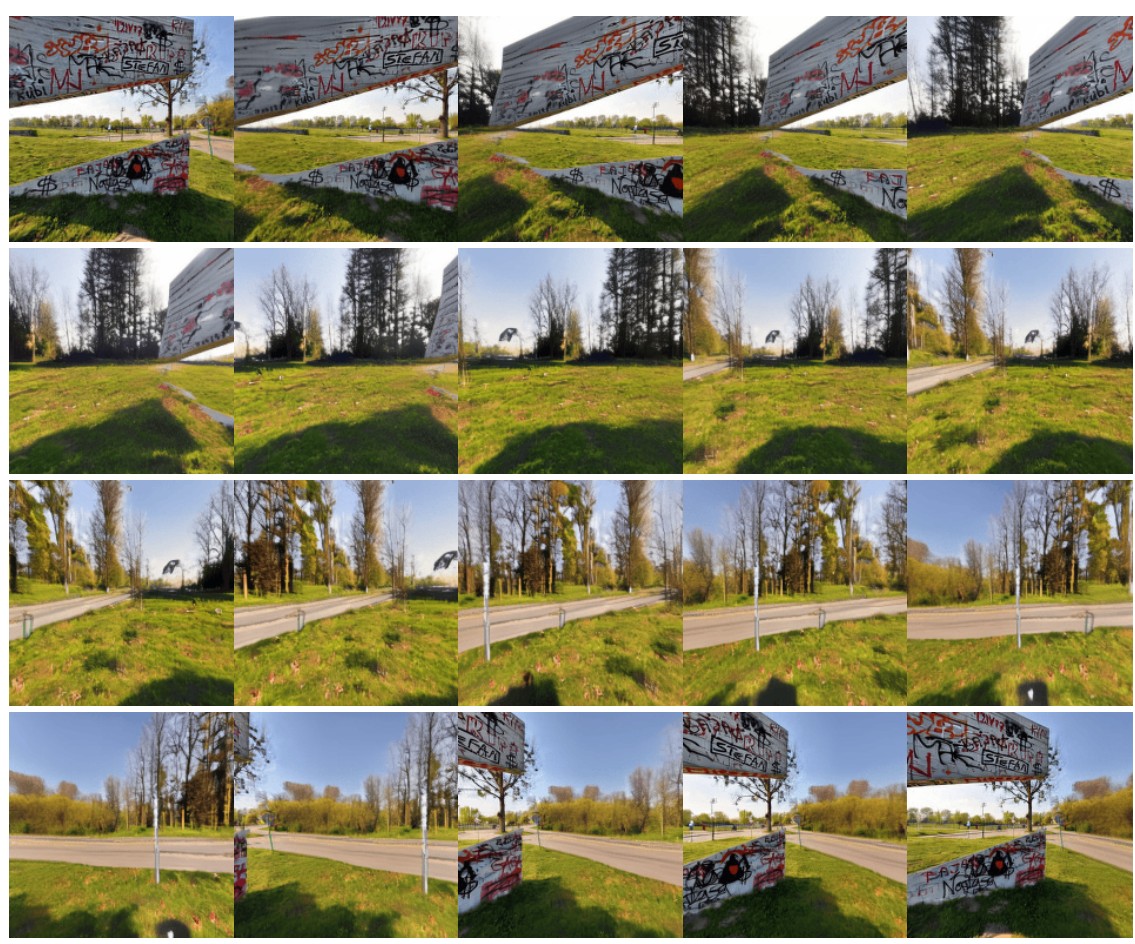

Figure 4: SUN360 Example #4 — Input Views, Stitch, and Upscale

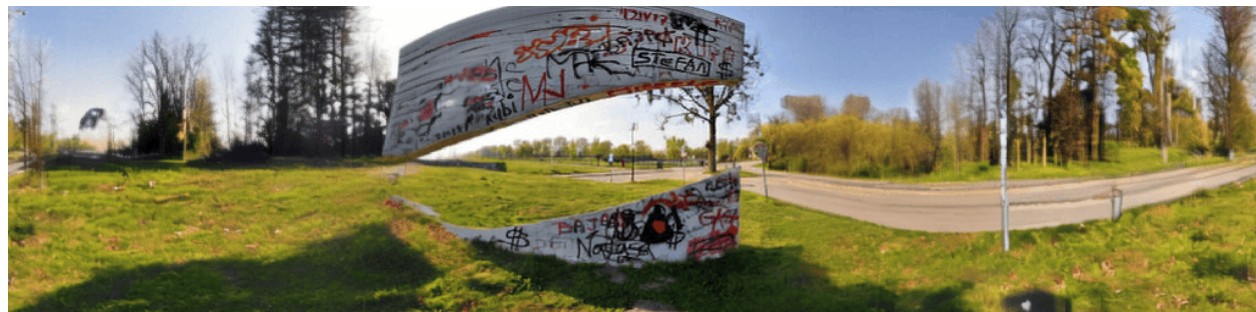

Stitched panorama

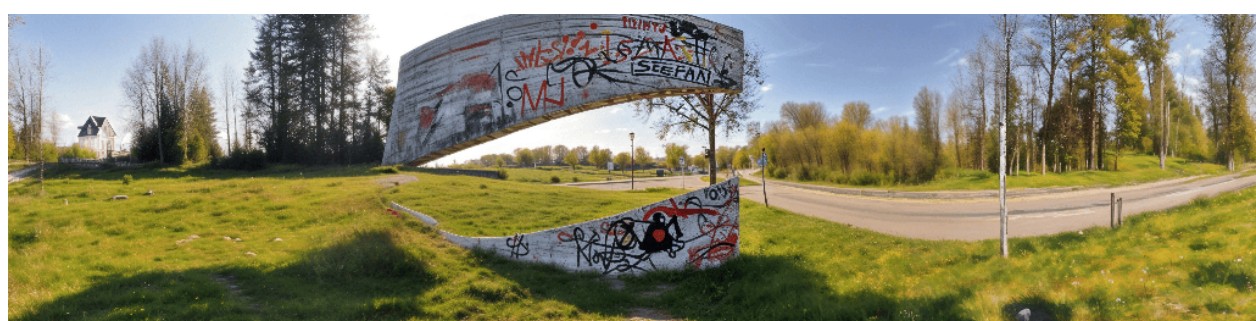

Upscaled final panorama

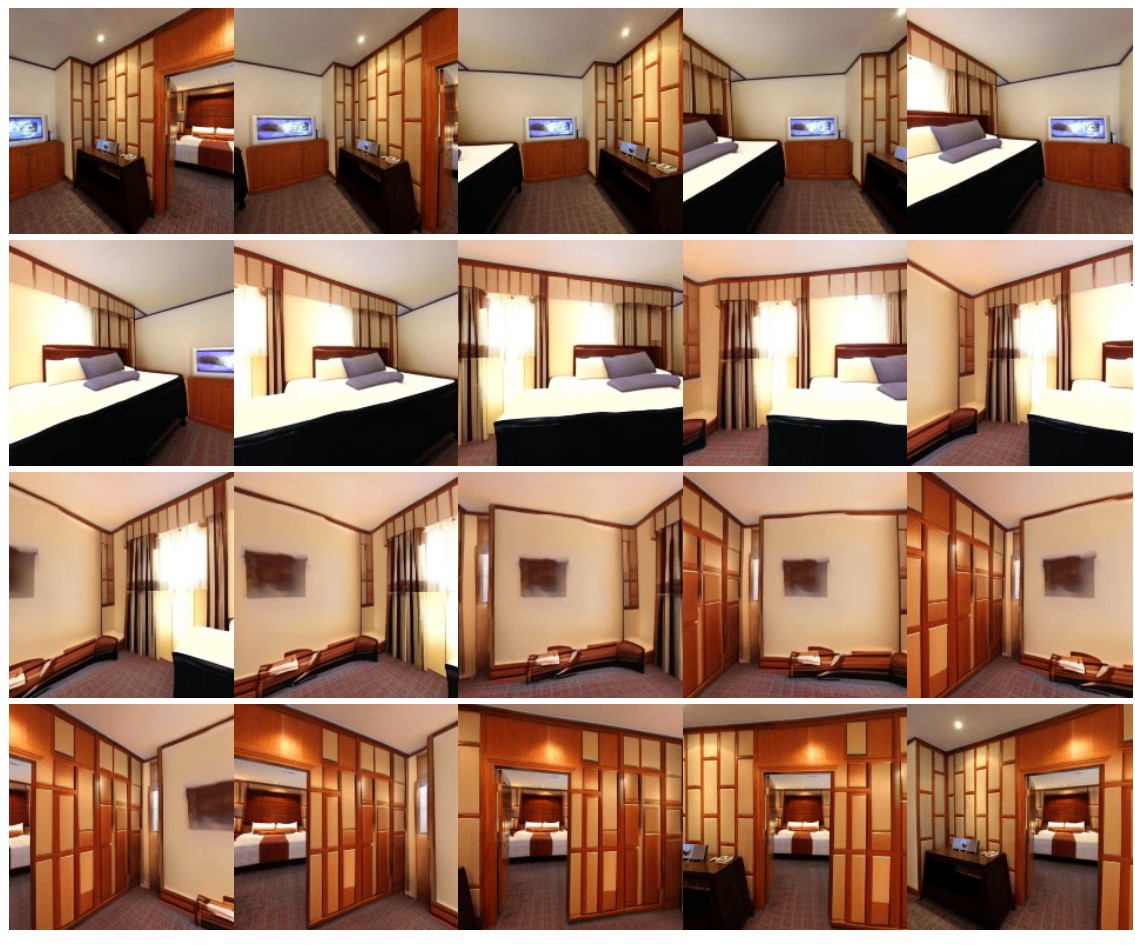

Figure 5: SUN360 Example #5 — Input Views, Stitch, and Upscale

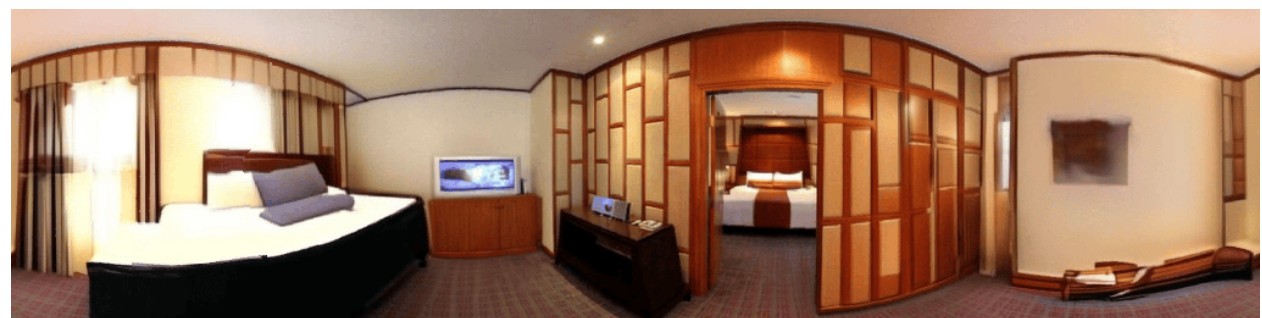

Stitched panorama

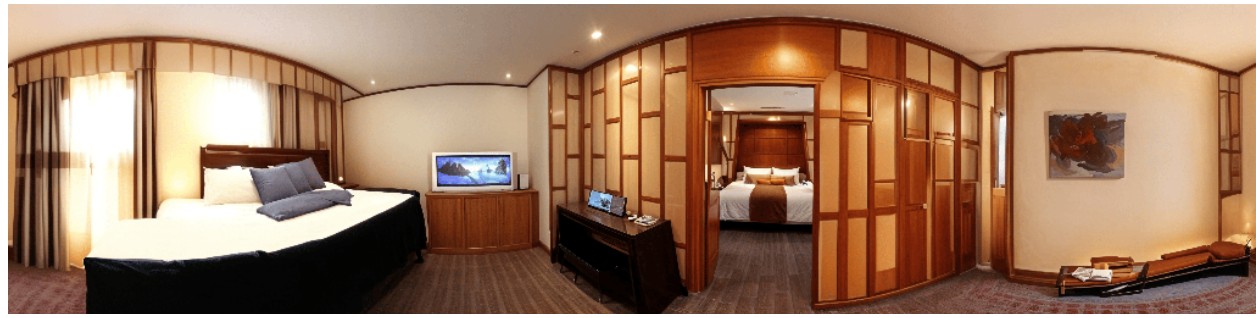

Upscaled final panorama

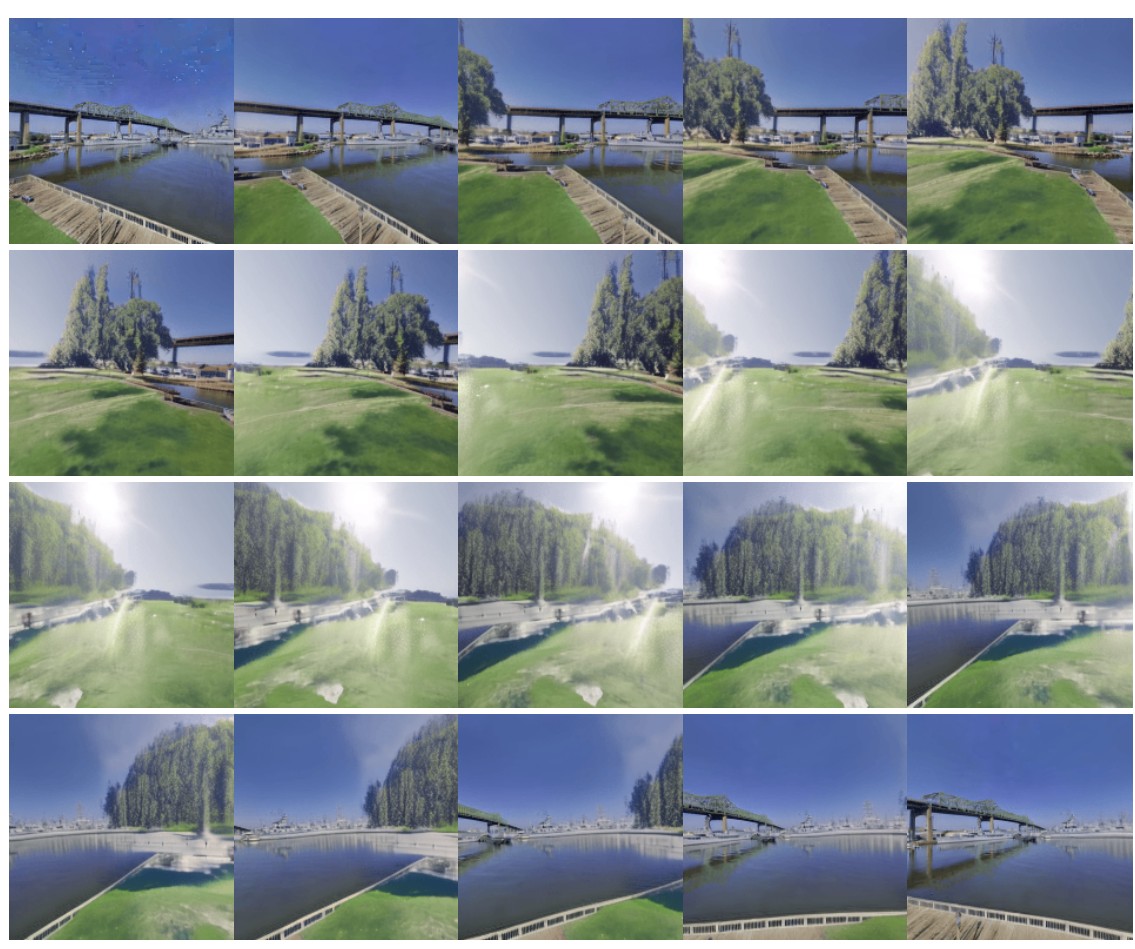

Figure 6: SUN360 Example #6 — Input Views, Stitch, and Upscale

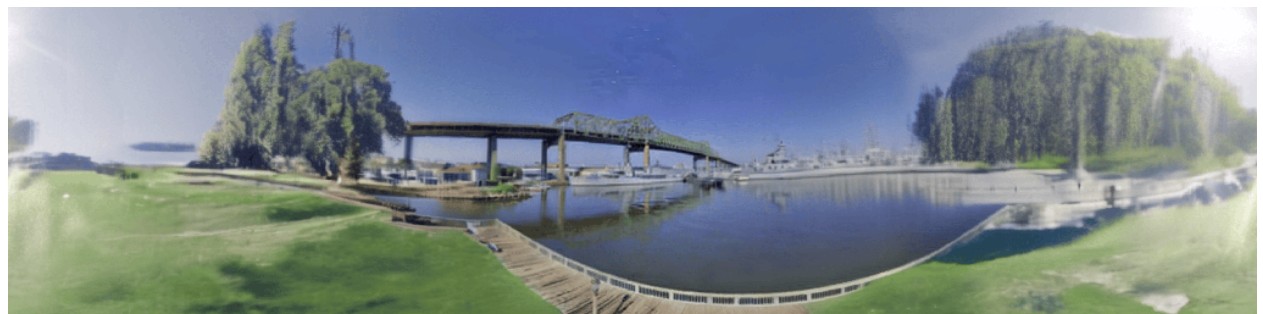

Stitched panorama

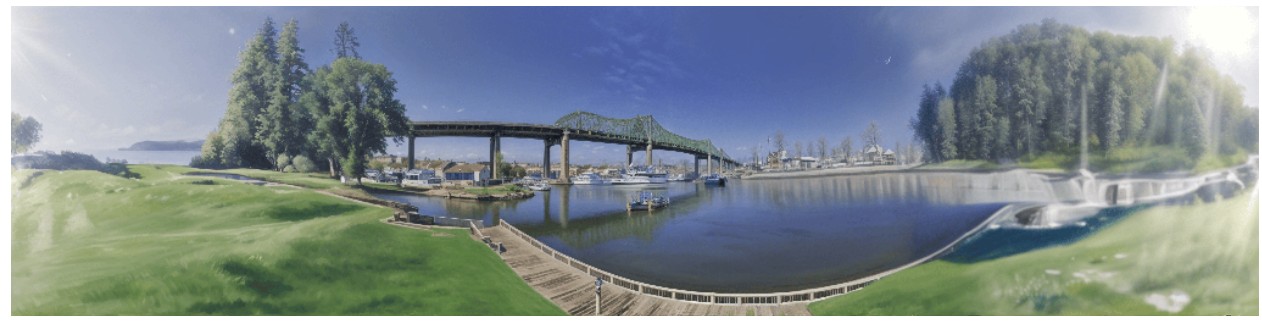

Upscaled final panorama

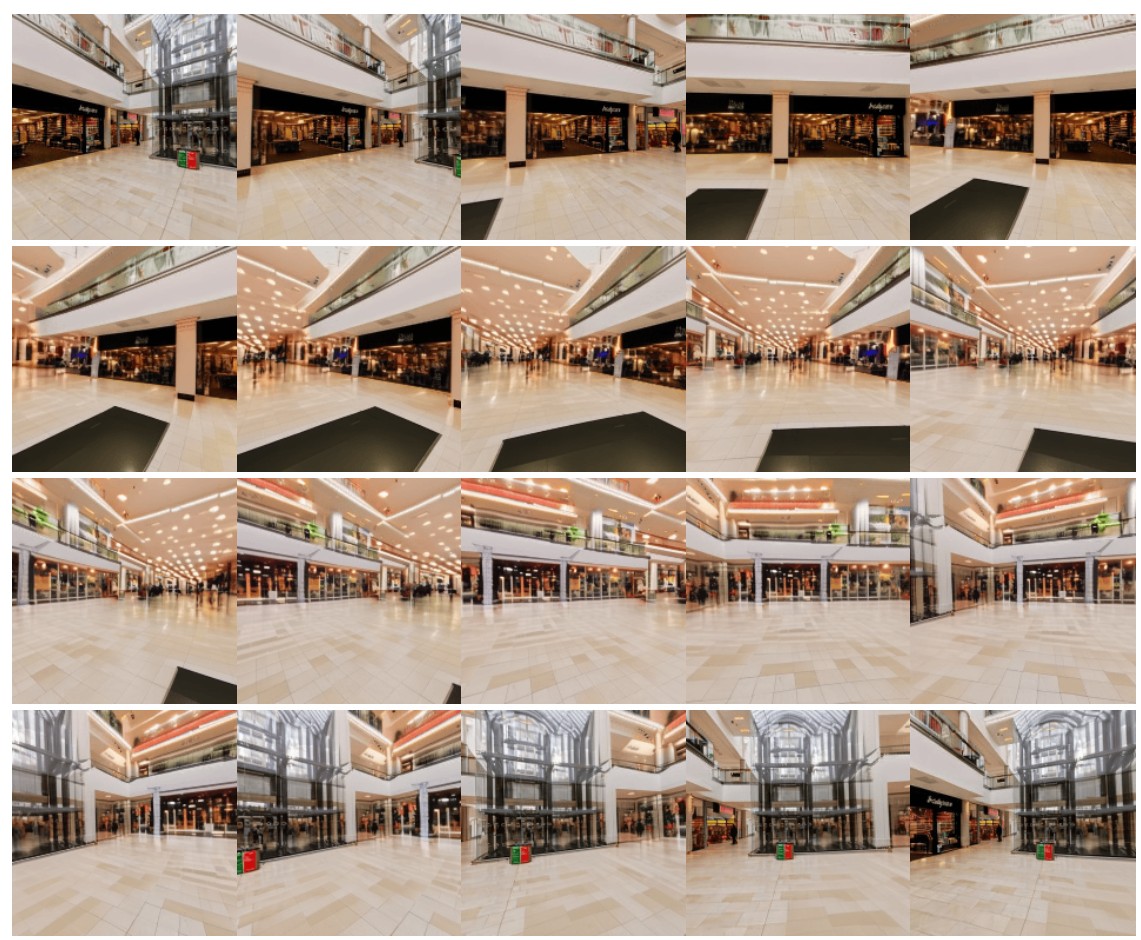

Figure 7: SUN360 Example #7 — Input Views, Stitch, and Upscale

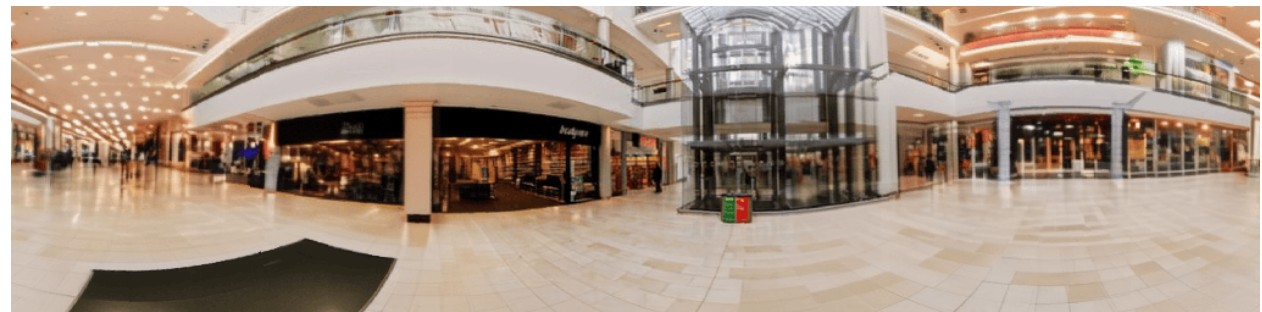

Stitched panorama

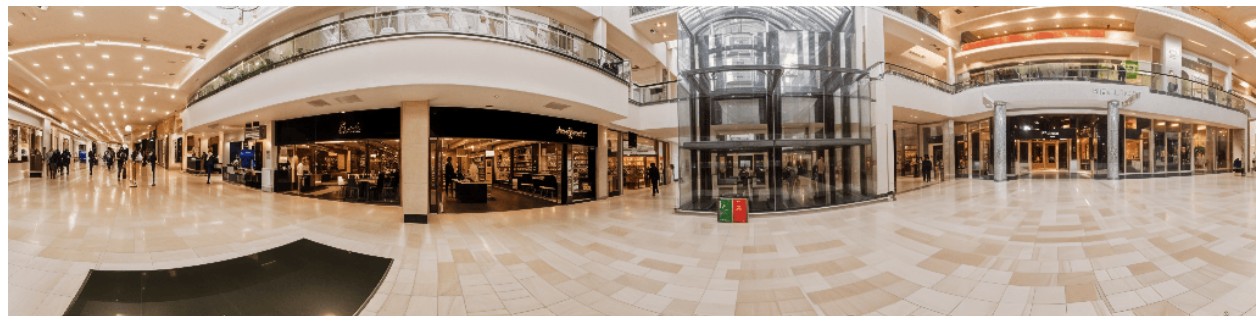

Upscaled final panorama

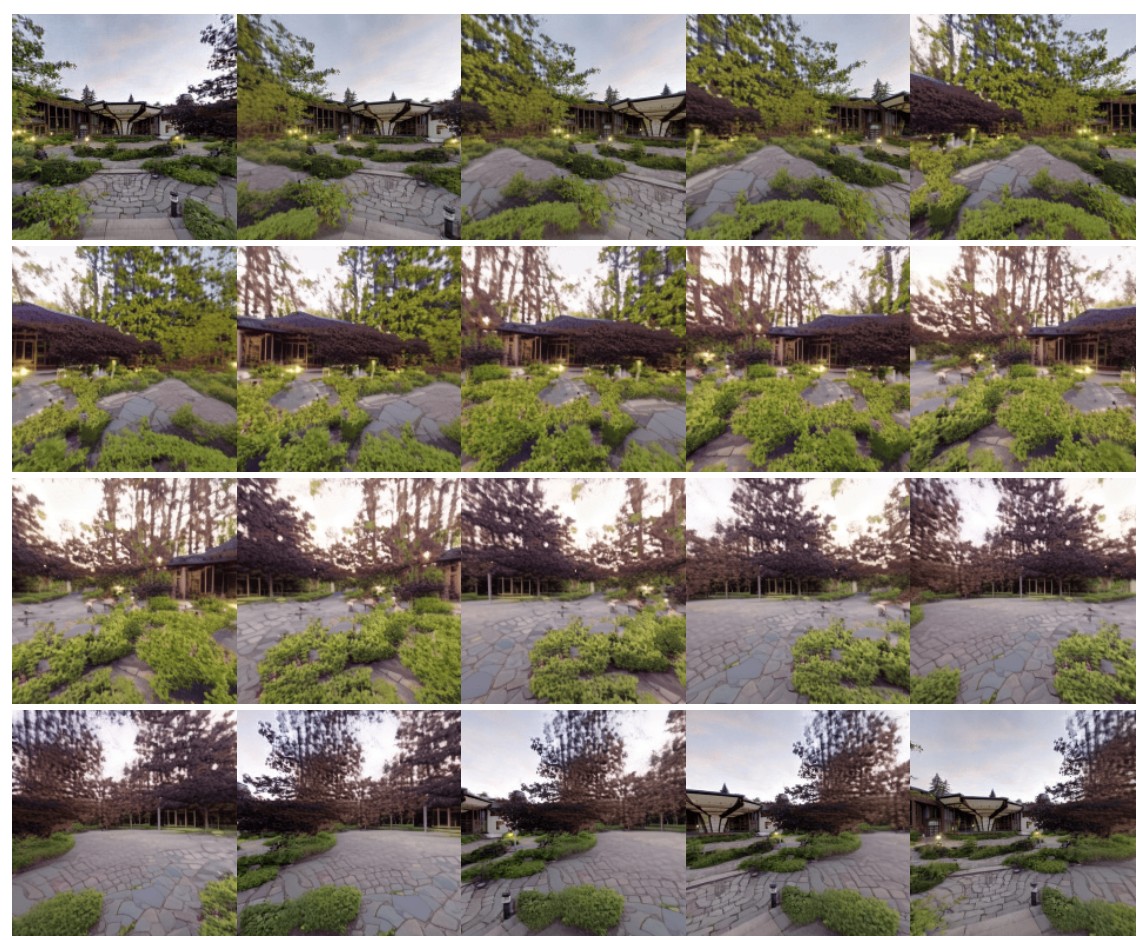

Figure 8: SUN360 Example #8 — Input Views, Stitch, and Upscale

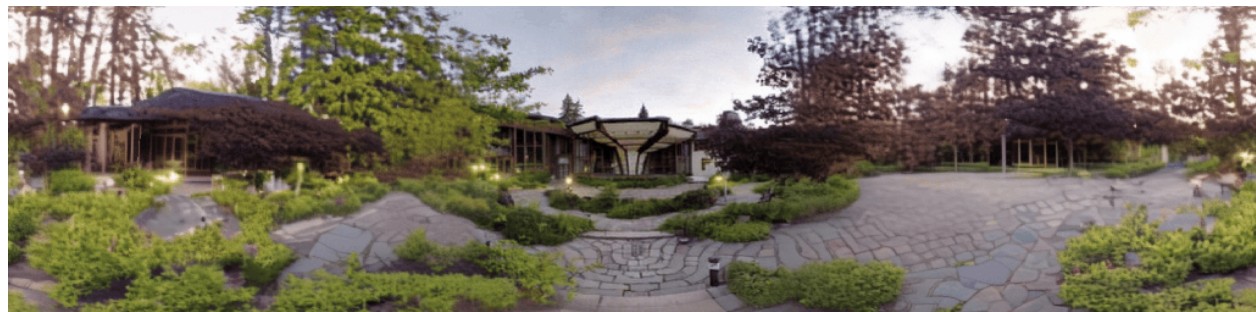

Stitched panorama

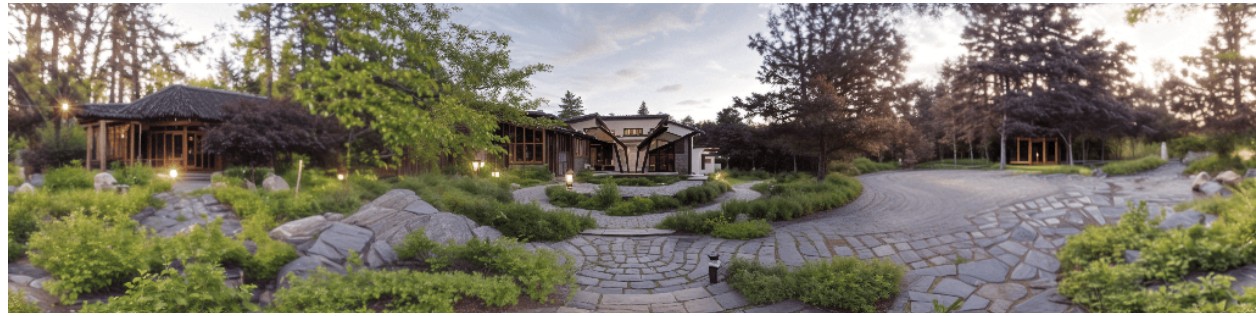

Upscaled final panorama

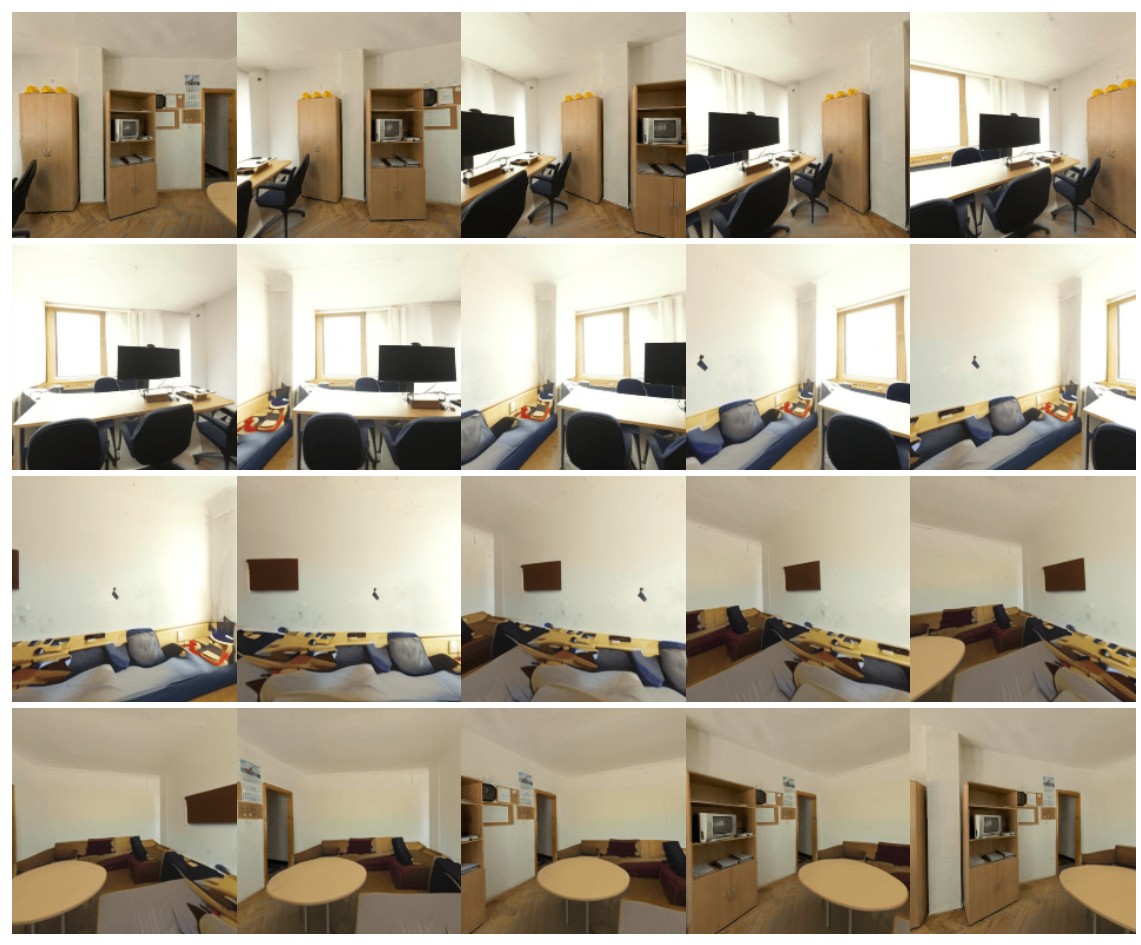

Figure 9: SUN360 Example #9 — Input Views, Stitch, and Upscale

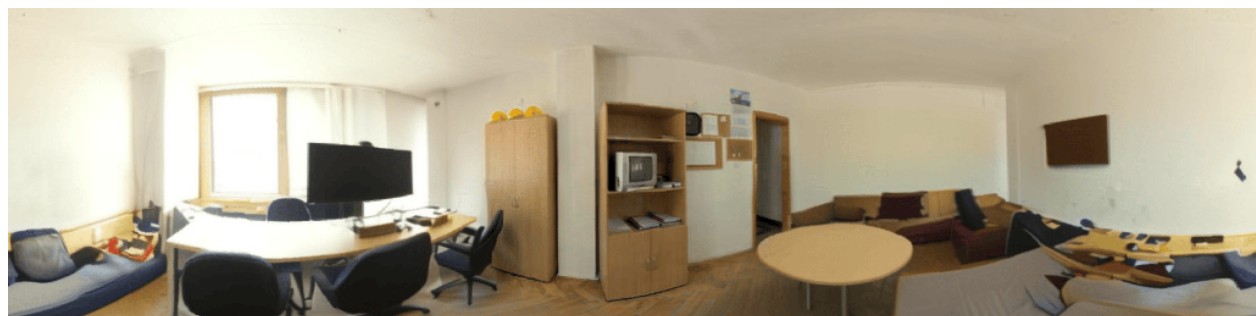

Stitched panorama

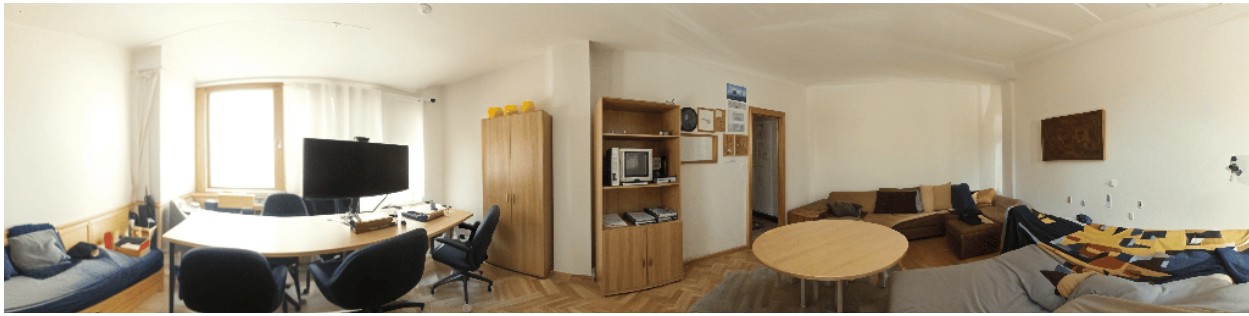

Upscaled final panorama

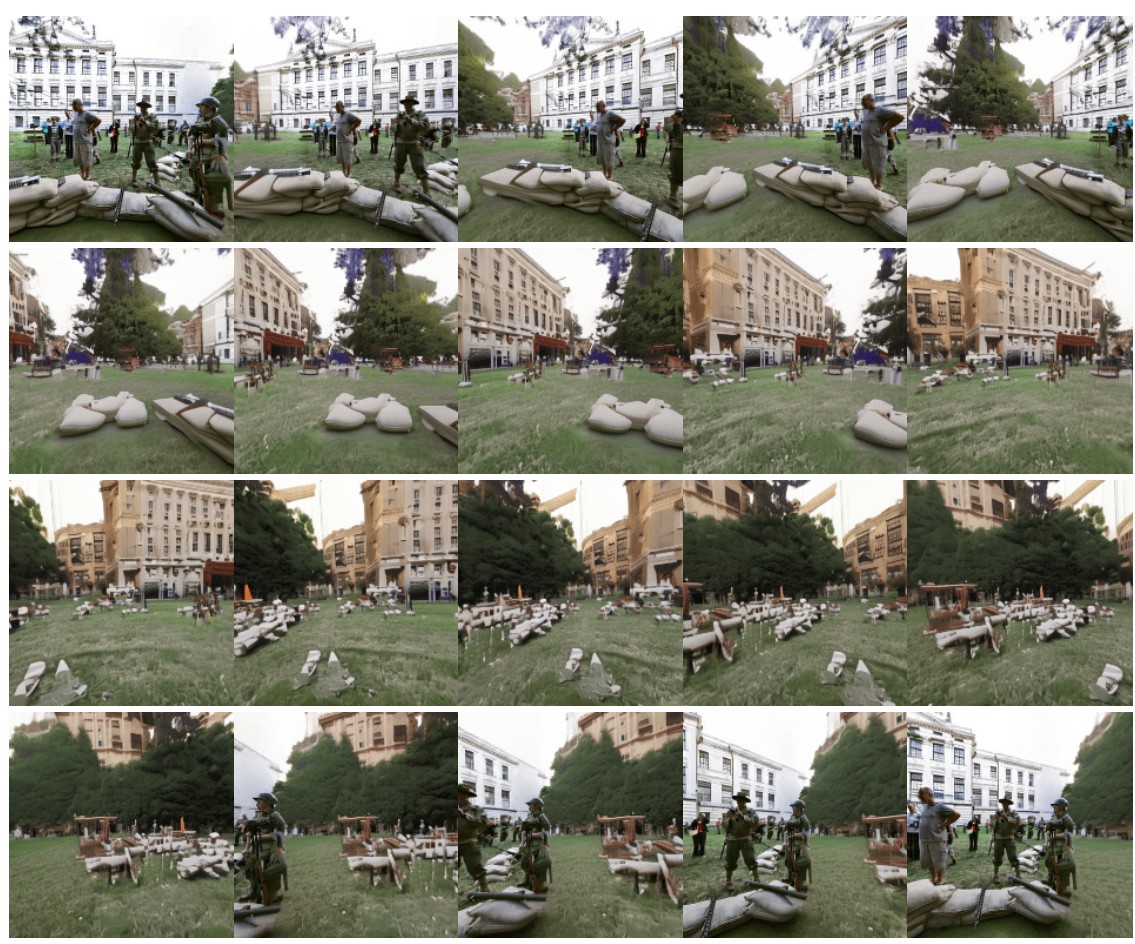

Figure 10: SUN360 Example #10 — Input Views, Stitch, and Upscale

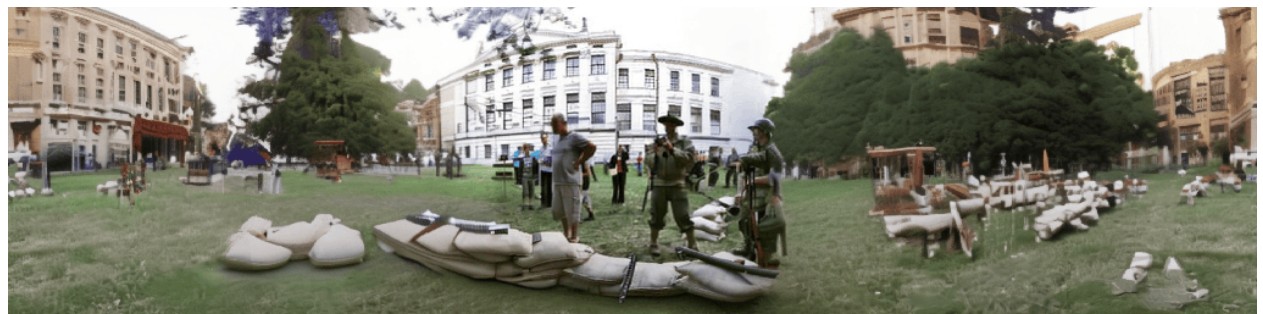

Stitched panorama

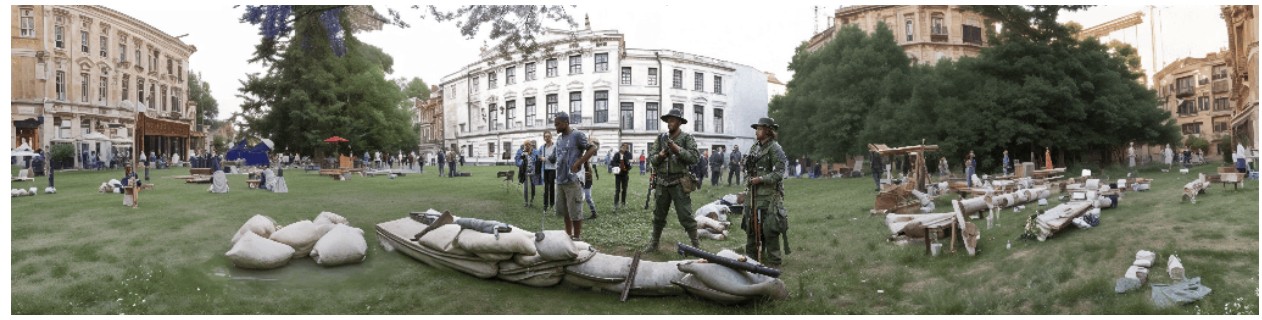

Upscaled final panorama

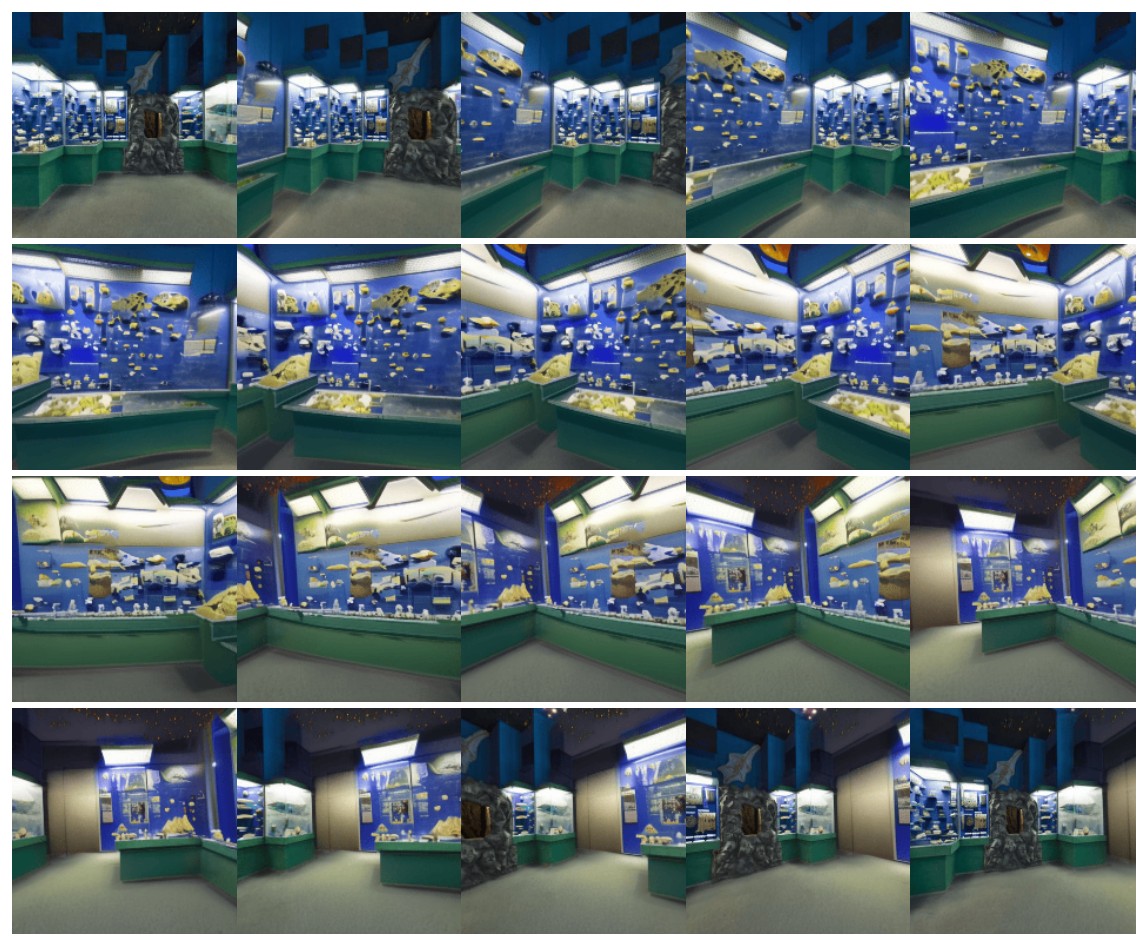

Figure 11: SUN360 Example #11 — Input Views, Stitch, and Upscale

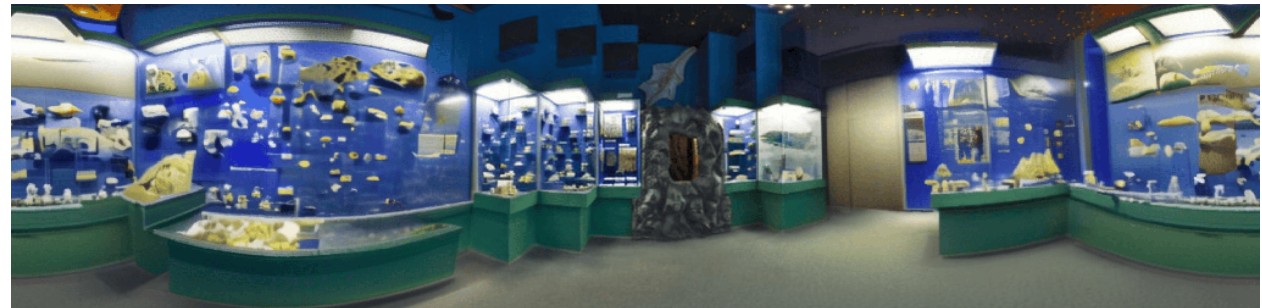

Stitched panorama

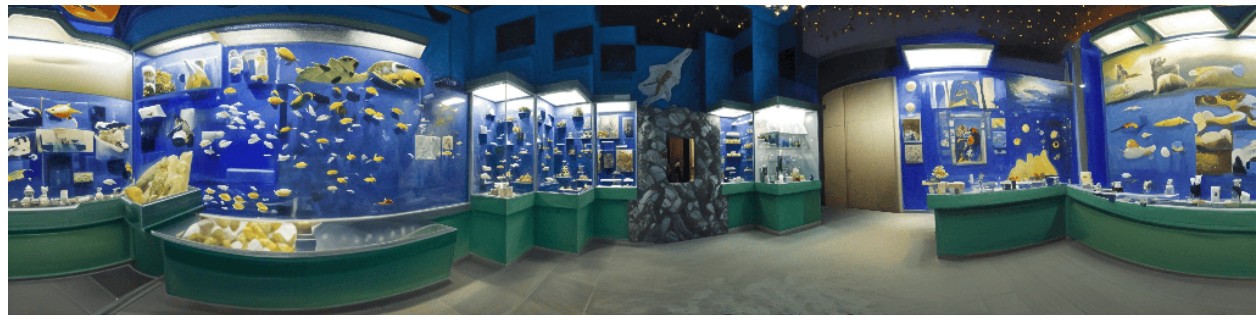

Upscaled final panorama

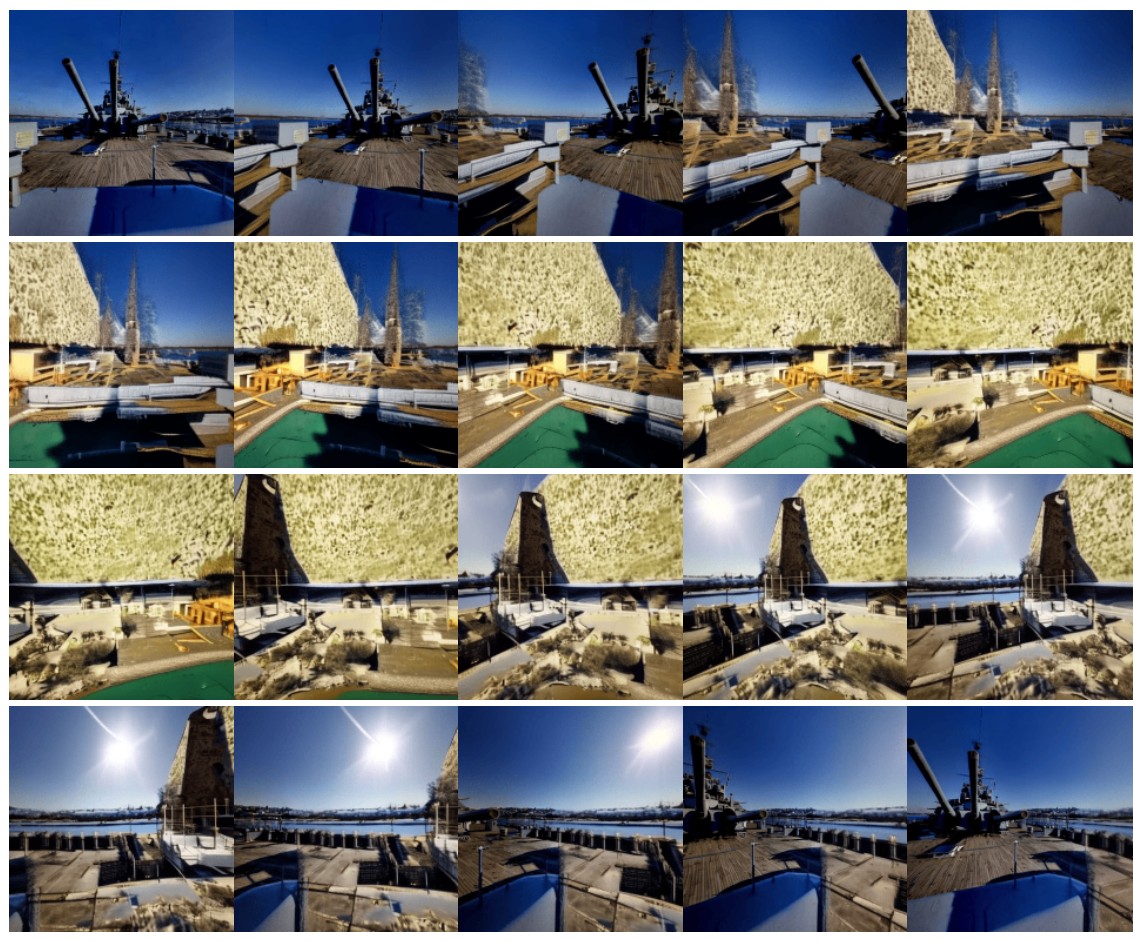

Figure 12: SUN360 Example #12 — Input Views, Stitch, and Upscale

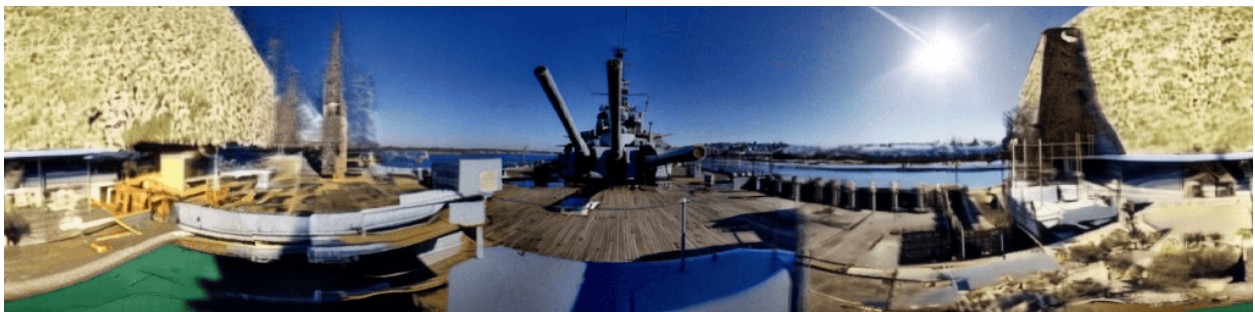

Stitched panorama

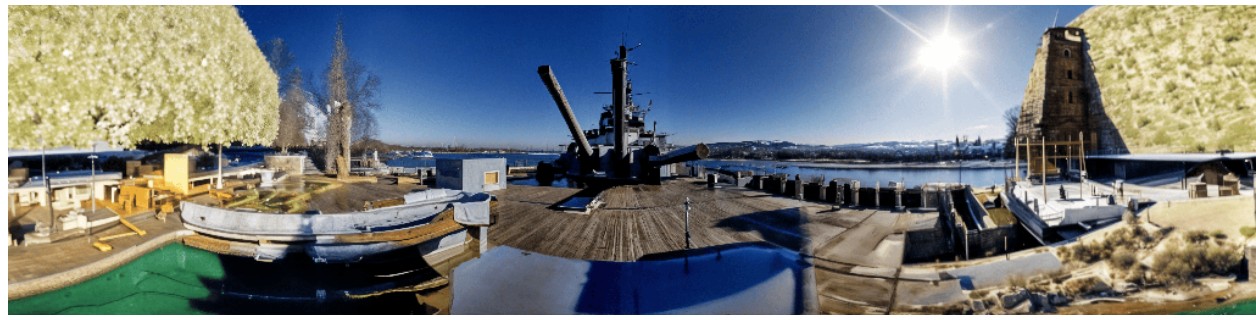

Upscaled final panorama

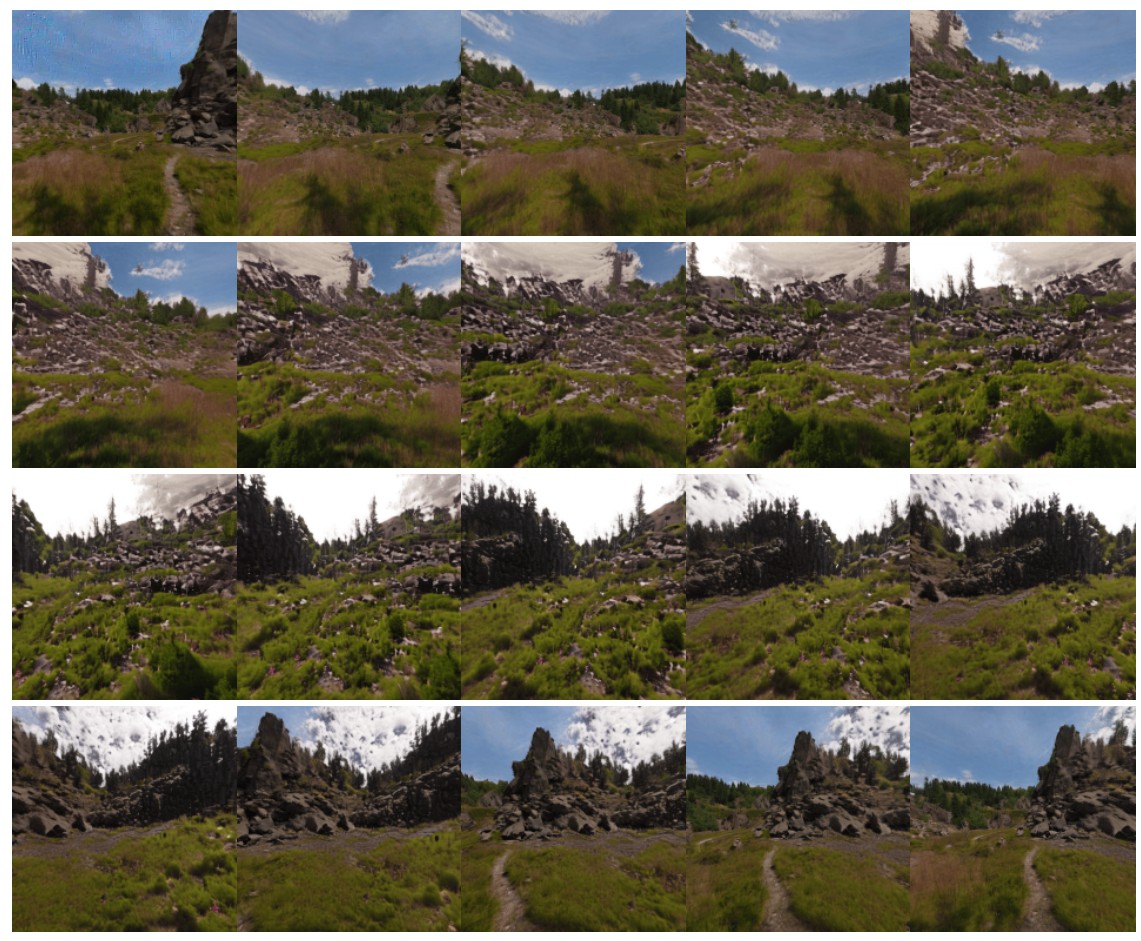

Figure 13: SUN360 Example #13 — Input Views, Stitch, and Upscale

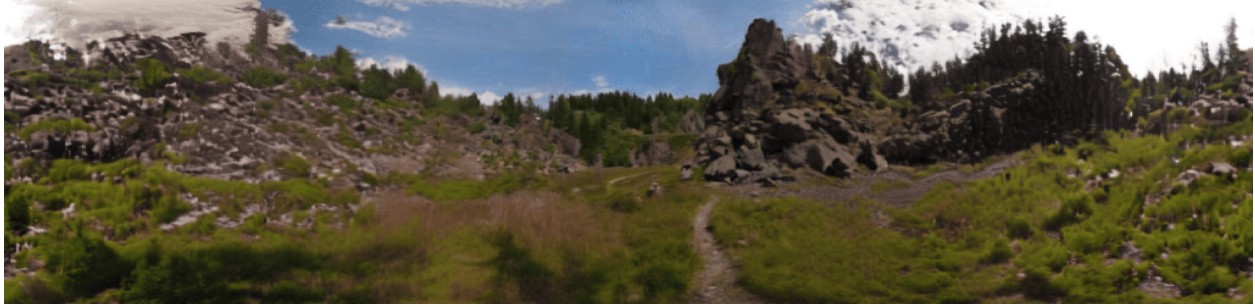

Stitched panorama

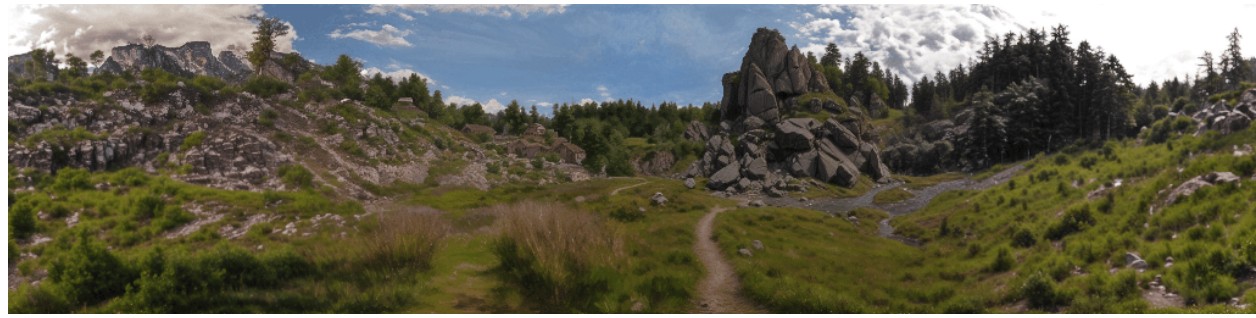

Upscaled final panorama

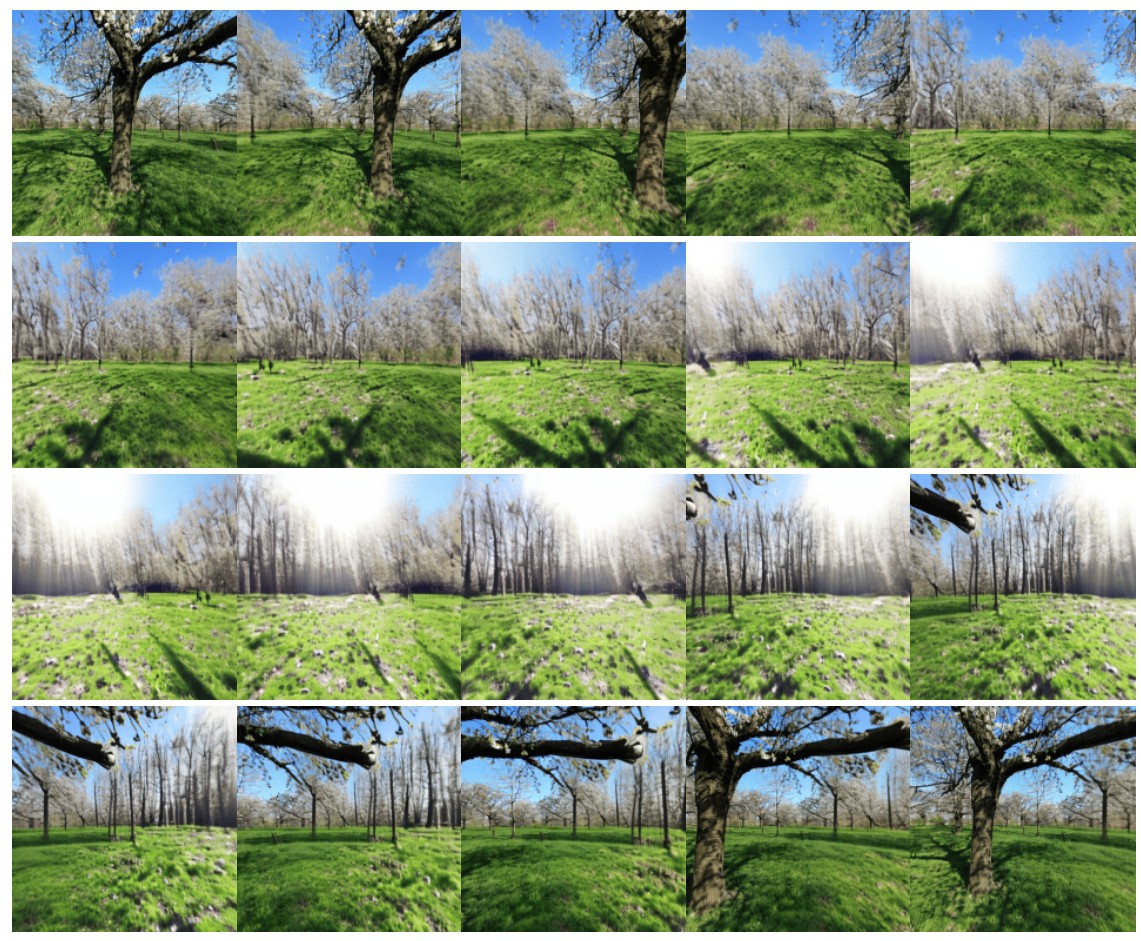

Figure 14: SUN360 Example #14 — Input Views, Stitch, and Upscale

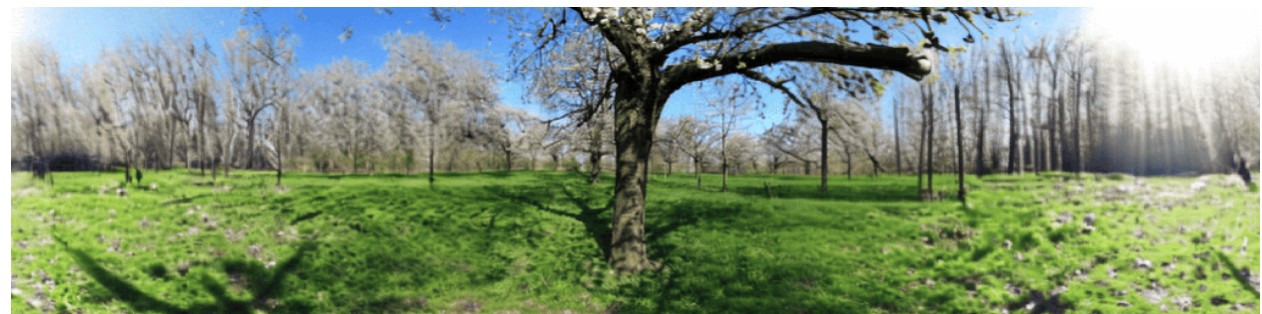

Stitched panorama

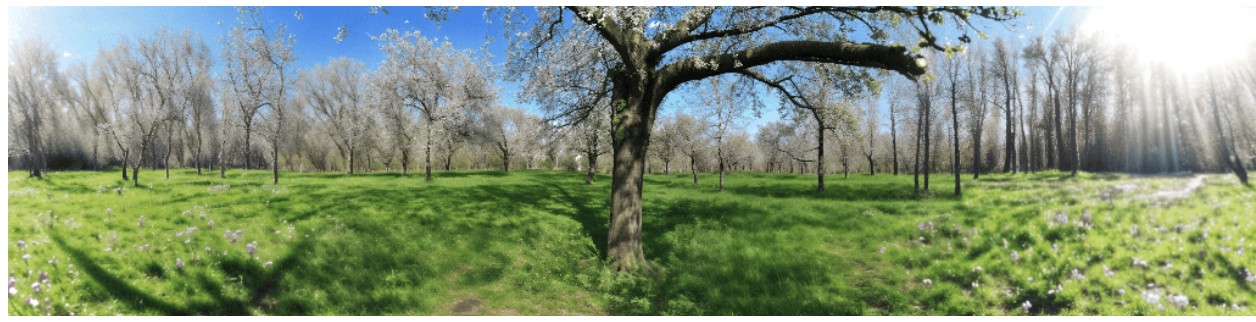

Upscaled final panorama

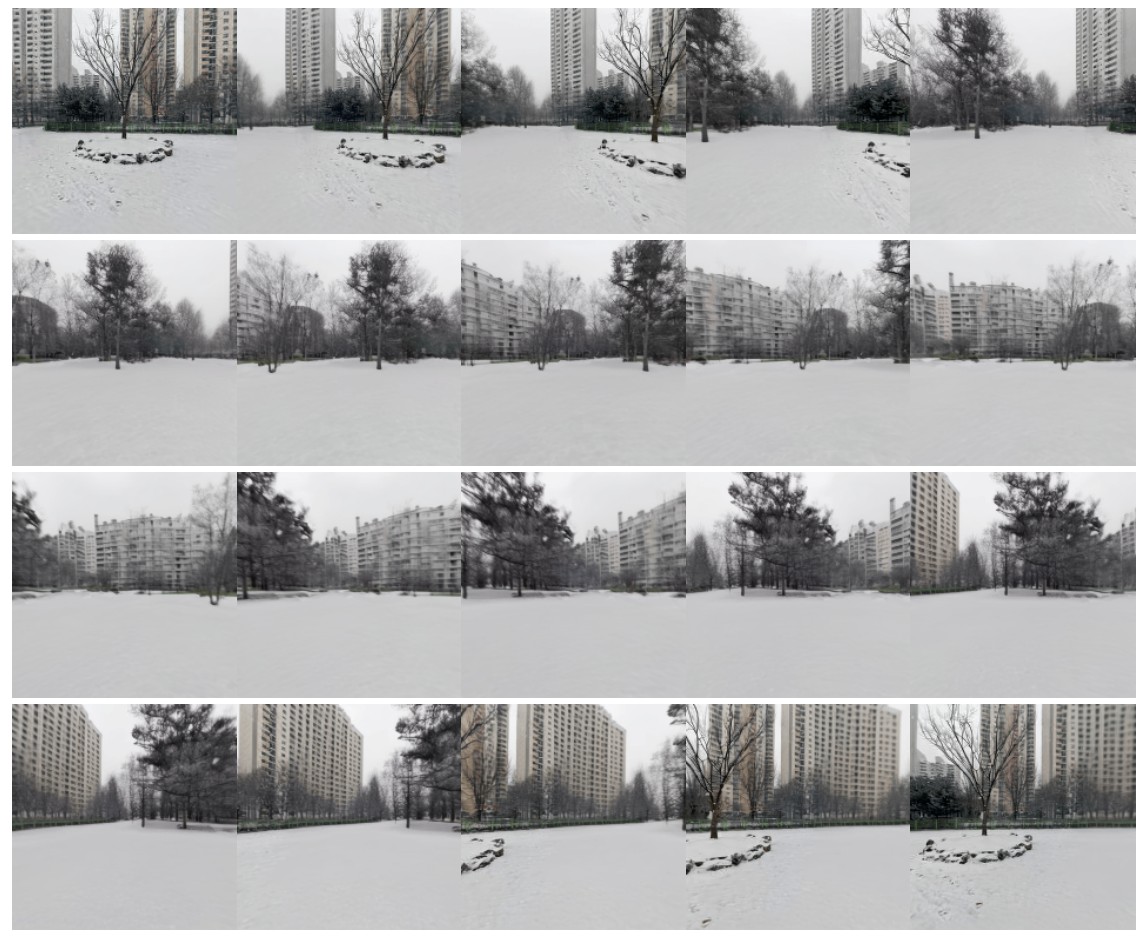

Figure 15: SUN360 Example #15 — Input Views, Stitch, and Upscale

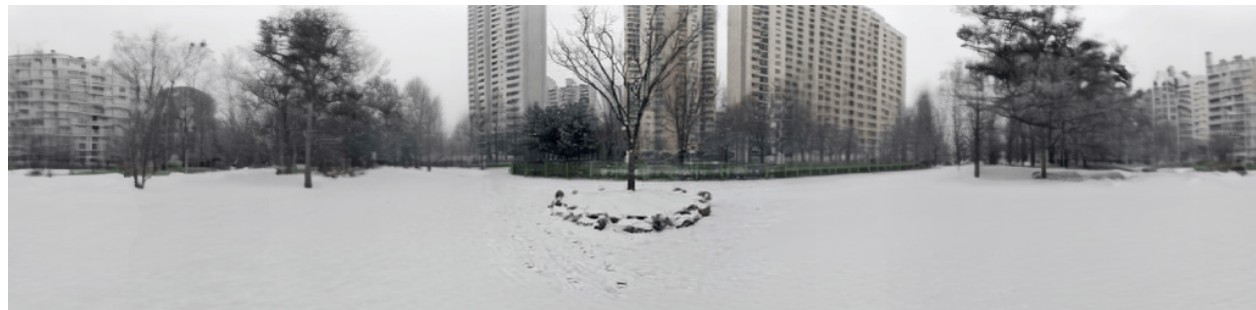

Stitched panorama

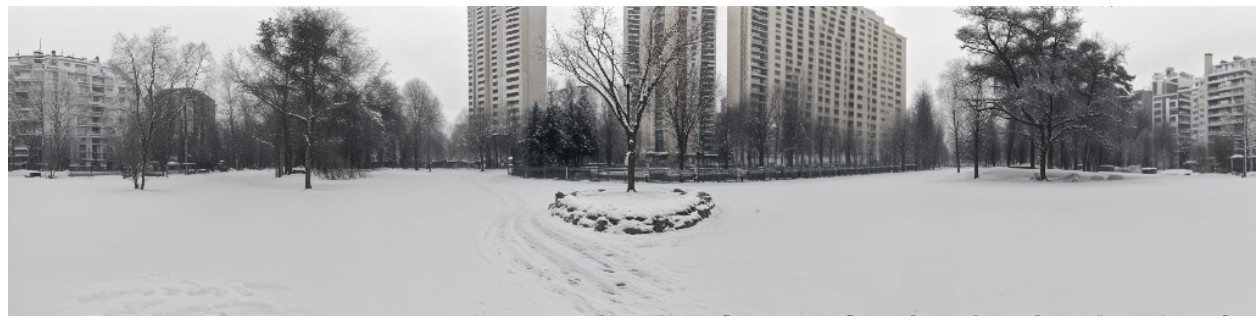

Upscaled final panorama

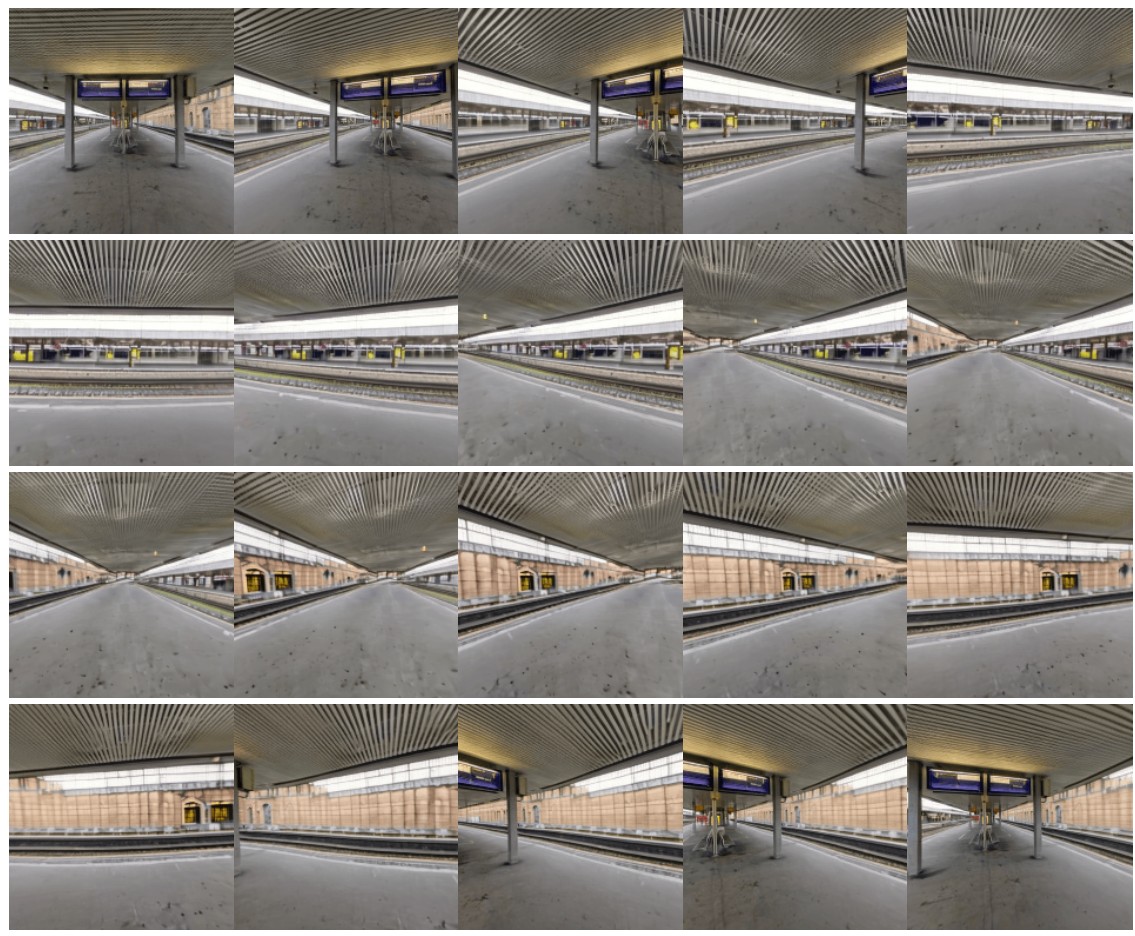

Figure 16: SUN360 Example #16 — Input Views, Stitch, and Upscale

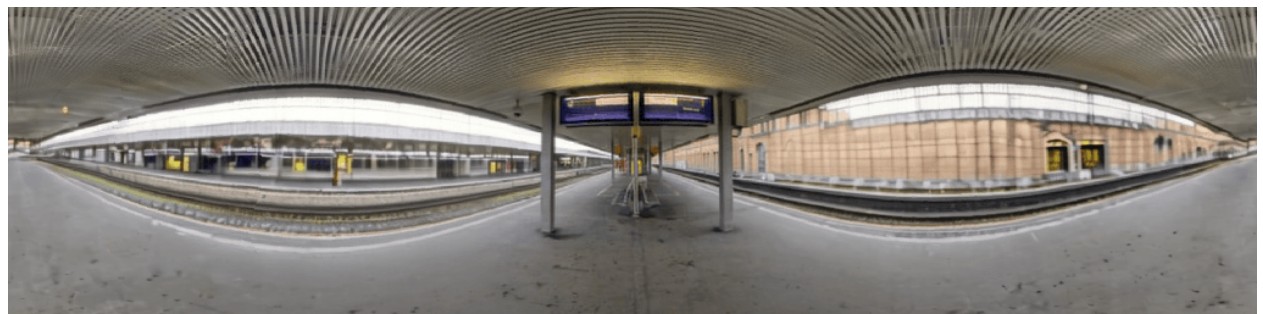

Stitched panorama

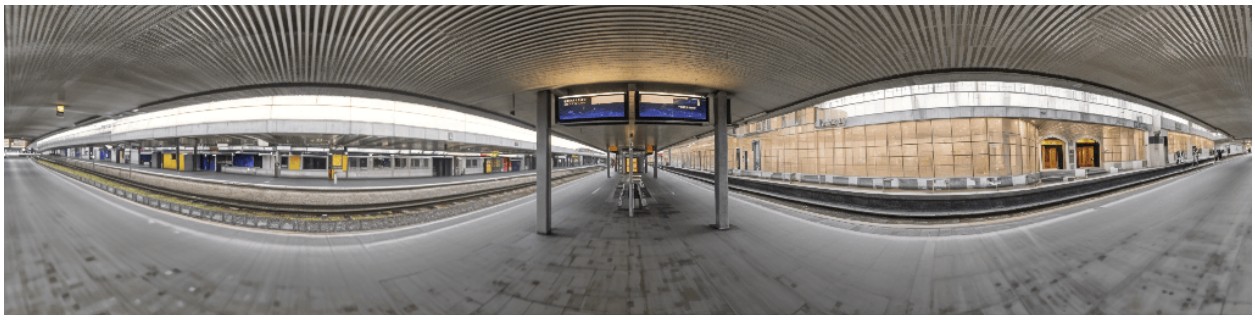

Upscaled final panorama

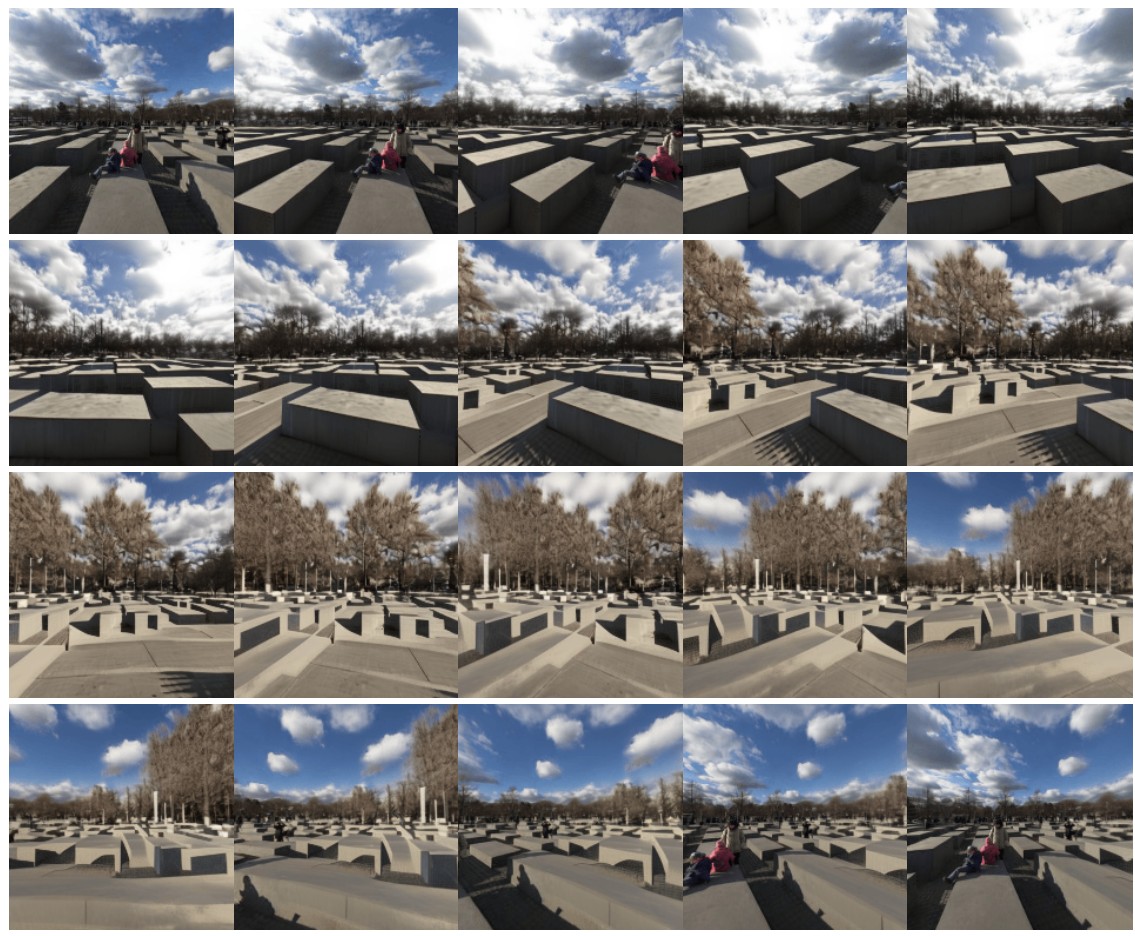

Figure 17: SUN360 Example #17 — Input Views, Stitch, and Upscale

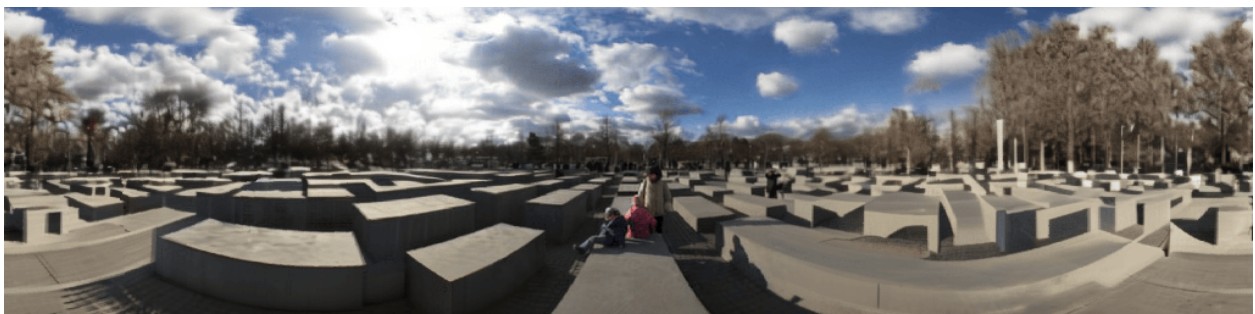

Stitched panorama

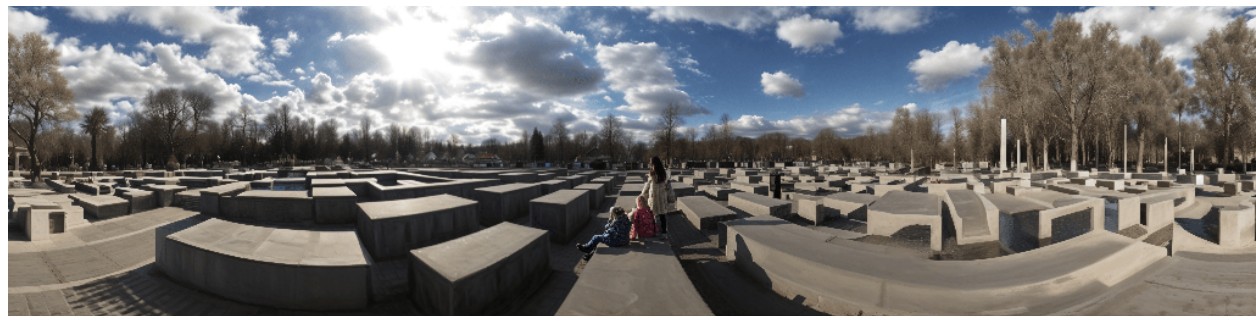

Upscaled final panorama

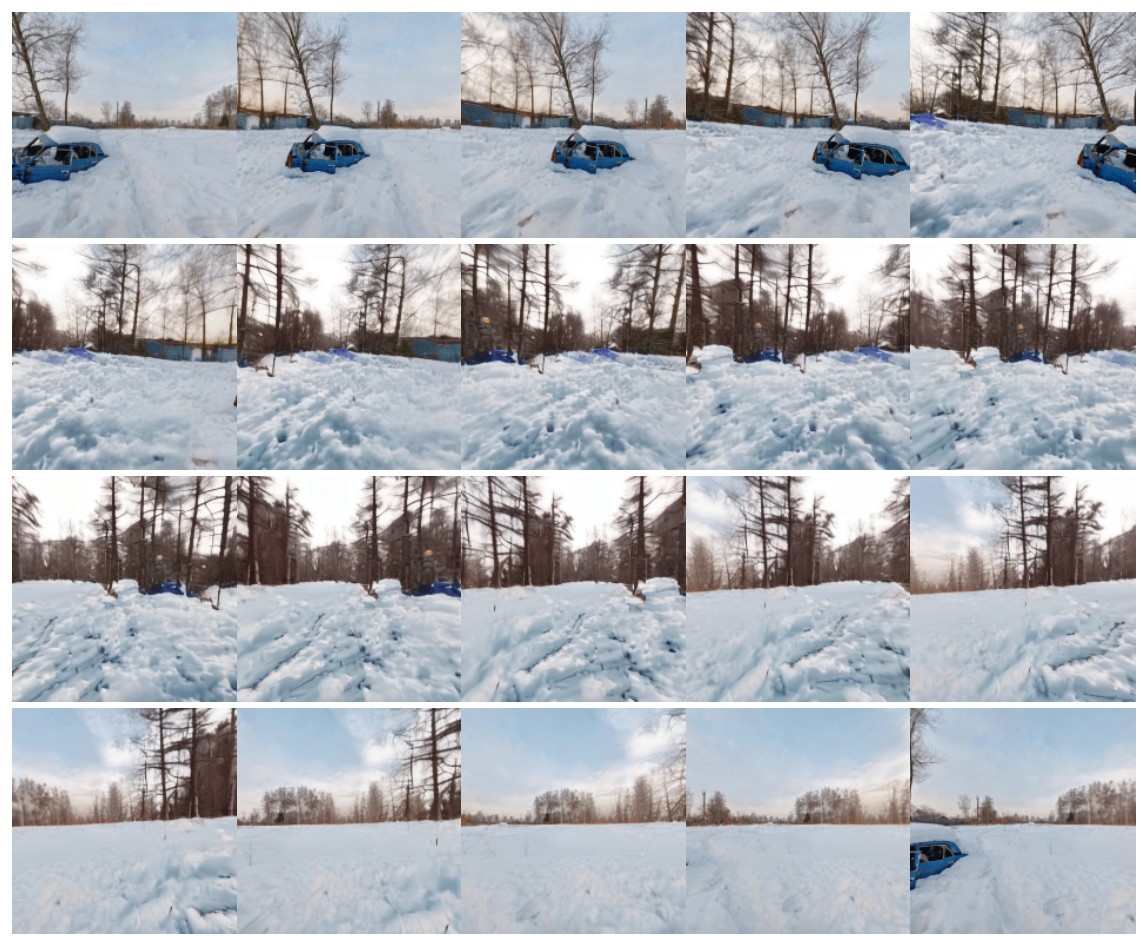

Figure 18: SUN360 Example #18 — Input Views, Stitch, and Upscale

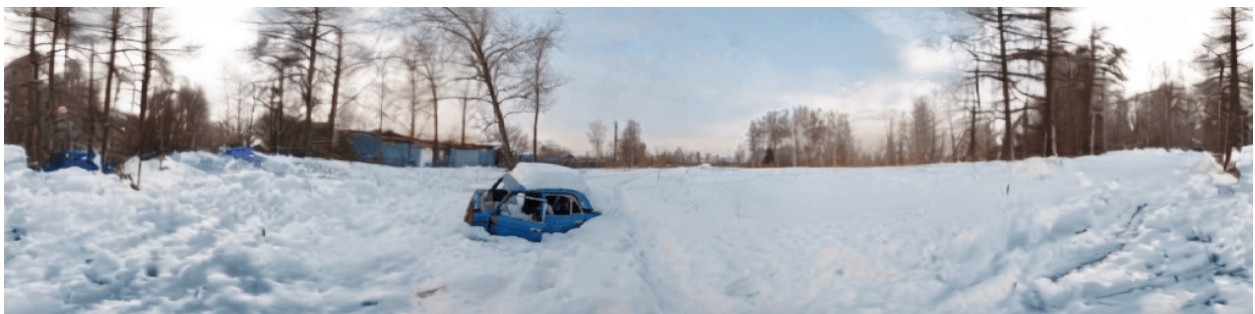

Stitched panorama

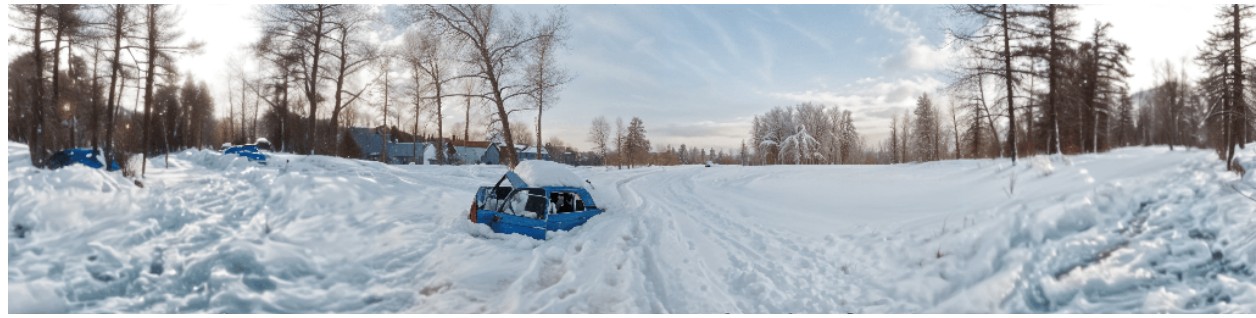

Upscaled final panorama

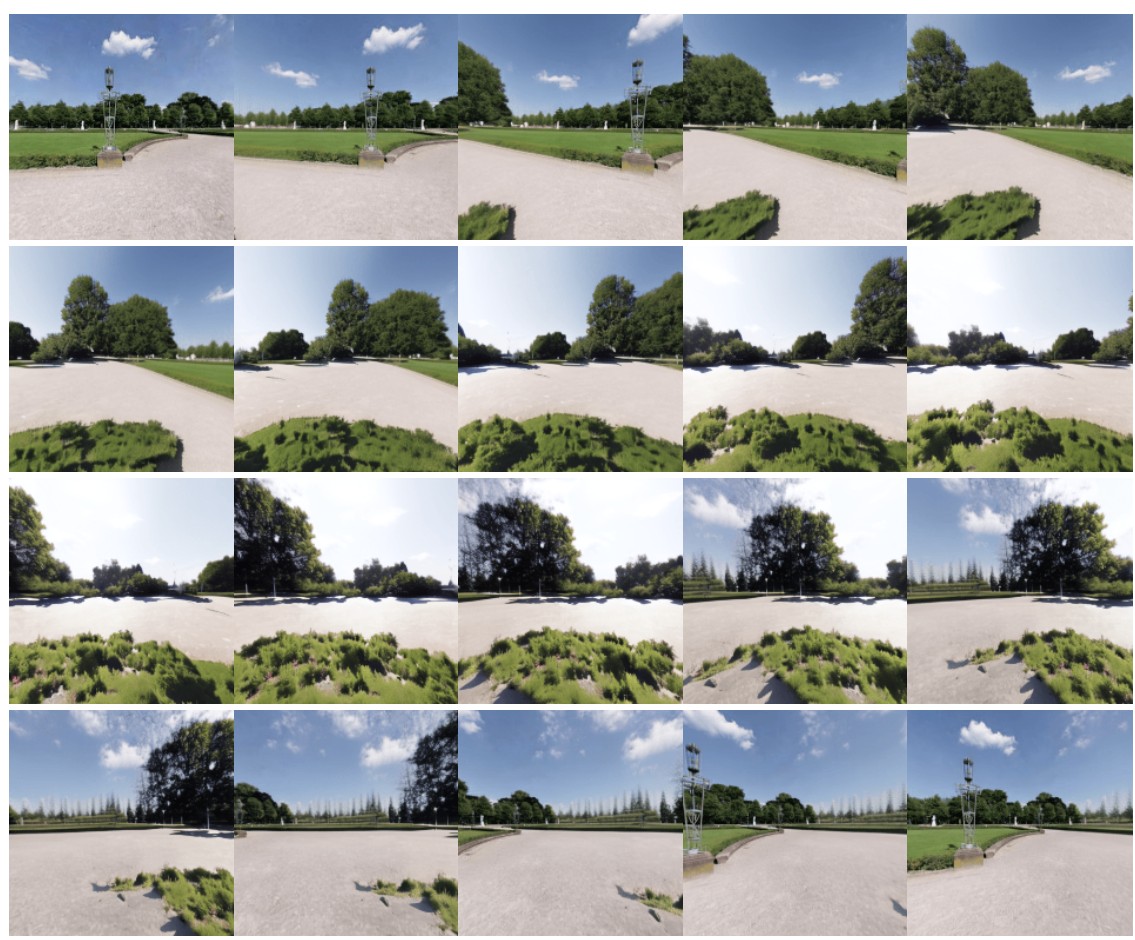

Figure 19: SUN360 Example #19 — Input Views, Stitch, and Upscale

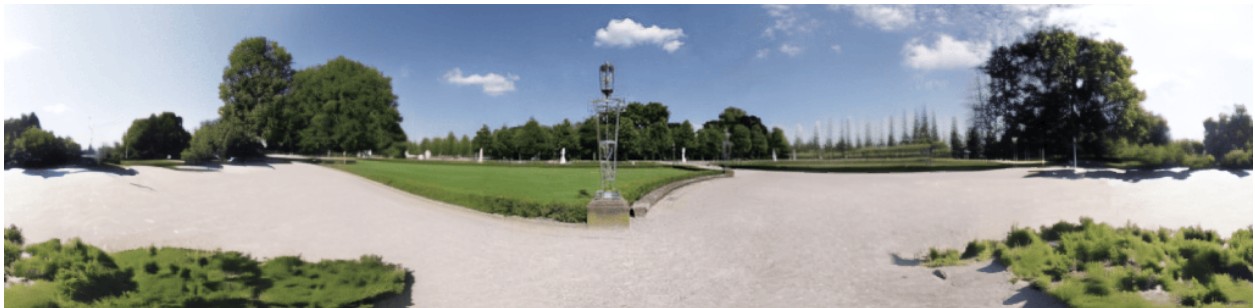

Stitched panorama

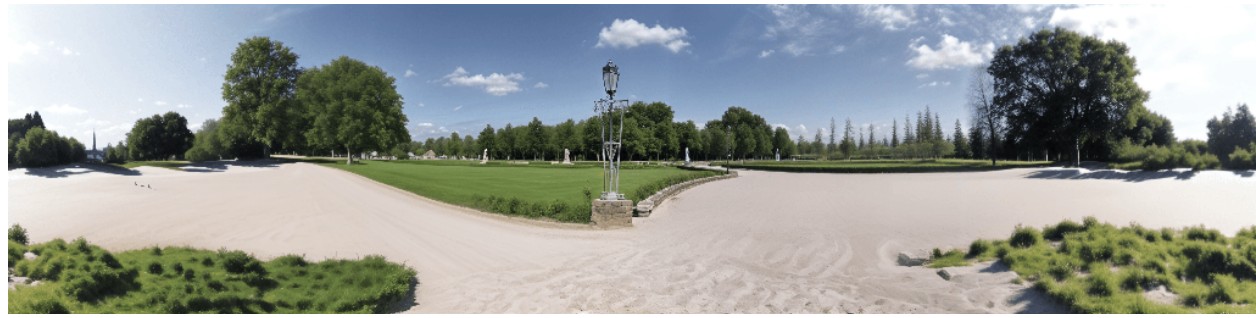

Upscaled final panorama

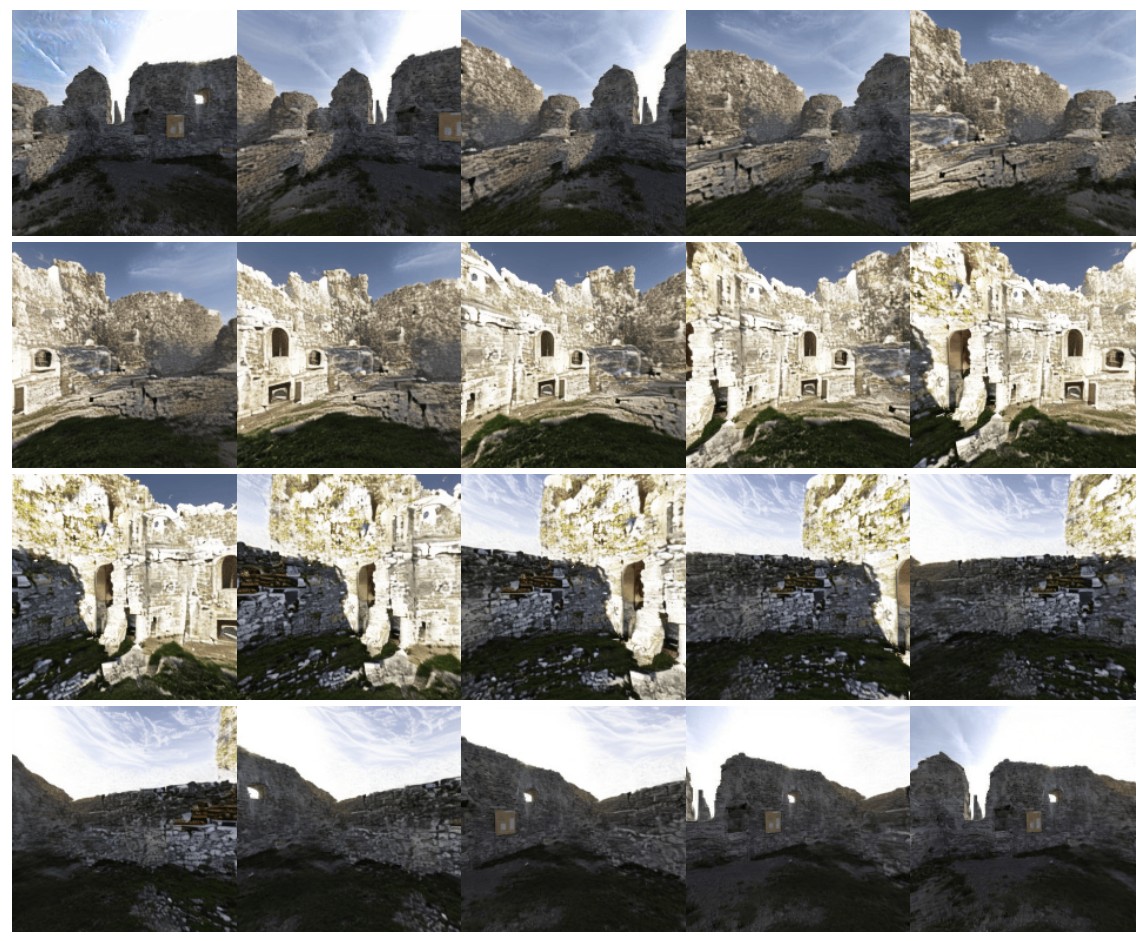

Figure 20: SUN360 Example #20 — Input Views, Stitch, and Upscale

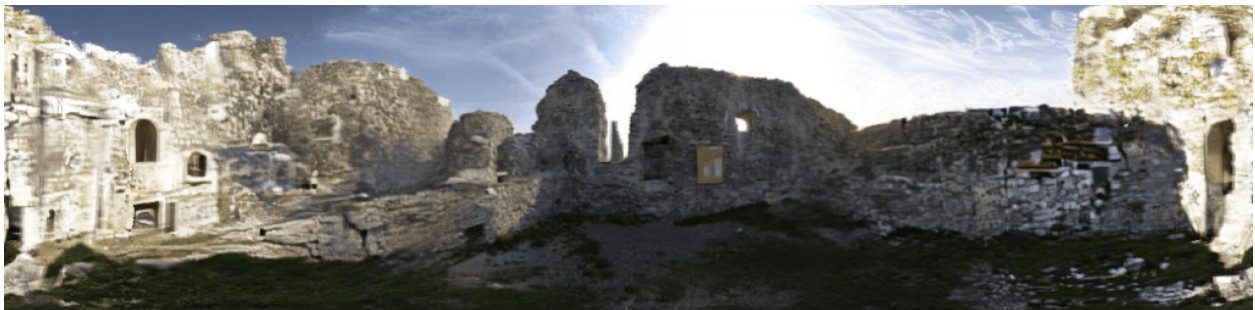

Stitched panorama

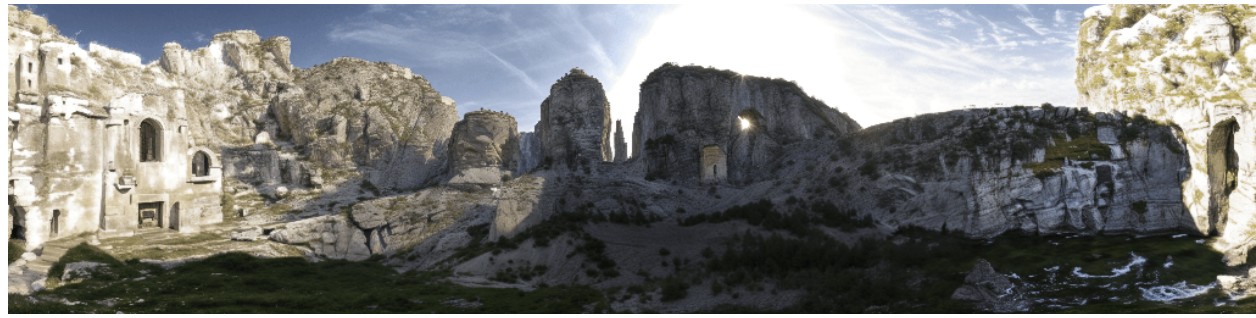

Upscaled final panorama

## 2.2 Laval Indoor Benchmark

# 3 Failure Cases: Extreme Pitch

SEVA struggles under extreme pitch rotations (full 360° up or down), producing distorted, inconsistent frames. We display the 20 raw SEVA-generated frames for each example.

## 3.1 SUN360 Failure Cases

### 3.1.1 Pitch-Up

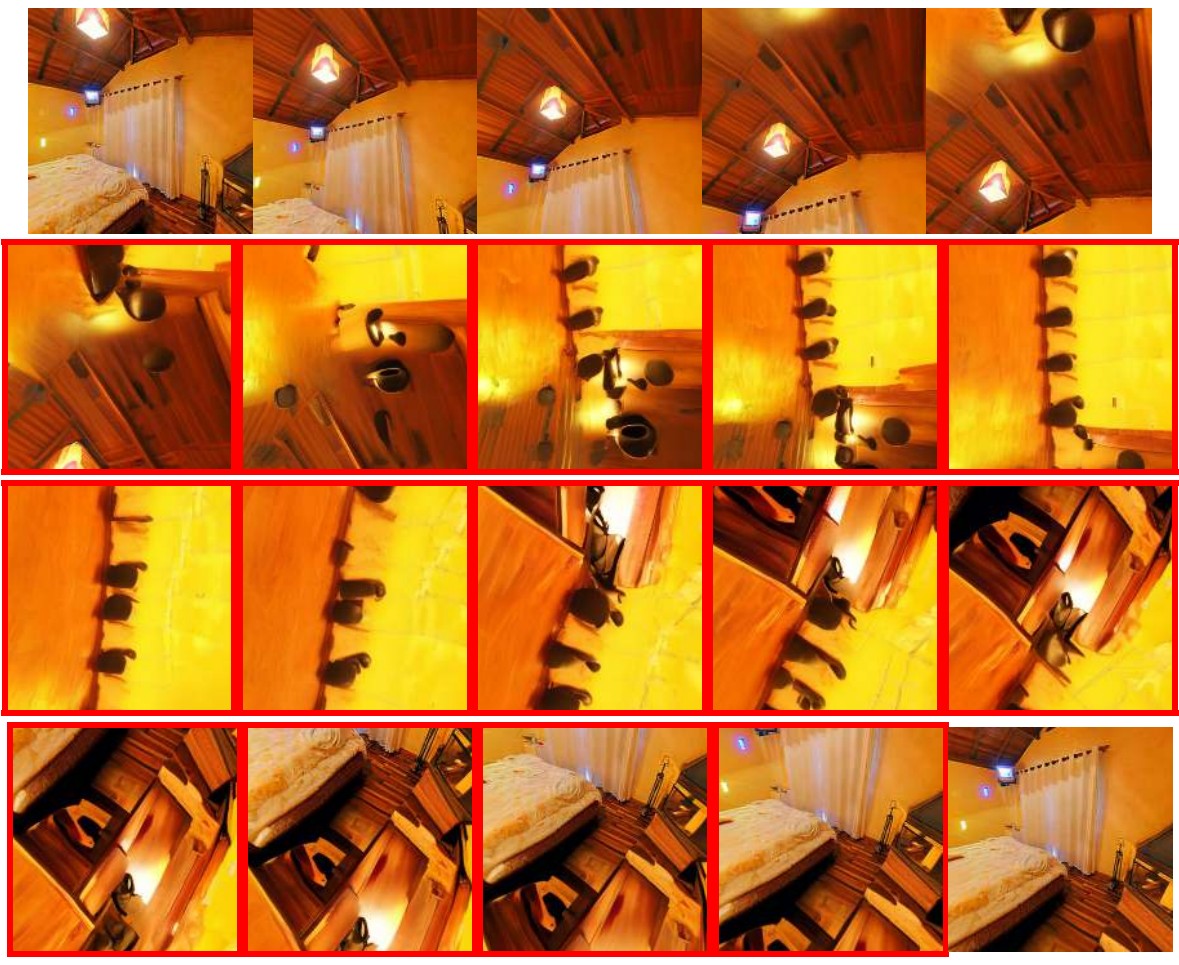

Figure 21: SUN360 Pitch-Up Case 1. Frames 5–18 (outlined in red) are inconsistent and distorted.

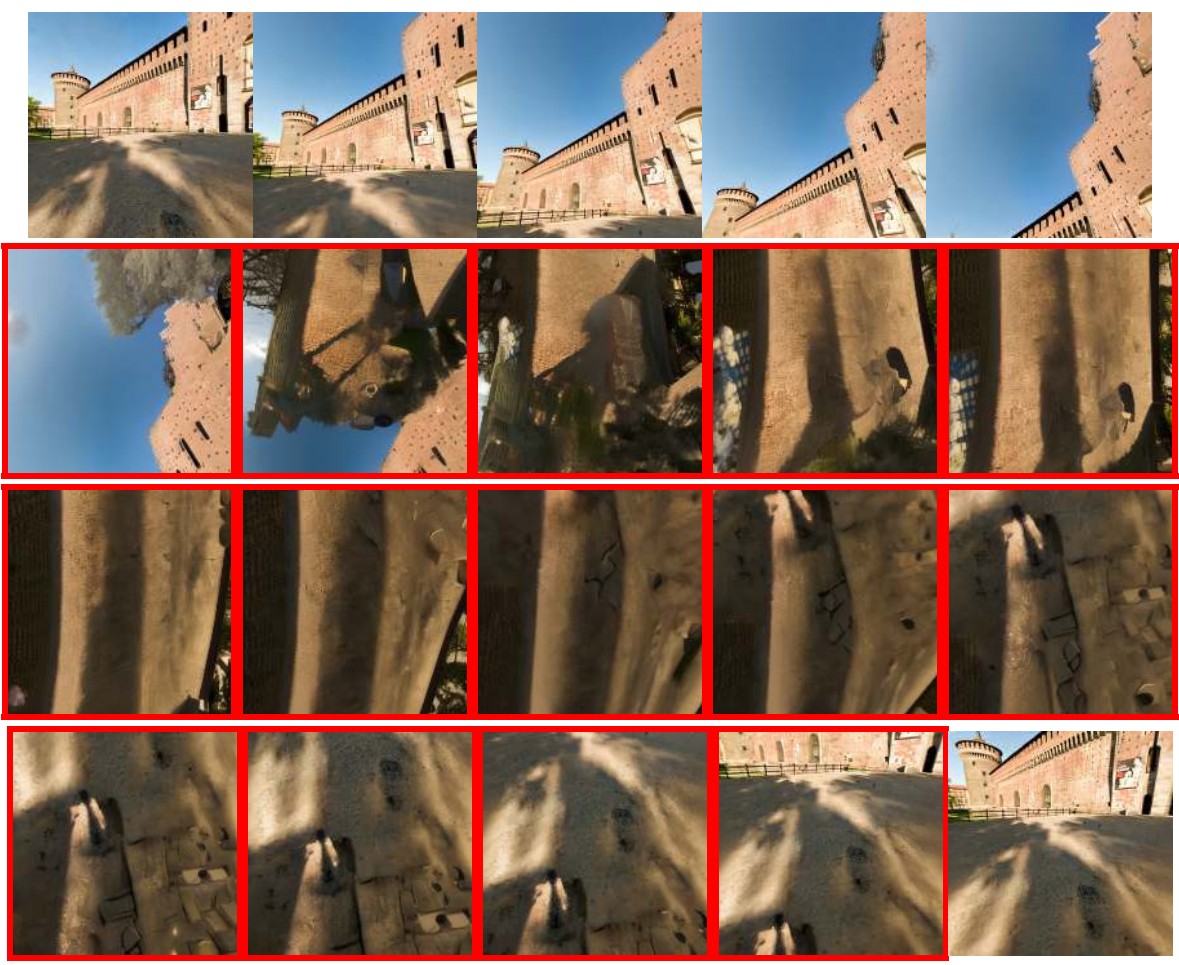

Figure 22: SUN360 Pitch-Up Case 2. Frames 5–18 (outlined in red) are inconsistent and distorted.

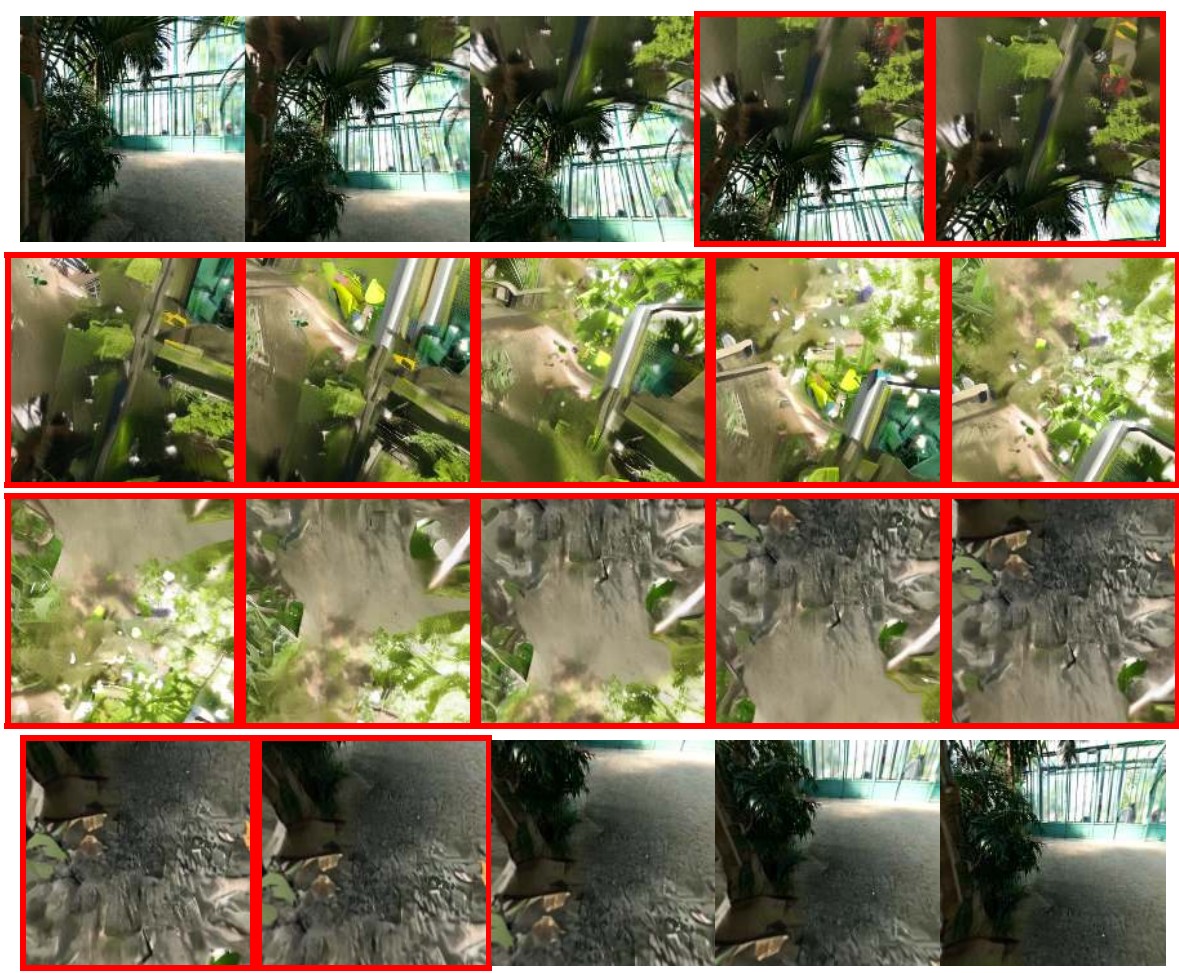

Figure 23: SUN360 Pitch-Up Case 3. Frames 3–16 (outlined in red) are inconsistent and distorted.

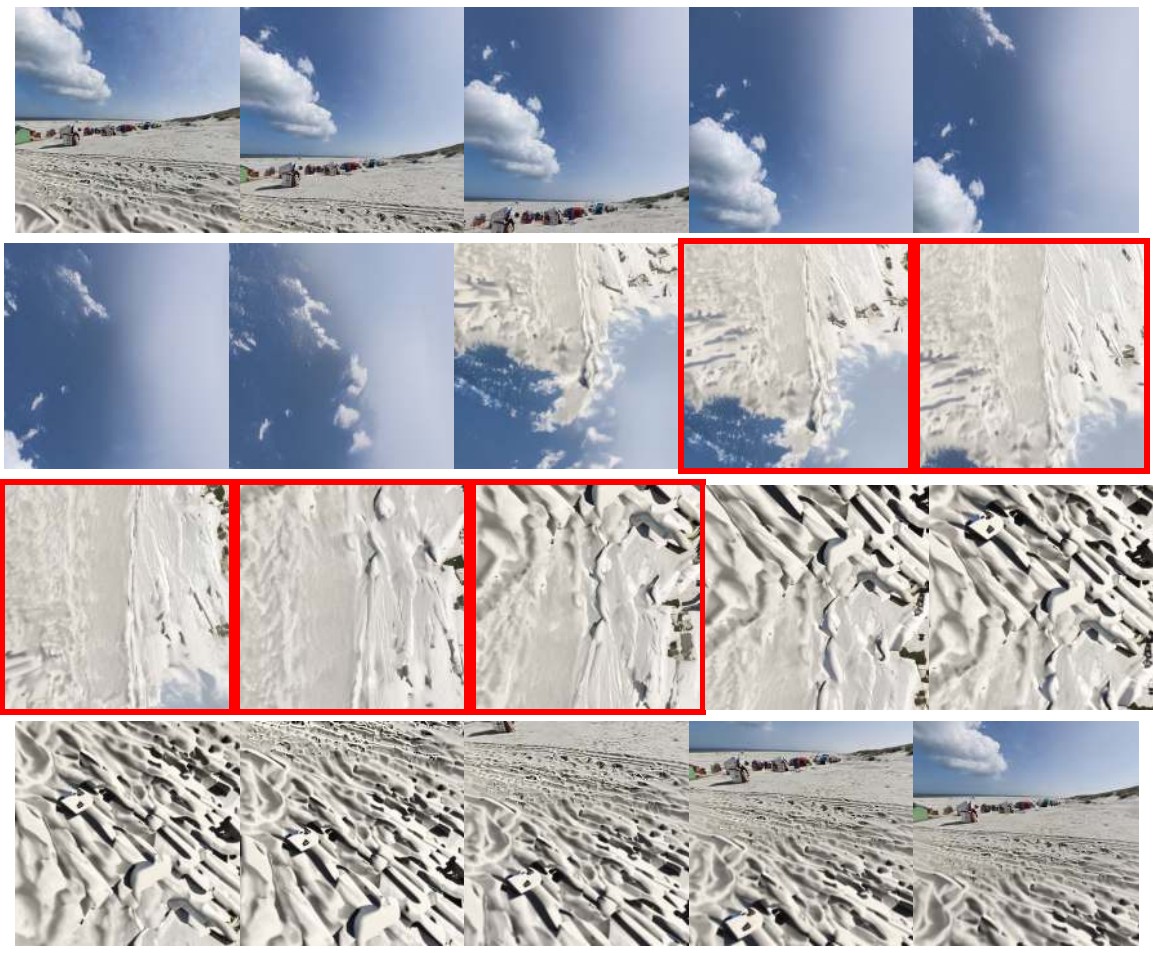

Figure 24: SUN360 Pitch-Up Case 4. Frames 8–12 (outlined in red) are inconsistent and distorted.

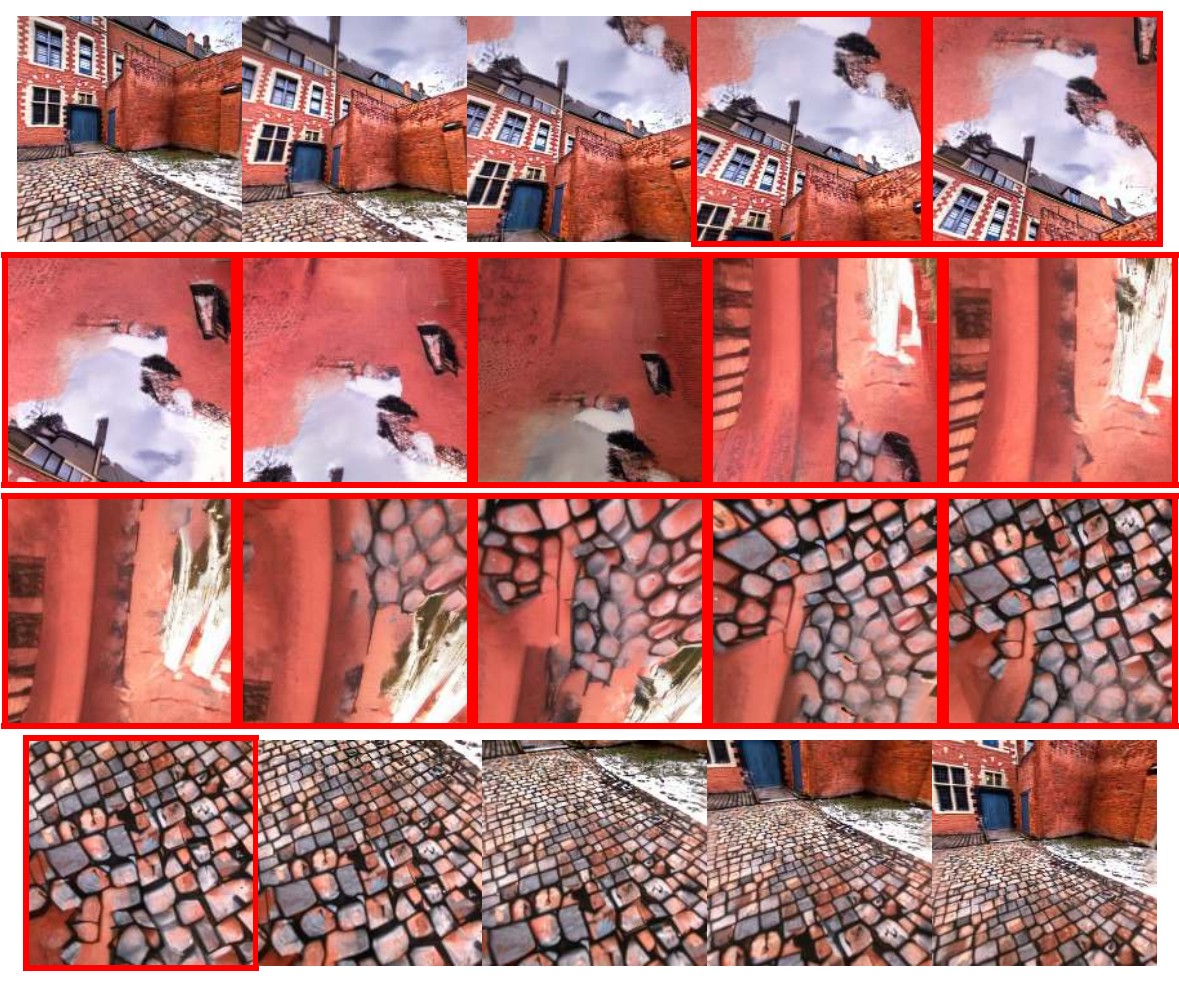

Figure 25: SUN360 Pitch-Up Case 5. Frames 3–15 (outlined in red) are inconsistent and distorted.

### 3.1.2   Pitch-Down

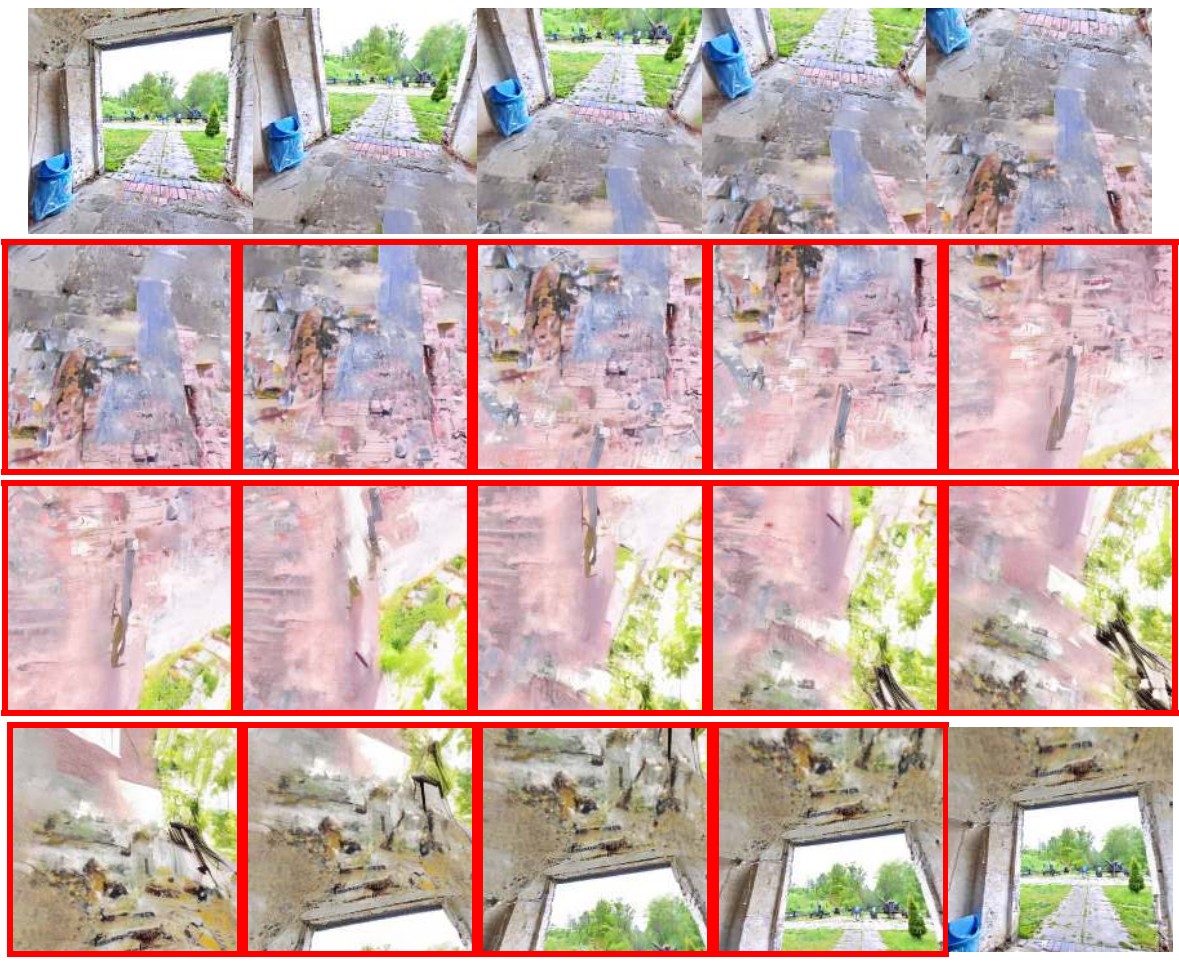

Figure 26: SUN360 Pitch-Down Case 1. Frames 5–18 (outlined in red) are inconsistent and distorted.

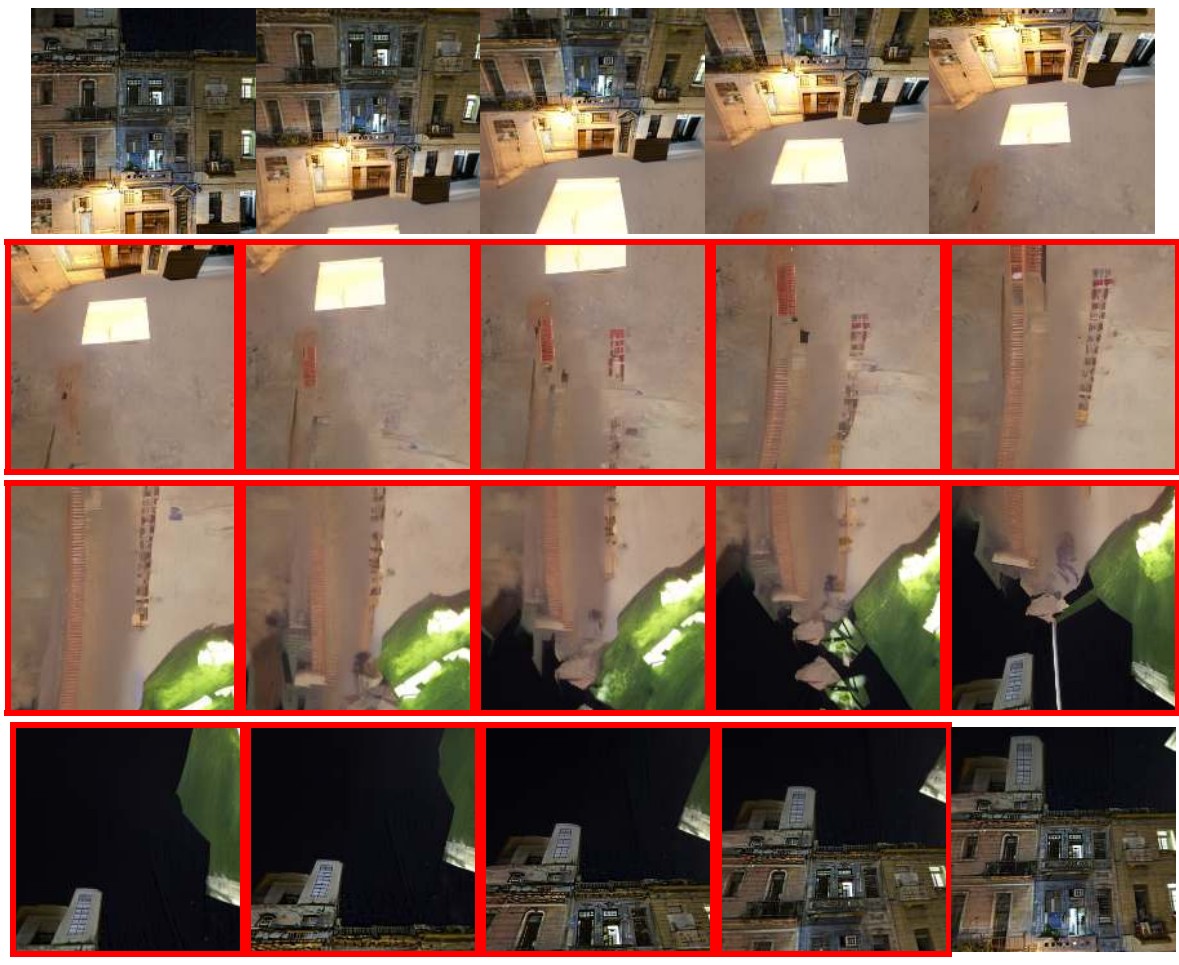

Figure 27: SUN360 Pitch-Down Case 2. Frames 5–18 (outlined in red) are inconsistent and distorted.

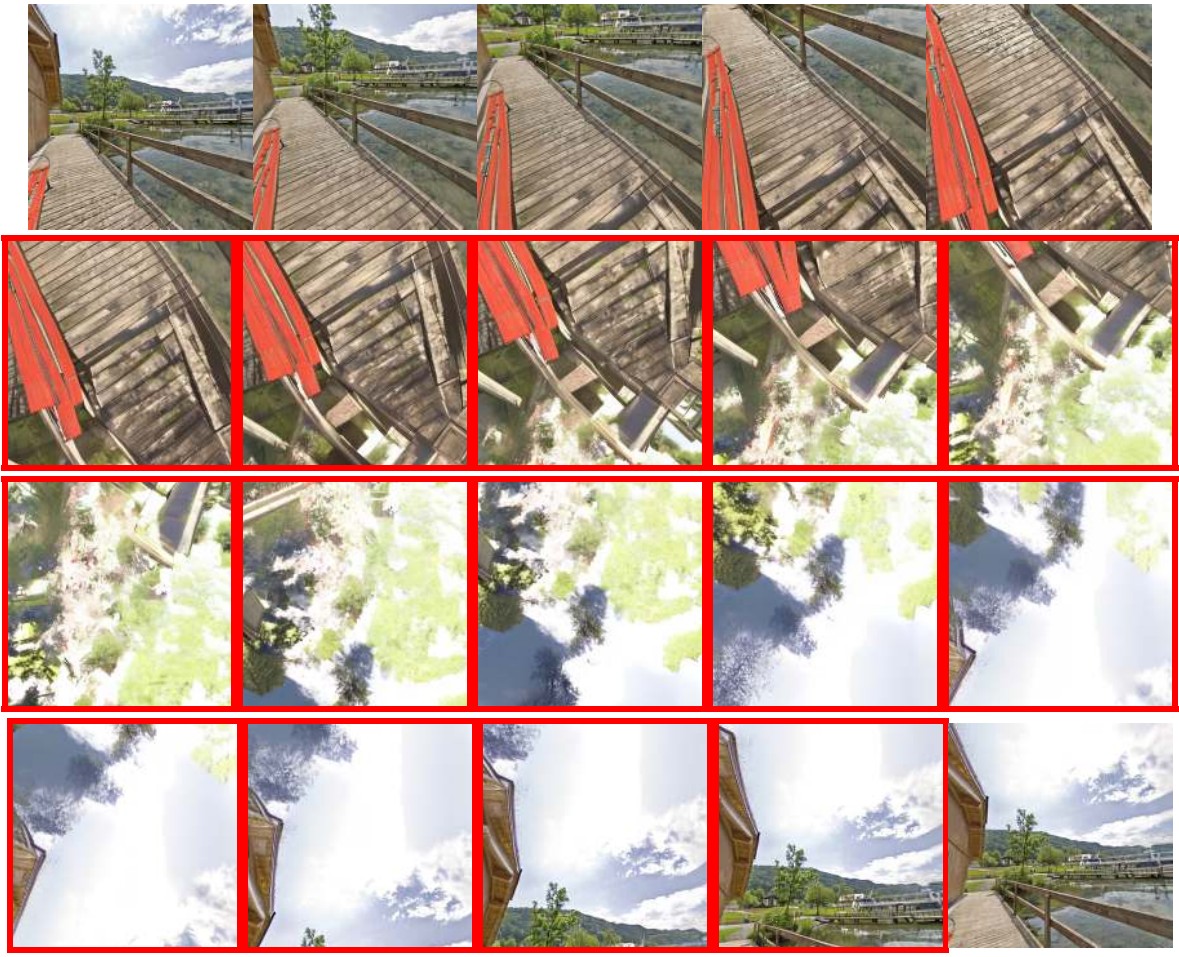

Figure 28: SUN360 Pitch-Down Case 3. Frames 5–18 (outlined in red) are inconsistent and distorted.

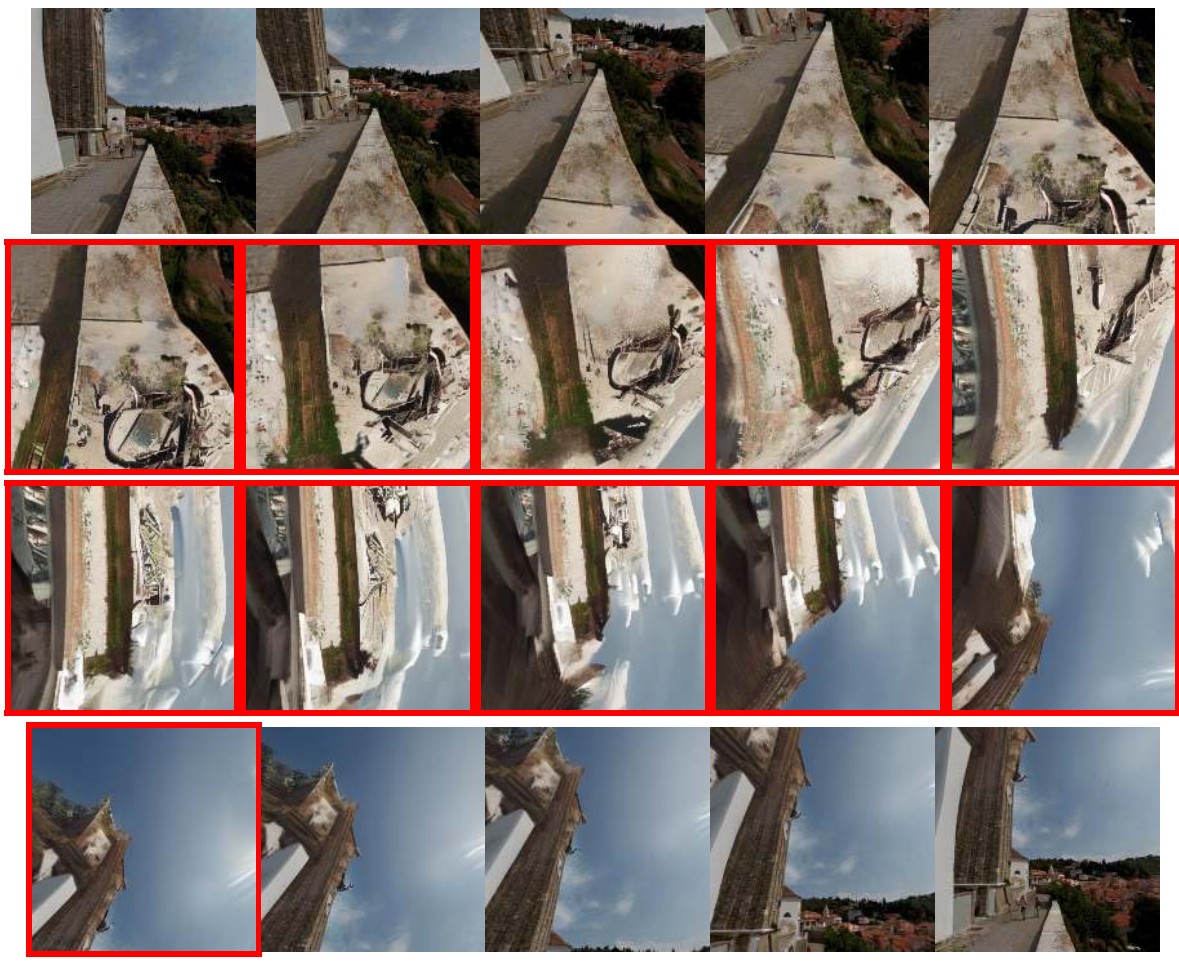

Figure 29: SUN360 Pitch-Down Case 4. Frames 5–15 (outlined in red) are inconsistent and distorted.

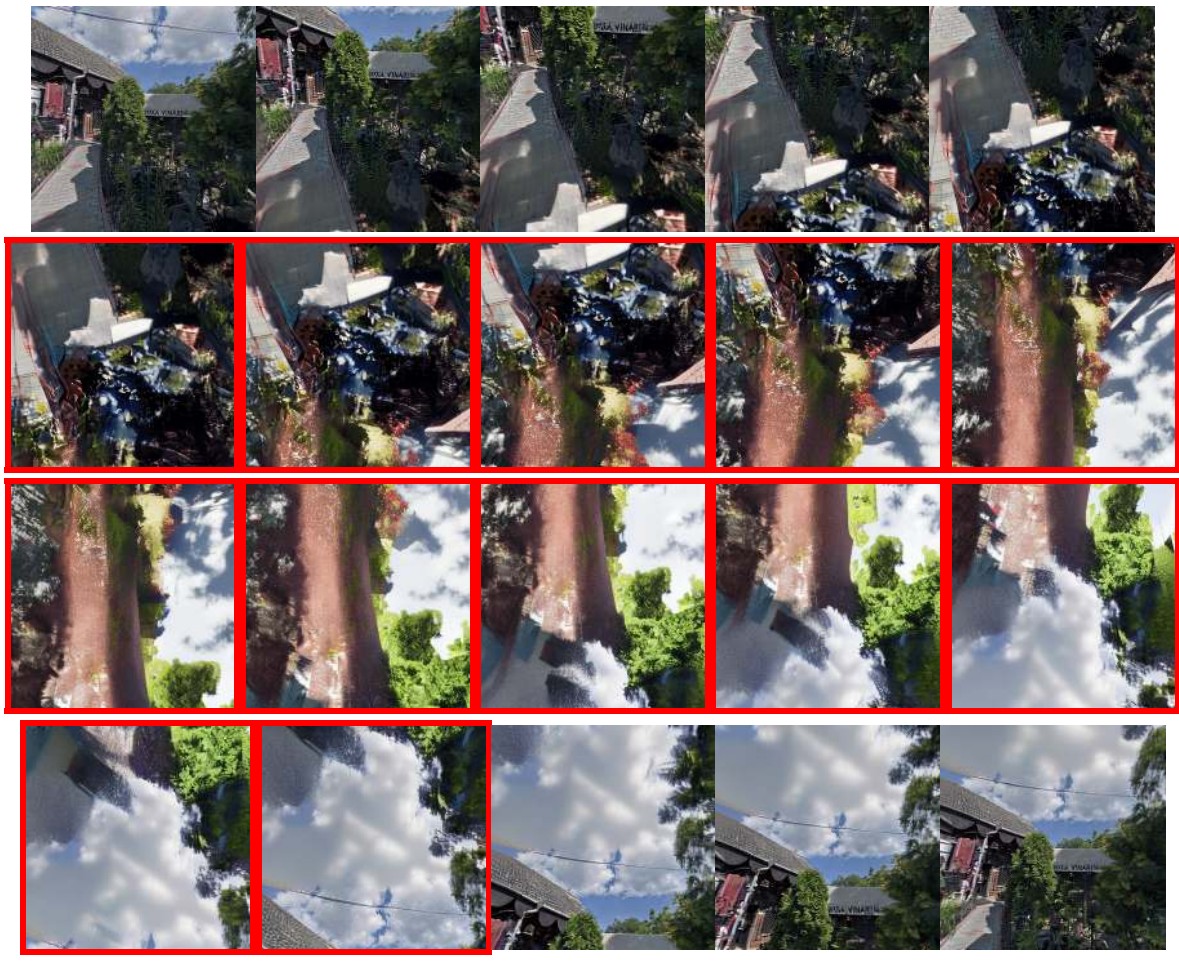

Figure 30: SUN360 Pitch-Down Case 5. Frames 5–16 (outlined in red) are inconsistent and distorted.

## 3.2 Laval Indoor Failure Cases

### 3.2.1 Pitch-Up

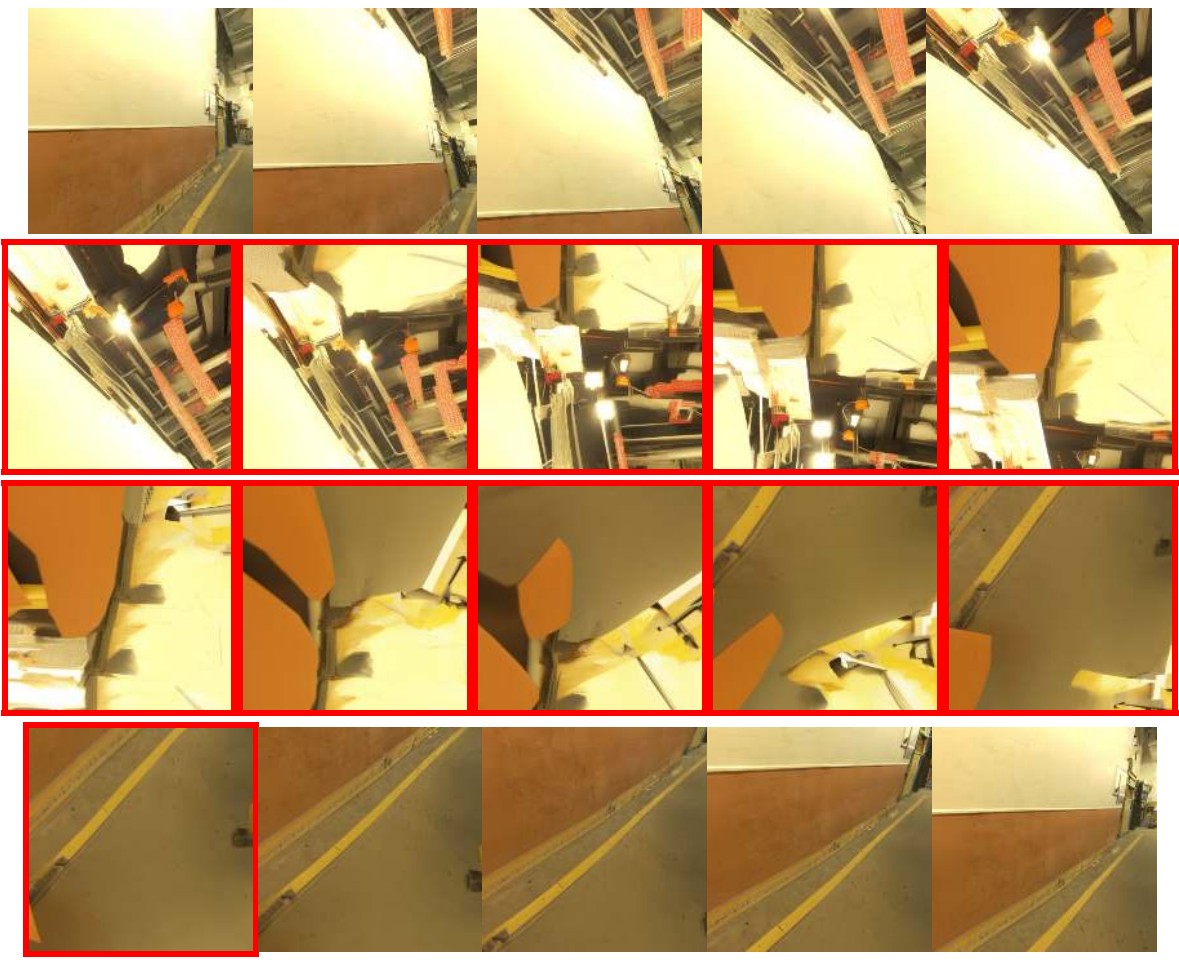

Figure 31: Laval Indoor Pitch-Up Case 1. Frames 5–15 (outlined in red) are inconsistent and distorted.

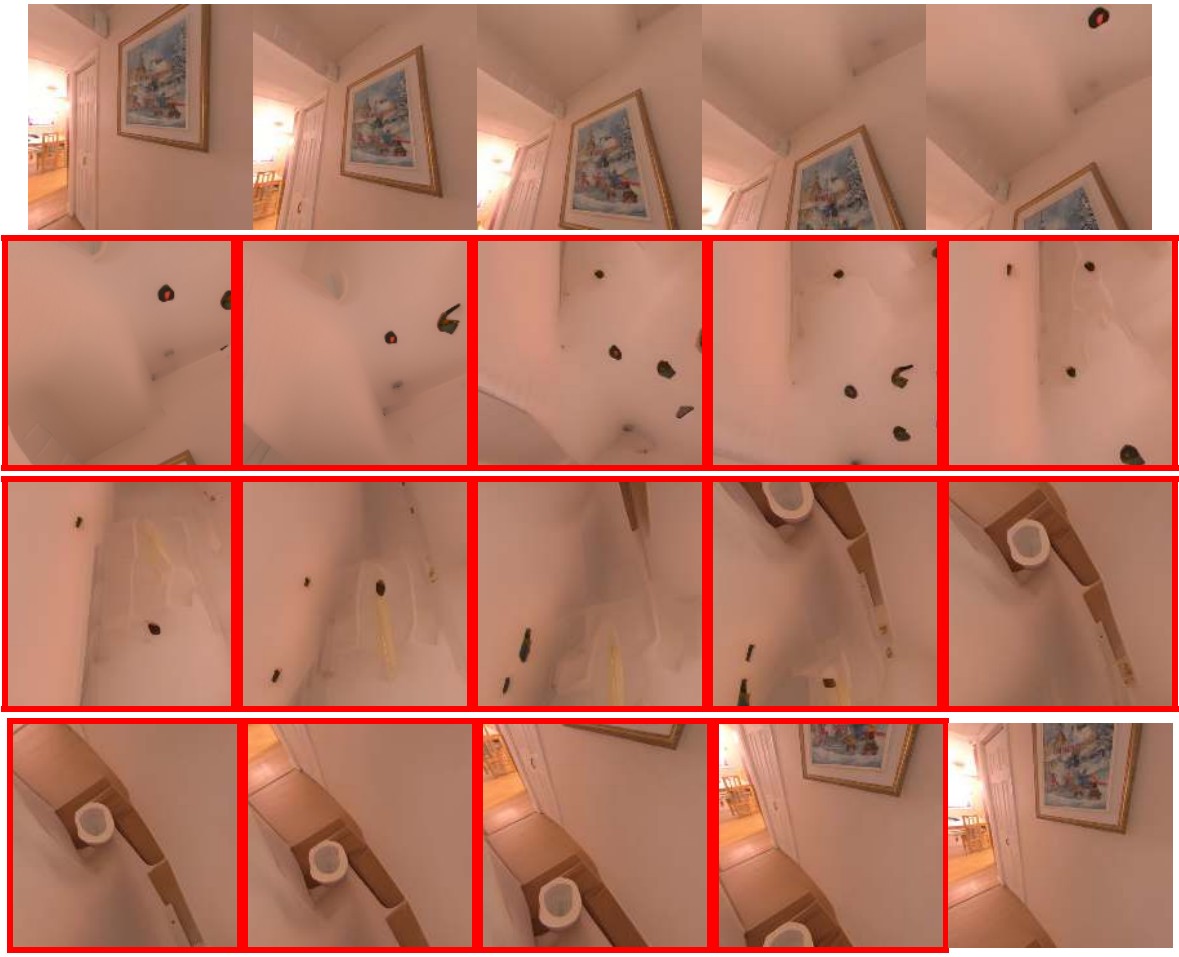

Figure 32: Laval Indoor Pitch-Up Case 2. Frames 5–18 (outlined in red) are inconsistent and distorted.

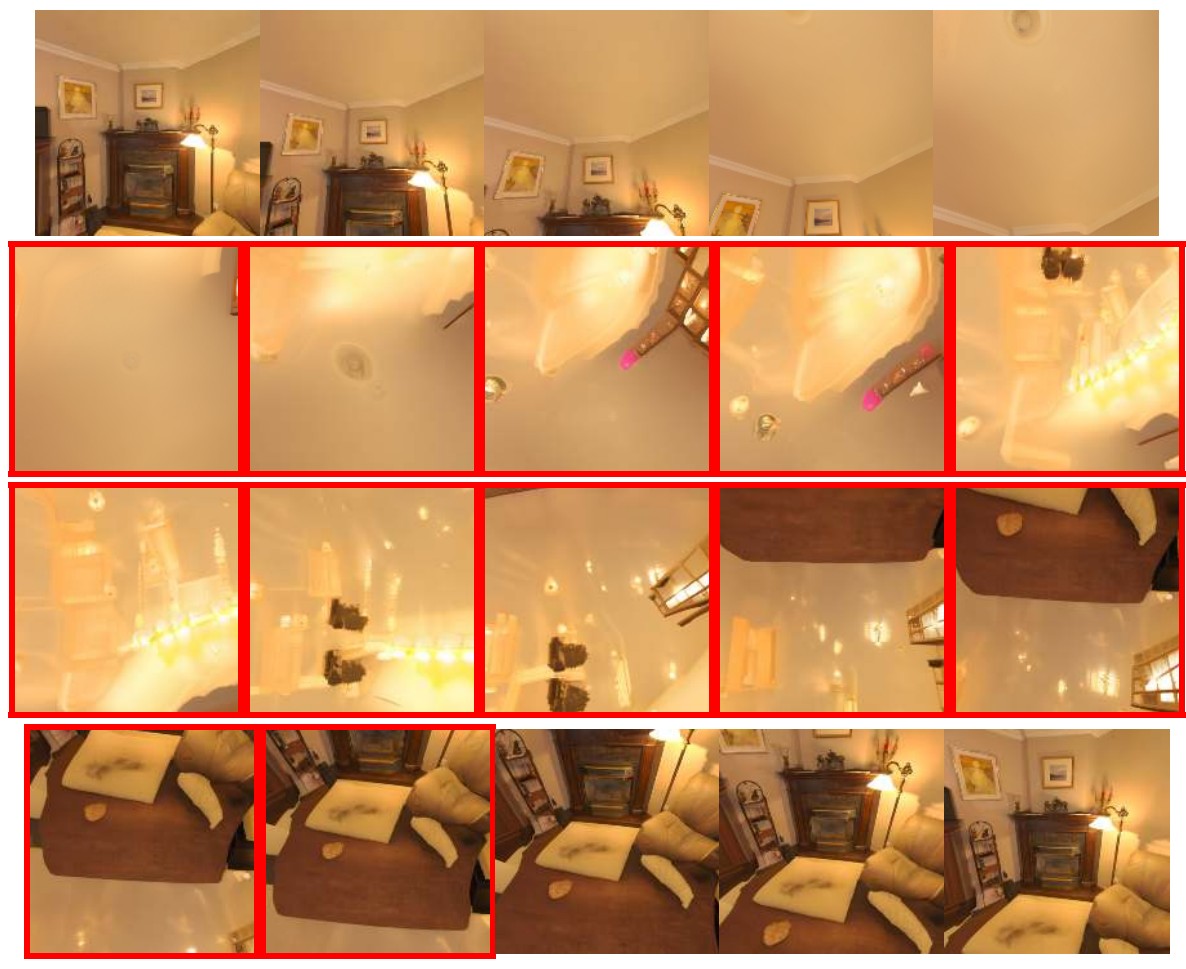

Figure 33: Laval Indoor Pitch-Up Case 3. Frames 5–16 (outlined in red) are inconsistent and distorted.

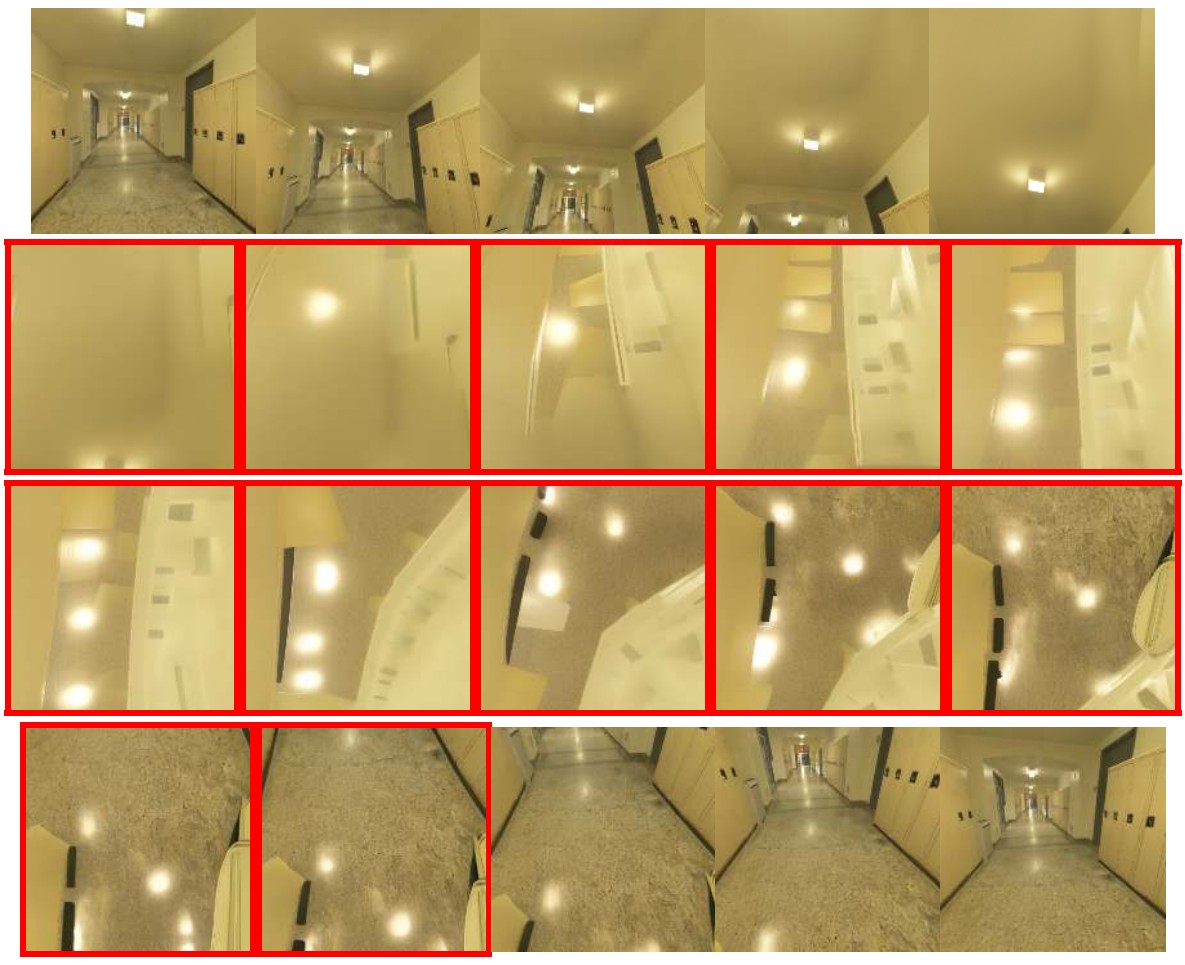

Figure 34: Laval Indoor Pitch-Up Case 4. Frames 5–16 (outlined in red) are inconsistent and distorted.

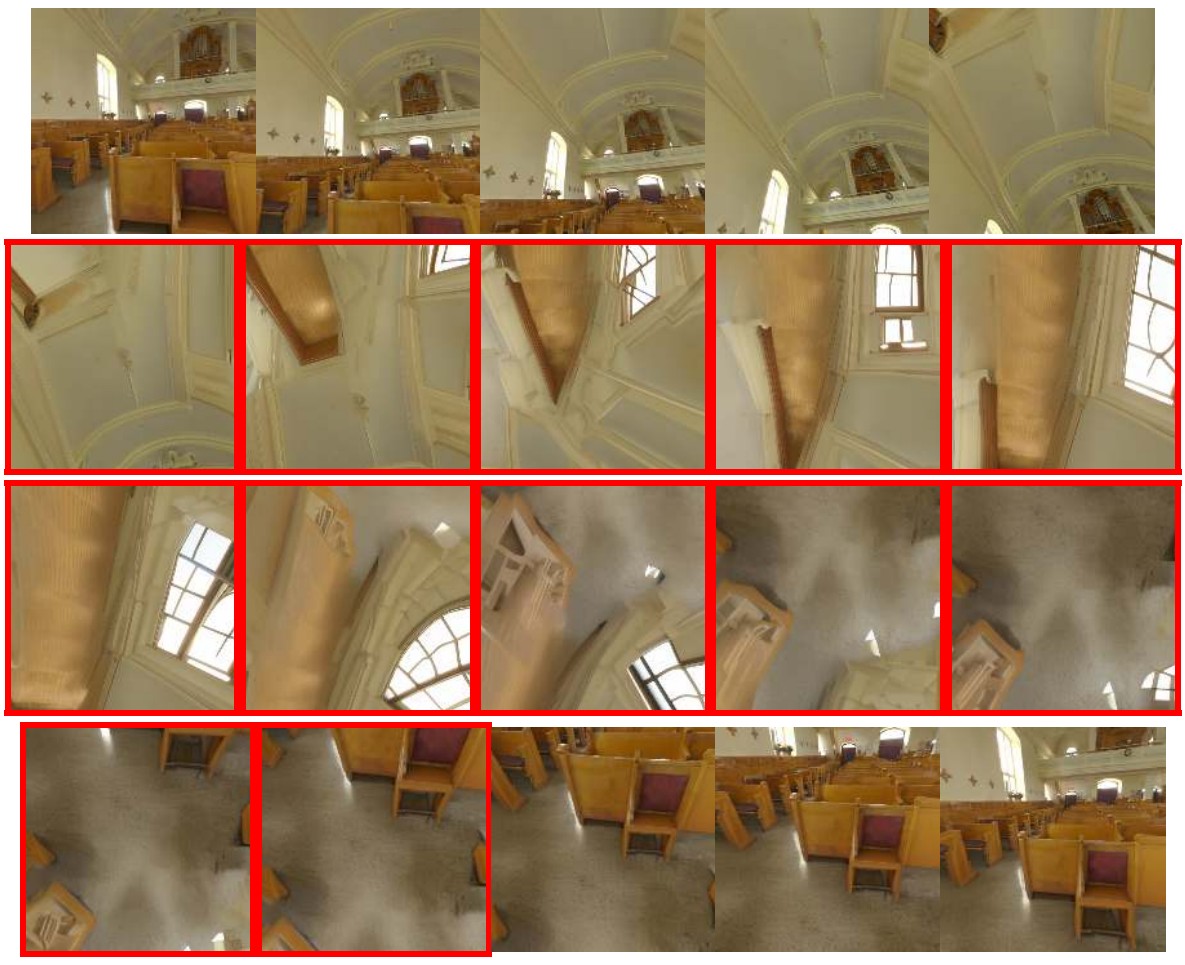

Figure 35: Laval Indoor Pitch-Up Case 5. Frames 5–16 (outlined in red) are inconsistent and distorted.

### 3.2.2 Pitch-Down

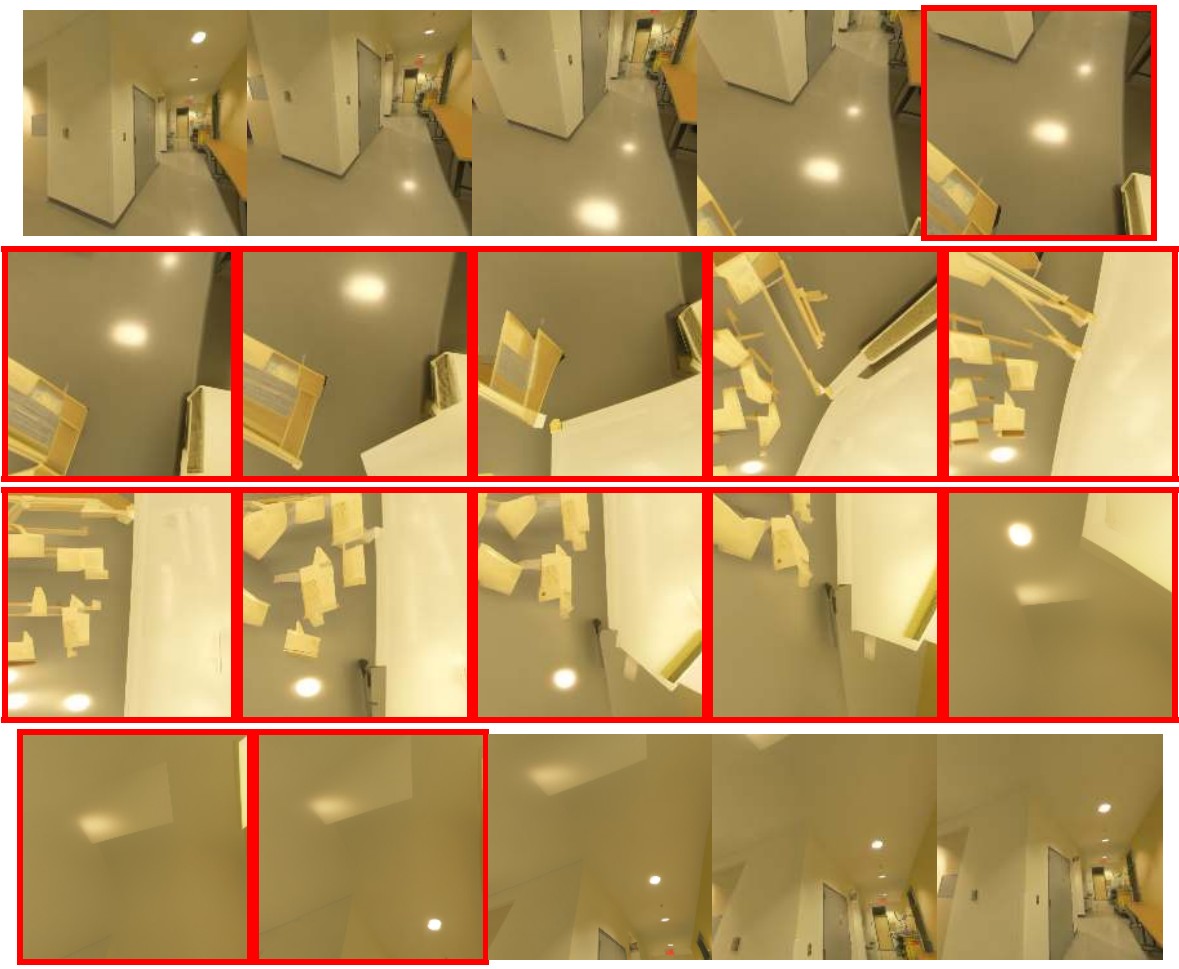

Figure 36: Laval Indoor Pitch-Down Case 1. Frames 4–16 (outlined in red) are inconsistent and distorted.

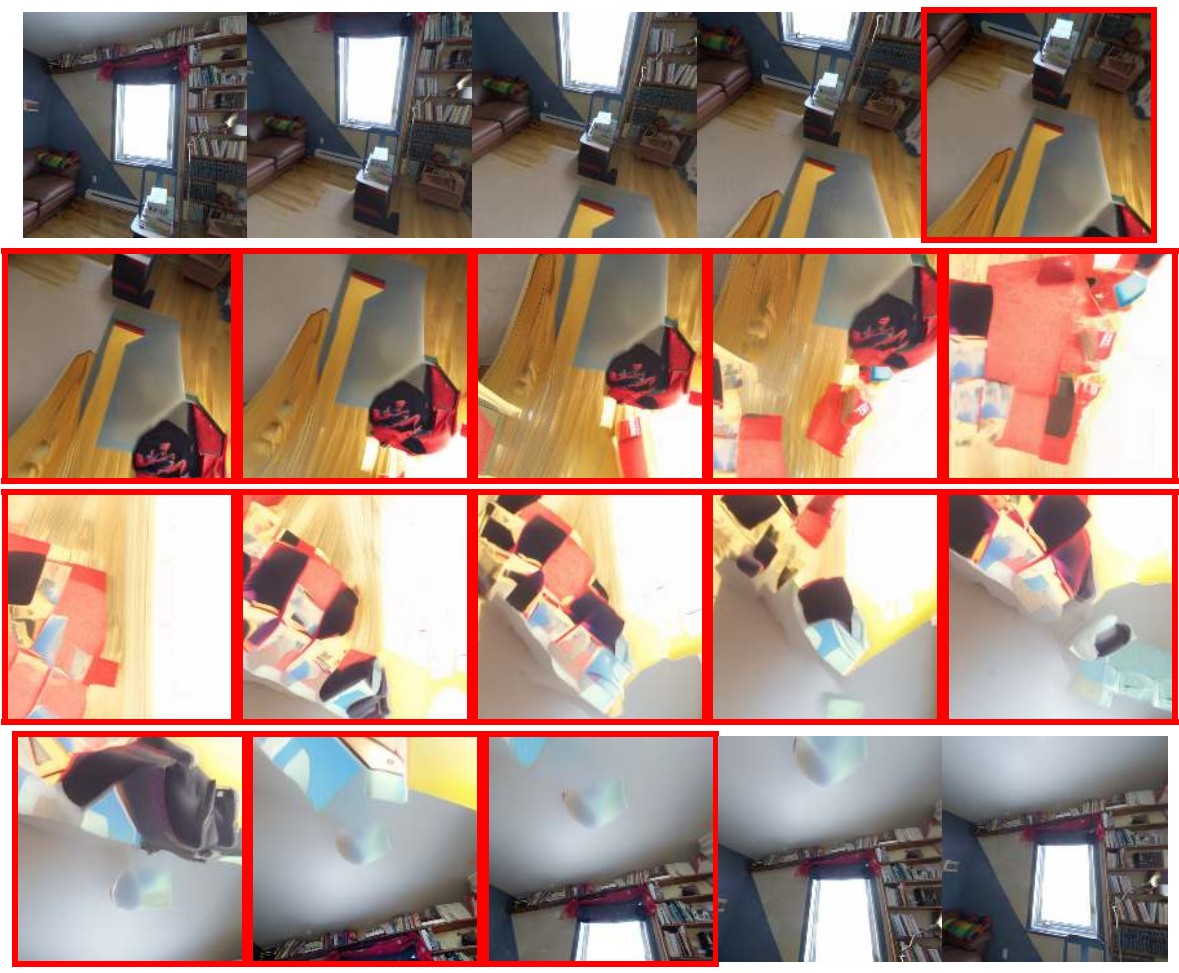

Figure 37: Laval Indoor Pitch-Down Case 2. Frames 4–17 (outlined in red) are inconsistent and distorted.

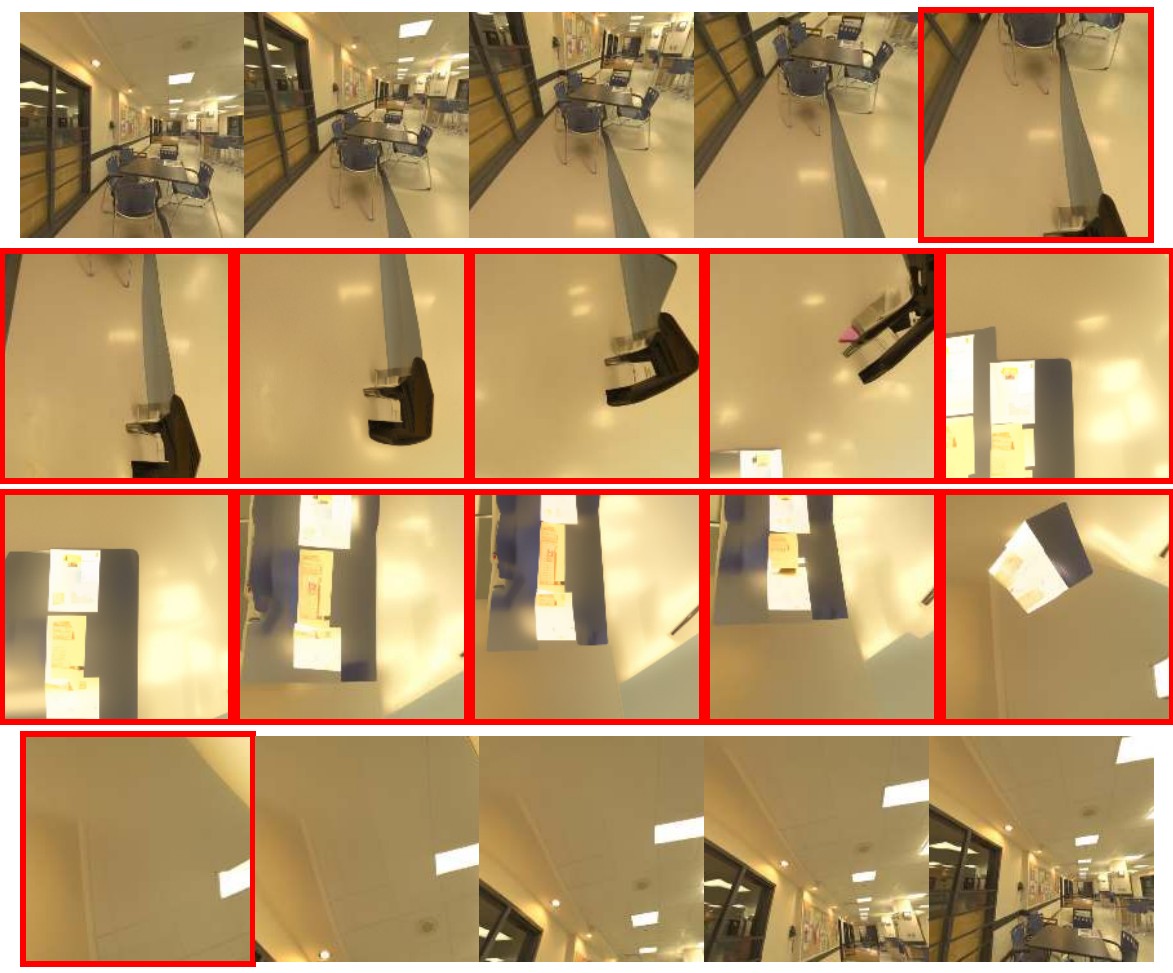

Figure 38: Laval Indoor Pitch-Down Case 3. Frames 4–15 (outlined in red) are inconsistent and distorted.

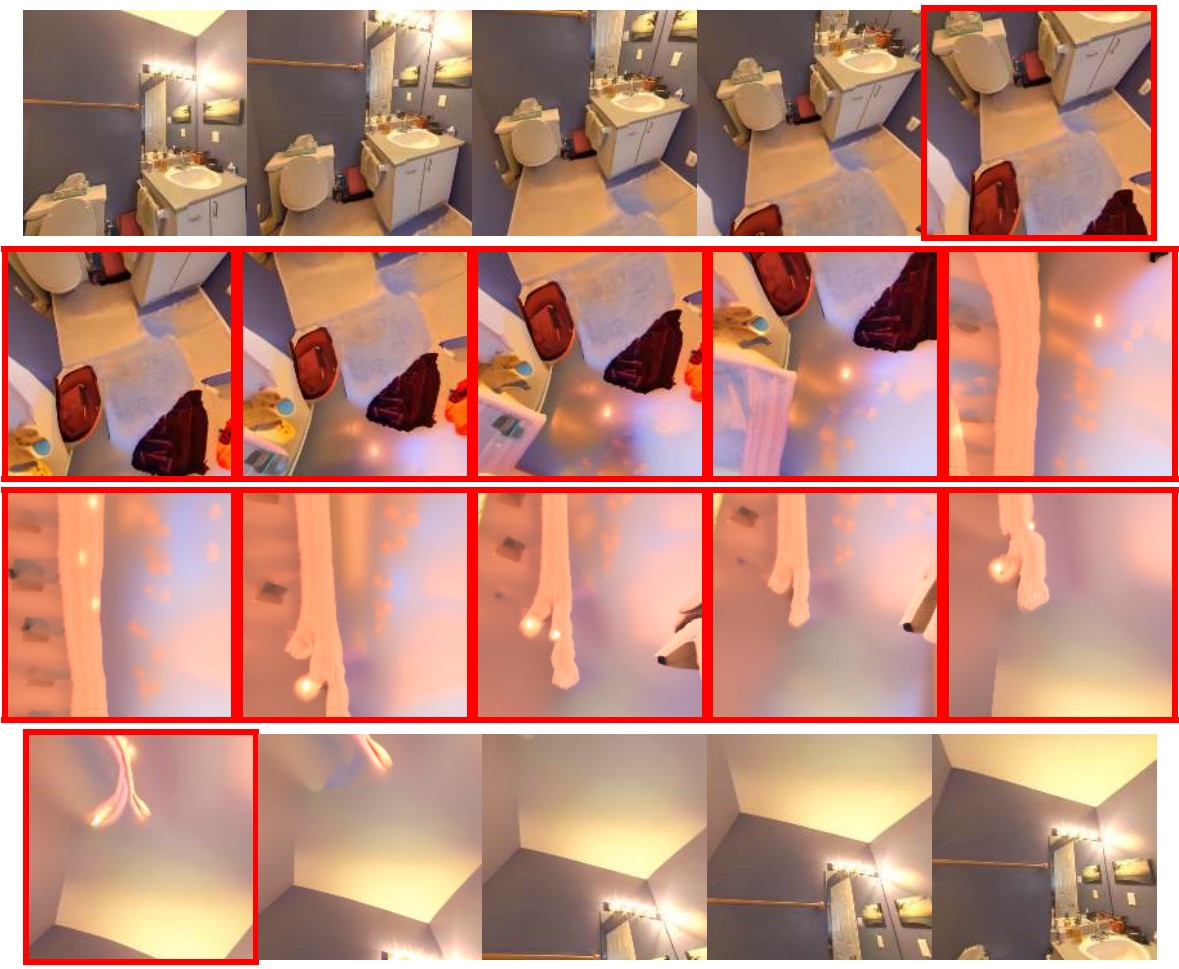

Figure 39: Laval Indoor Pitch-Down Case 4. Frames 4–15 (outlined in red) are inconsistent and distorted.

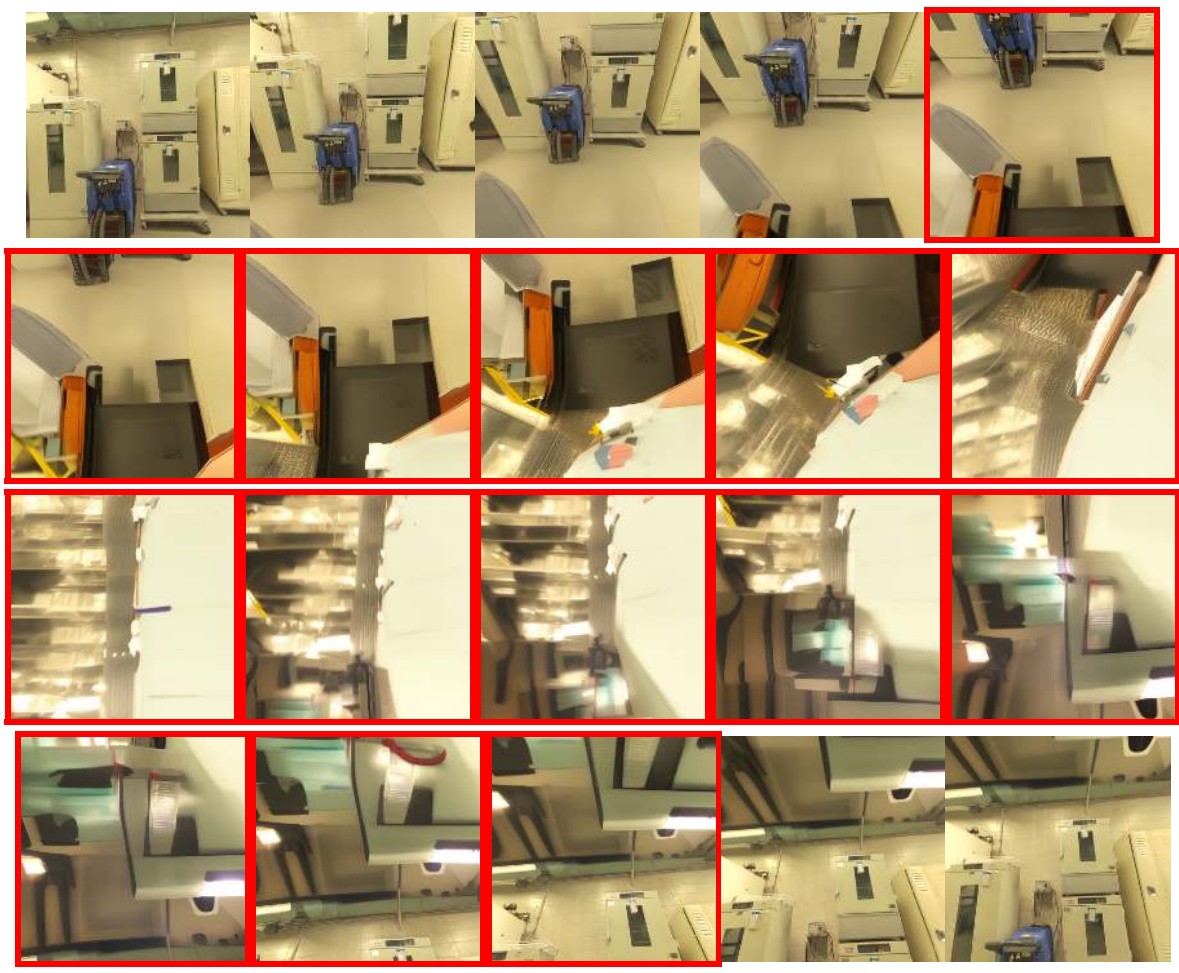

Figure 40: Laval Indoor Pitch-Down Case 5. Frames 4–17 (outlined in red) are inconsistent and distorted.

# 4  Stitching & Enhancement Analysis

We report two complementary observations relevant to understanding the full behavior of our pipeline. Please note that all examples in this section originate from crops within the Laval Indoor or SUN360 benchmarks.

- **Type A: Stitching Artifacts Resolved.** SEVA may generate frames that do not align perfectly, introducing ghosting or seams. The enhancement module effectively repairs these.

- **Type B: Enhancement-Induced Visuals.** In rare cases, the enhancement module may hallucinate human-like shapes or amplify incorrect textures.

## 4.1  Type A: Stitching Artifacts Resolved by Enhancement

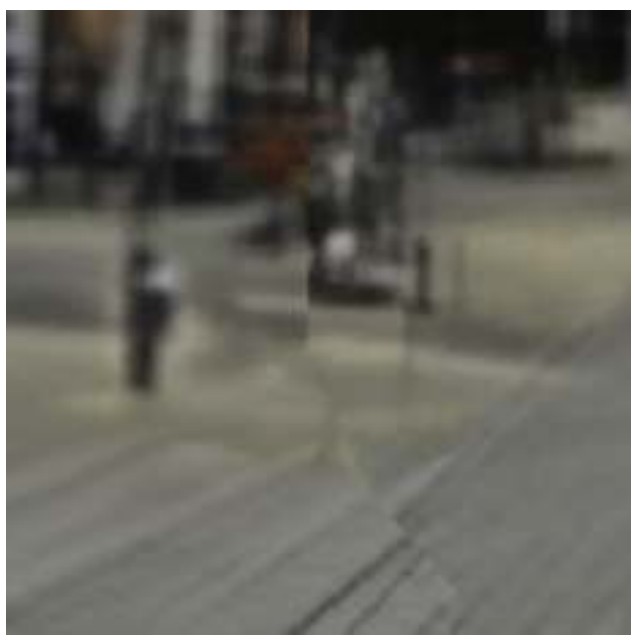

Before enhancement (stitched panorama)

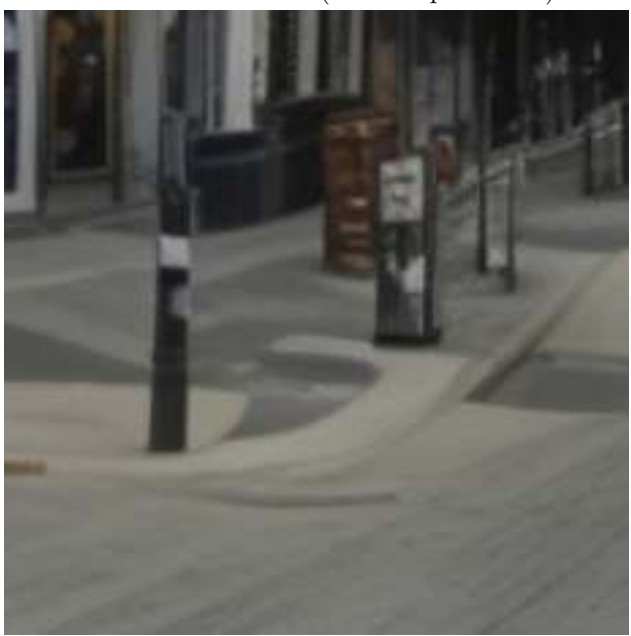

After enhancement module

Figure 41: Stitching Artifact Case 1 — The enhancement module removes the seam line.

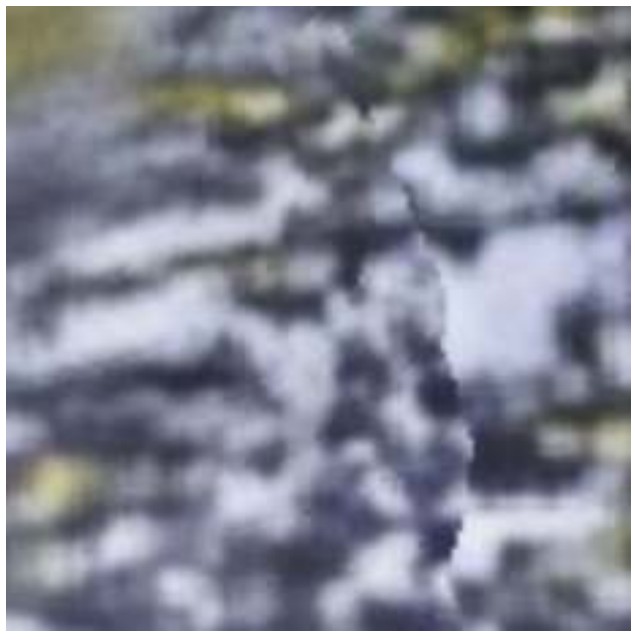

Before enhancement (stitched panorama)

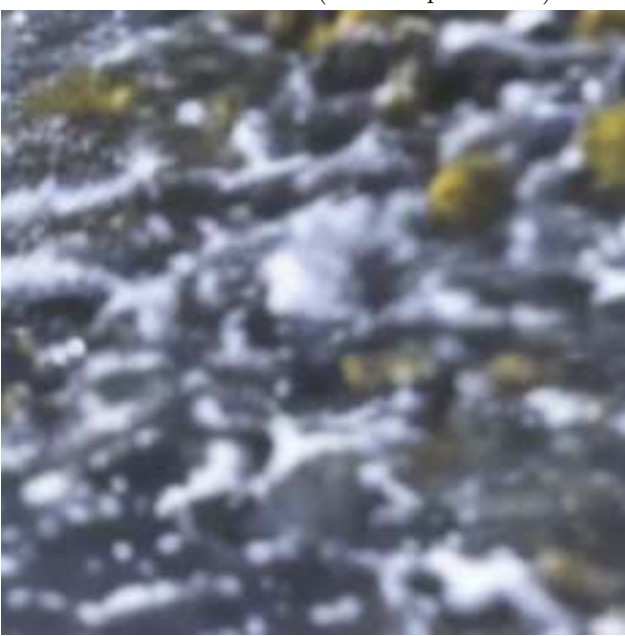

After enhancement module

Figure 42: Stitching Artifact Case 2 — The enhancement module removes the seam line.

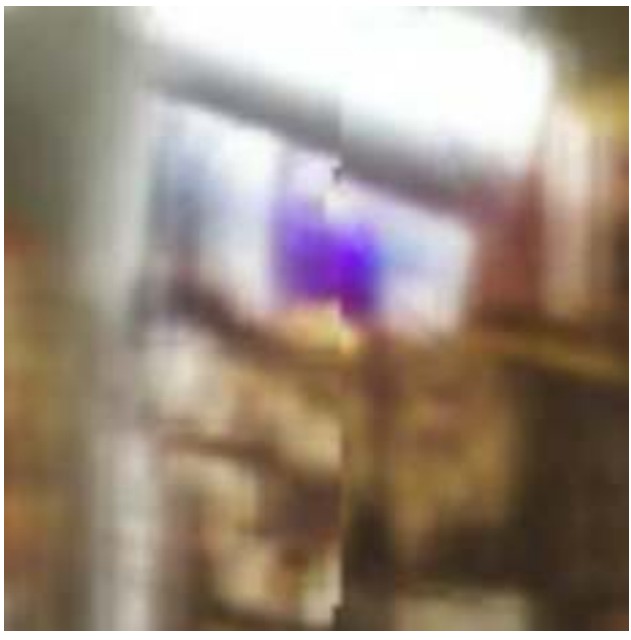

Before enhancement (stitched panorama)

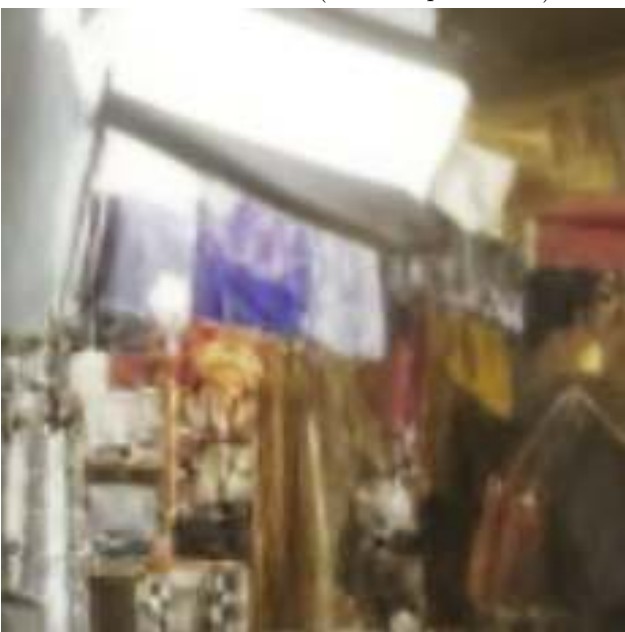

After enhancement module

Figure 43: Stitching Artifact Case 3 — The enhancement module removes the seam line.

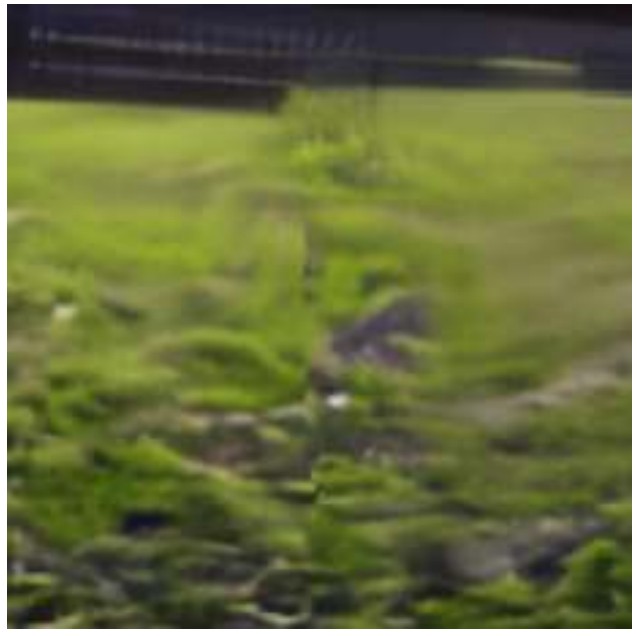

Before enhancement (stitched panorama)

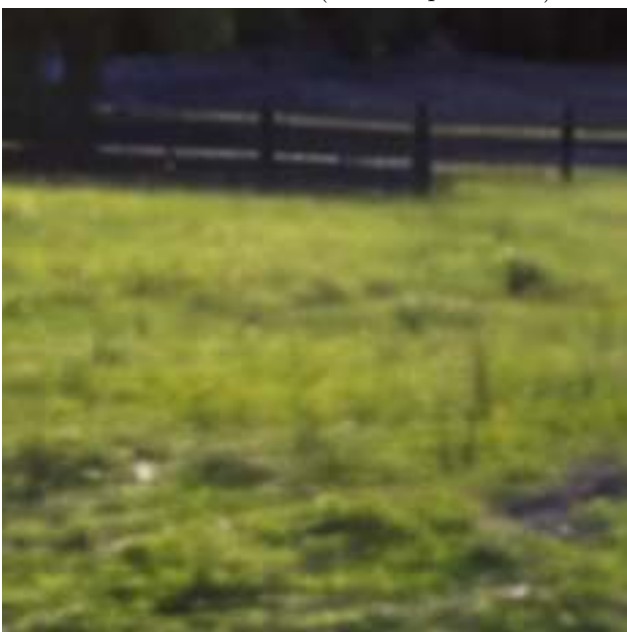

After enhancement module

Figure 44: Stitching Artifact Case 4 — The enhancement module removes the seam line.

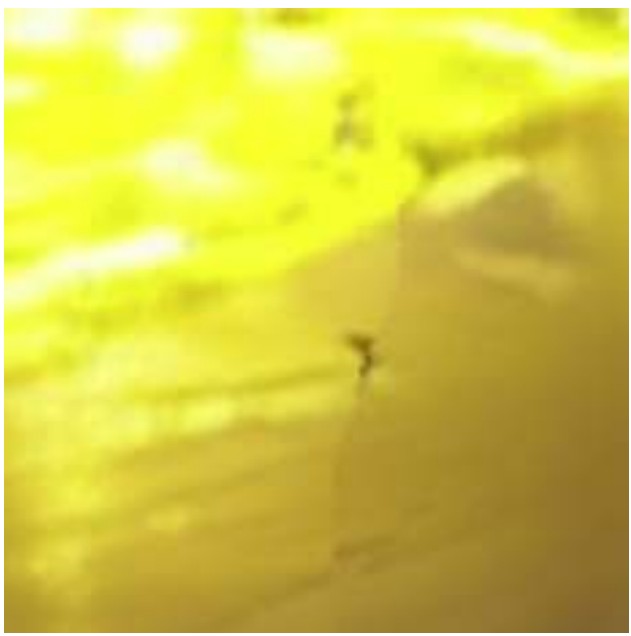

Before enhancement (stitched panorama)

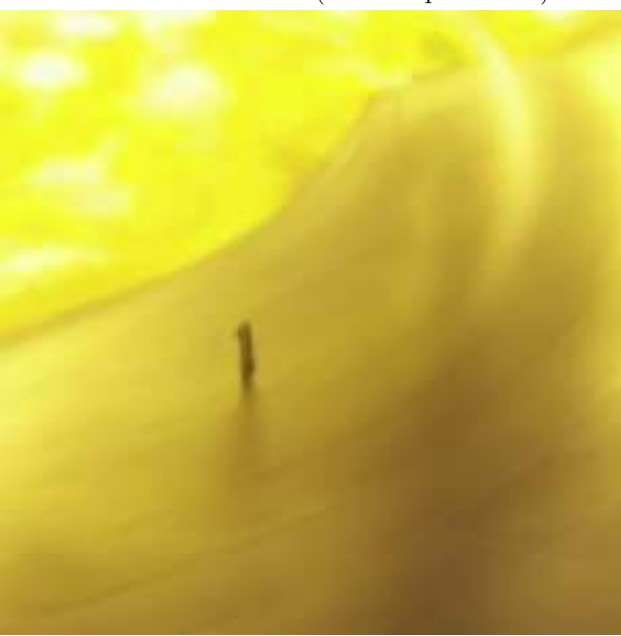

After enhancement module

Figure 45: Stitching Artifact Case 5 — The enhancement module removes the seam line.

## 4.2 Type B: Enhancement-Induced Unwanted Visuals

Unwanted artifacts can appear as a side-effect of the enhancing and upscaling module. All examples below are crops taken from full panoramas generated on the SUN360 and Laval Indoor benchmarks. The primary issue arises when humans or animals are present in the input images; while the module attempts to refine these subjects, it can occasionally produce distorted or unnatural representations. Furthermore, in rare instances, the upscaling stage may incorrectly transform non-human objects into human-like figures.

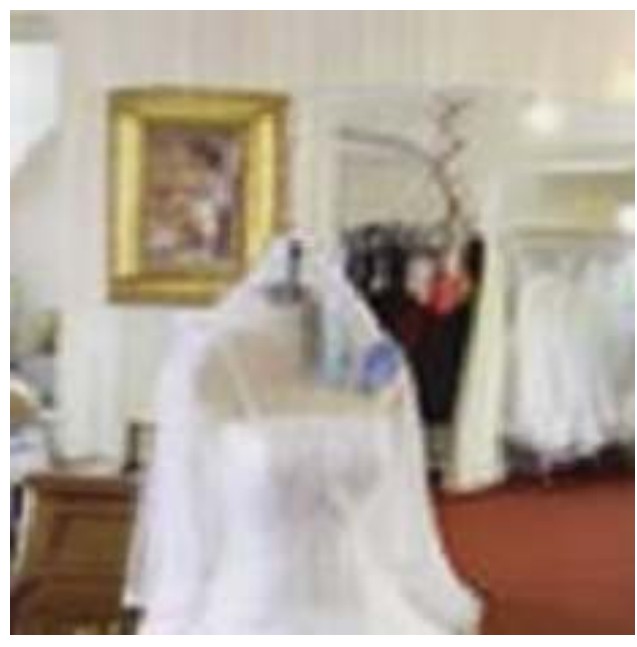

Before enhancement (stitched panorama)

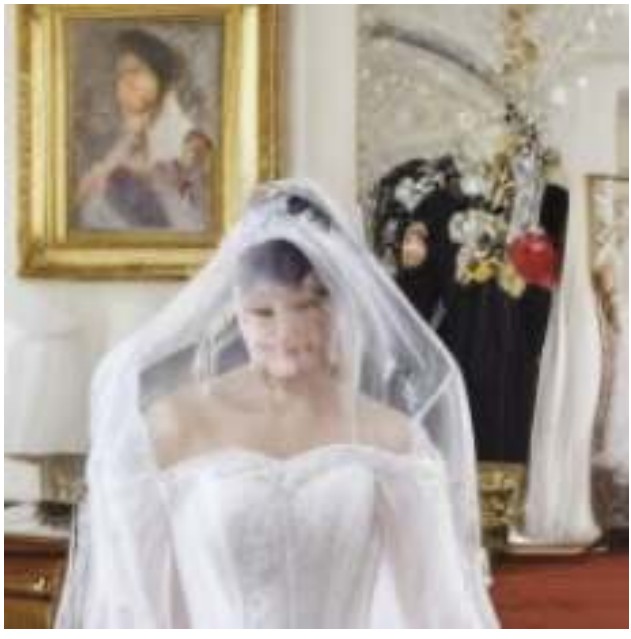

After enhancement module

Figure 46: Enhancement Failure Case 1 — The enhanced result hallucinates human-like structures absent in the stitched input.

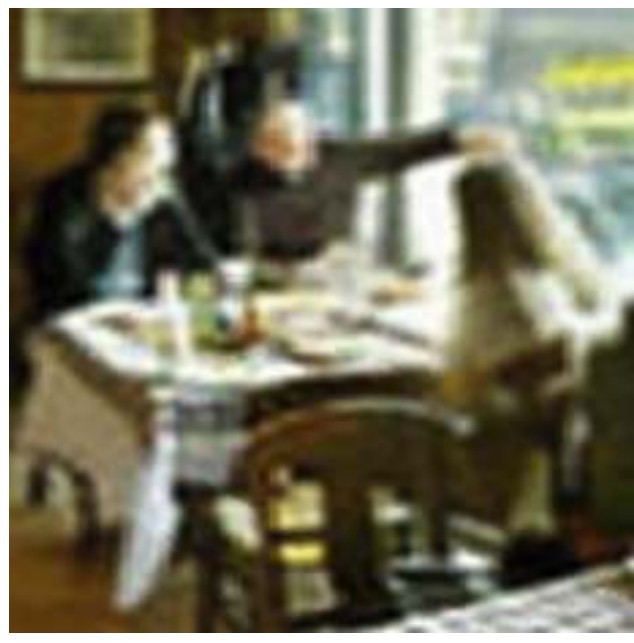

Before enhancement (stitched panorama)

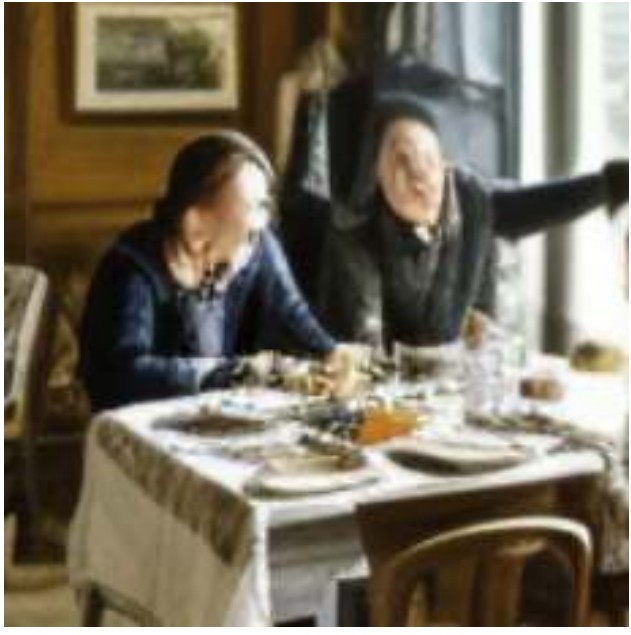

After enhancement module

Figure 47: Enhancement Failure Case 2 — Human-like textures may appear, but they are often not realistic or consistent.

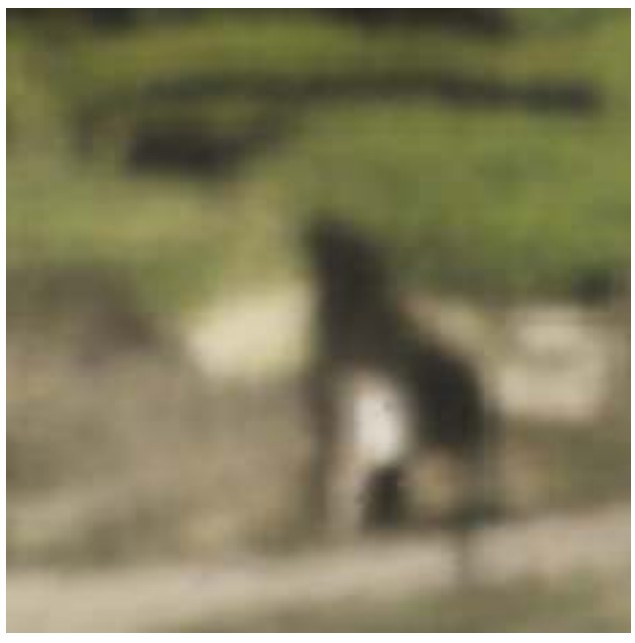

Before enhancement (stitched panorama)

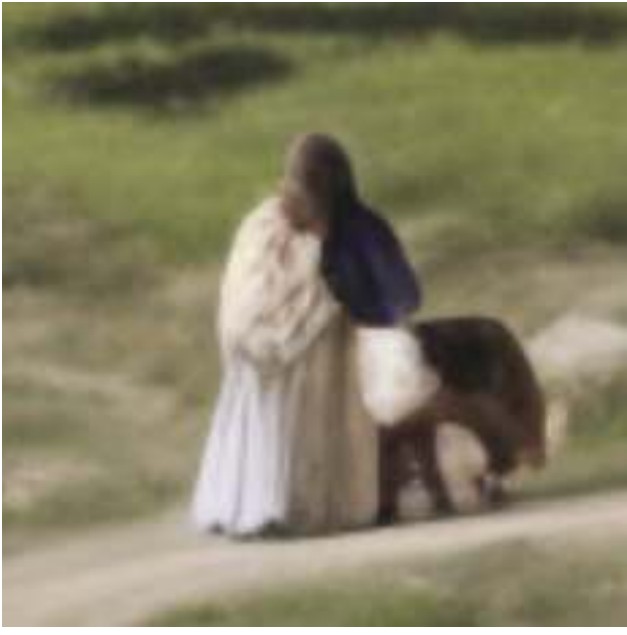

After enhancement module

Figure 48: Enhancement Failure Case 3 — The module may generate textures that do not resemble any identifiable object.

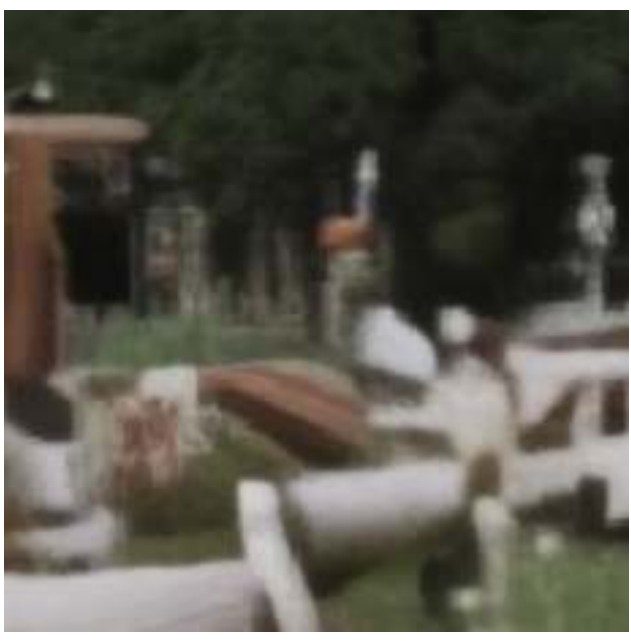

Before enhancement (stitched panorama)

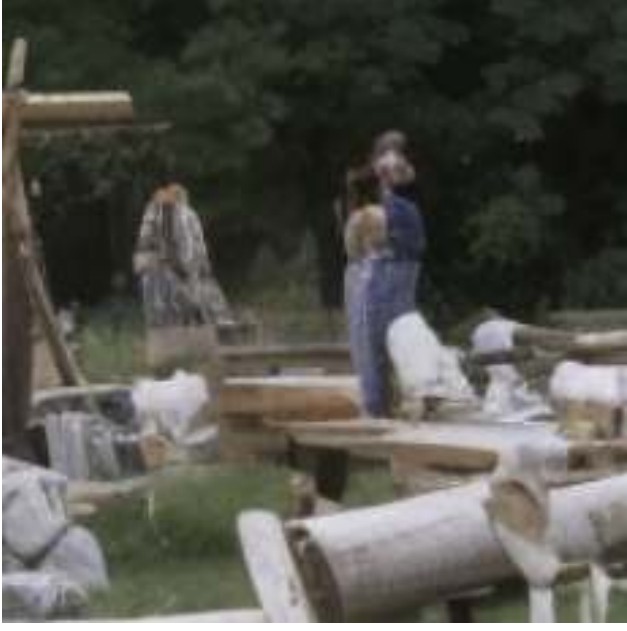

After enhancement module

Figure 49: Enhancement Failure Case 4 — Additional human-like shapes emerge that were not present before enhancement.

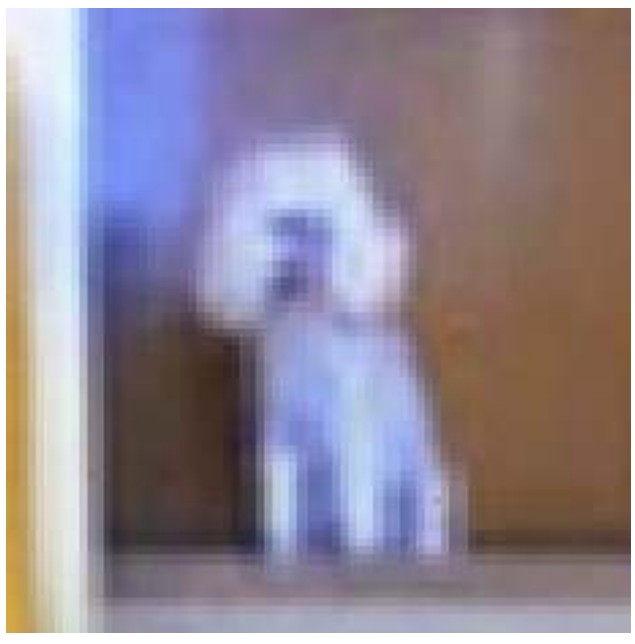

Before enhancement (stitched panorama)

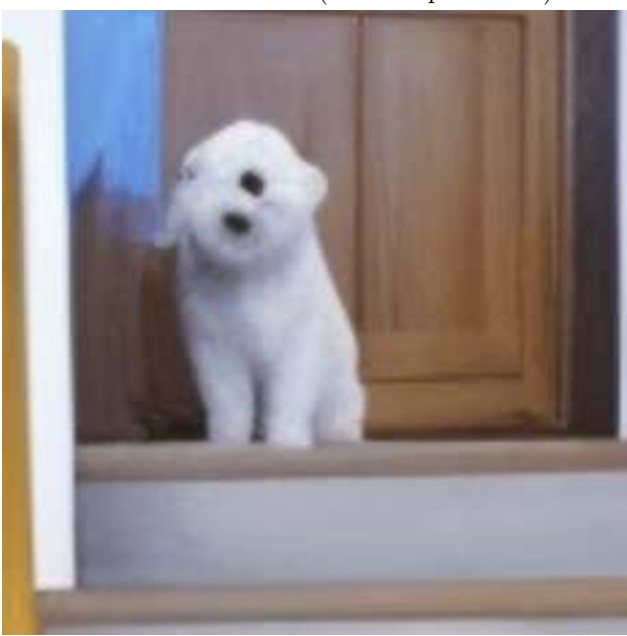

After enhancement module

Figure 50: Enhancement Failure Case 5 — Similar hallucinations may occur for animal-like patterns as well.