# OpenReview forum: "MultiViewPano: Training-Free 360° Panorama Generation via Multi-View Diffusion and Pose-Aware Stitching"
_ICLR.cc/2026/Conference — ICLR 2026 Conference Withdrawn Submission_

### Official Review · Reviewer_Zpeo · 2025-10-30

**Soundness:** 2
**Presentation:** 2
**Contribution:** 2
**Rating:** 4
**Confidence:** 3

**Summary:**

This paper introduces a novel, training-free method for generating 360° panoramas from pinhole camera images. The approach leverages the existing large-scale video model, SEVA, to synthesize a series of pinhole images along a defined camera trajectory. These images are then seamlessly combined into a final panorama using a custom-designed "Pose-based Stitcher" module.
A key advantage of this method is its flexibility, supporting both single- and multi-view inputs. The authors note that using multiple inputs can improve output quality. This framework overcomes critical limitations of prior work, such as the rigid requirement for 90° field-of-view inputs, the need for dataset-specific fine-tuning, and the inability to handle multiple input images. The method demonstrates strong, competitive results on the Laval Indoor and SUN360 datasets.

**Strengths:**

The paper proposes a training-free method that requires no additional training, offering strong generalization. Its performance on the Laval Indoor and SUN360 datasets even surpasses some methods that require dataset-specific fine-tuning.

The proposed method is highly flexible. Unlike prior work, it does not require input images to have a precise 90° field-of-view (FoV) or be captured from a fixed position. Instead, it supports any pinhole camera images as input, as long as their intrinsics and extrinsics (poses) are known.

The paper is well-structured, with clear logic that is easy to follow.

**Weaknesses:**

1. Lack of 360° Wraparound Consistency: The paper claims superior realism and semantic consistency. However, I observed a significant lack of 360° wraparound consistency at the horizontal edges (left and right) of the panoramas. When attempting to "wrap" the panorama by stitching the far-right edge to the far-left, the seams do not align seamlessly. This issue is evident in the first example of Figure 5, as well as in the second and fourth (top-to-bottom) examples in Figure 6. I question if this is a failure of the blending/feathering trick at the boundary, as this issue is notably absent in the CubeDiff results. This left-to-right consistency is crucial for panoramic images.


2. Critique of the 2D Stitching Method: I have reservations about the proposed stitching method. The approach, which relies on dynamic programming based on 2D pixel-wise L2 costs, seems to prioritize 2D pixel similarity. A more geometrically-sound approach should perhaps be considered. For instance, in overlapping regions, priority should be given to points closer to the camera to correctly handle 3D occlusion, rather than preserving content based solely on surrounding pixel consistency.

3. Confounding Variable (Clarity AI Upscaler): The use of the powerful "Clarity AI Upscaler" as a post-processing step introduces a confounding variable. It becomes difficult to determine whether the final seamless quality is achieved by the proposed dynamic programming stitcher or by the upscaler (which itself performs seamless tiling).

4. Unfair Comparison: Related to the point above, it seems unfair to compare their final, post-processed (upscaled) results against the non-upscaled outputs of the baseline methods.

5. Missing Discussion on Computational Cost: The framework relies on multiple large-scale models (SEVA, Clarity AI Upscaler). This likely results in significant computational overhead (e.g., resource consumption, inference latency) compared to other methods, yet this critical aspect is not discussed or benchmarked.

6. Insufficient Validation of Multi-View Claims: The claim to handle any number and pose of inputs is not sufficiently validated.

（1）The experiments are missing a quantitative ablation study on how the number of input images or the nature of their trajectories impacts the final output quality.

（2）Additionally, the methodological choice of using the camera pose centroid as the "scene center" for multi-view inputs seems arbitrary. This is not the "true" scene center, and it is questionable whether this is the most robust or geometrically reasonable choice.

7. Missing Related Work: The related work section is missing comparisons to other recent and highly relevant methods that also generate panoramas from pinhole inputs, such as PanoDecouple[1] (CVPR'25). While DA2: Depth Anything in Any Direction[2] might be too recent, a discussion and comparison against PanoDecouple seem warranted.

8. Missing LLM Usage Disclosure: A required section detailing the use of Large Language Models (LLMs) appears to be missing. I thoroughly reviewed the main paper and the supplementary materials, but could not find the LLM usage disclosure as mandated by ICLR.

**Questions:**

1. On Geometric Consistency and 3DGS Potential: The paper highlights improved consistency. How robust is this consistency in practice? Specifically, are the final stitched panoramas geometrically accurate enough to be used for downstream 3D scene reconstruction tasks, such as 3D Gaussian Splatting (3DGS)? If this method can serve as a reliable "pseudo-camera" rig for 3DGS, I would consider this a significant contribution and would be inclined to raise my score.

2. Request for Deeper Multi-View and Failure Analysis:

（1） The quantitative experiments are conducted entirely in the single-image-input regime. Could the authors provide more qualitative examples and discussion on both success and failure cases specifically for the multi-view input scenario?

（2） Could the authors also analyze how well the synthesized panorama preserves the content and fidelity of the original input pinhole images when multiple inputs are provided?

---

> ### Author Response · Authors · 2025-11-23
> **Addressing concerns about 360° wrap-around consistency, pose-aware stitching, upscaling, compuational cost analysis, scene-center choice and related works.**
>
> W1: Lack of 360° wrap-around consistency
> We acknowledge this limitation. The inconsistency at the left-right boundary is not caused by the stitcher, but by the ClarityAI upscaler, which currently does not incorporate equirectangular geometry. Please see supplementary (https://openreview.net/attachment?id=uYXHqNg87h&name=supplementary_material) for qualitative analysis of artefacts generated by ClarityAI upscaling.
>
> W2: Concerns regarding the 2D stitching method
> The stitcher is intentionally simple and built around pose-aware projection rather than 3D reasoning. However, we note:
> The stitcher is not the source of wrap-around issues (see W1).
> Existing classical image-based panorama pipelines also rely on 2D seam selection methods but they don’t rely on camera pose information and this yields lower quality stitched panoramas (Table 2).
> We don’t need 3D occlusion reasoning as SEVA’s synthetic views already ensure strong view overlap.
> We will clarify the design motivations and limitations in the revised draft.
>
> W3: Confounding effect of ClarityAI Upscaler
> We agree that the upscaler influences visual quality. The draft already contains quantitative evaluation before and after the upscaling step (Table 1). We would include more qualitative analysis of ClarityAI upscaling. Also, Diffusion360 [1] also uses similar approach (ESRGAN upscaling followed by ControlNet refinement which is very similar to ClarityAI).
>
> W4: Unfair comparison with non-upscaled baselines
> Our upscaling step specifically corrects artifacts that occur in training-free pose-based stitching. Other baselines do not suffer from these issues; therefore, upscaling is not required on their outputs. In the revised draft we will include discussion about this.
>
> W5: Missing computational cost analysis
> In the revised draft, we will add the breakdown of computational cost of different components of the framework.
> W6: Insufficient validation of multi-view claims
> The revised version will include Qualitative multi-view → panorama examples. However due to lack of benchmark, full quantitative evaluation is currently out of scope.
>
> Regarding the scene-center choice:
> For single-view, the scene center is simply the camera center.
> For multi-view, we used the centroid of camera poses as a heuristic.
> We acknowledge that this is not ideal and will discuss better initialization strategies as future work.
> Regarding ablation with the number of frames, SEVA always processes a fixed context of 21 images; using fewer inputs does not reduce SEVA inference time. So we fix the number of frames used to the SEVA context size. We will clarify this in the revision.
>
> W7: Missing related work
> We will add discussions of PanoDecouple (CVPR 2025) and DA2: Depth Anything in Any Direction. Direct quantitative comparison is not feasible, since PanoDecouple’s code is not released,
> They use different dataset splits and do not report CubeDiff comparisons.
> Our experimental setting follows CubeDiff and other baselines, enabling a consistent comparison.
>
> W8: Missing LLM usage disclosure
> We will add the required LLM usage disclosure section in the revised submission.
>
> 1. Feng, Mengyang, et al. "Diffusion360: Seamless 360 degree panoramic image generation based on diffusion models." arXiv preprint arXiv:2311.13141 (2023).
>
> Response to Questions
>
> Q1: Geometric consistency and 3DGS potential
> We will add:
> - Quantitative 3D consistency analysis using the MET3R metric [1].
> - Discussion of using generated panoramas as input for 3D Gaussian Splatting.
> - Several recent works [2-3] demonstrate that panorama → 3DGS is feasible, and our panoramas could serve as pseudo-captures in such pipelines.
>
> Q2: Deeper multi-view and failure analysis
> The revised version will include:
> - Qualitative examples of multi-view → panorama success and failure cases.
> - Fidelity analysis showing how much of the input content is preserved in the final panorama, measured via PSNR between input-image regions and the corresponding panorama regions.
>
> A full multi-view benchmark is planned as future work and beyond the scope of this revision.
>
> 1. Asim, Mohammad, et al. "Met3r: Measuring multi-view consistency in generated images." Proceedings of the Computer Vision and Pattern Recognition Conference. 2025.
> 2. Zhang, C., Zhang, W., Bao, Y., Chen, Y., Zhang, Y., Wang, J., & Wei, Y. (2025). PanSplat: 4K panorama synthesis with feed-forward Gaussian splatting. In Proceedings of the IEEE/CVF Conference on Computer Vision and Pattern Recognition (CVPR).
> 3. Zhang, C., Zhang, W., Bao, Y., Chen, Y., Zhang, Y., Wang, J., & Wei, Y. (2025). PanSplat: 4K panorama synthesis with feed-forward Gaussian splatting. In Proceedings of the IEEE/CVF Conference on Computer Vision and Pattern Recognition (CVPR).

---

### Official Review · Reviewer_RLVo · 2025-11-01

**Soundness:** 3
**Presentation:** 3
**Contribution:** 2
**Rating:** 4
**Confidence:** 4

**Summary:**

The paper introduces MultiViewPano, a training-free framework for generating 360° panoramas from one or more arbitrarily posed images. The method leverages a pre-trained multi-view diffusion model (SEVA) to synthesize novel views and employs a pose-aware stitching algorithm for seamless panorama assembly. The approach is evaluated on standard image-to-pano benchmarks and claims competitive results with state-of-the-art methods, while supporting flexible input configurations without retraining.

**Strengths:**

- The pipeline is well-thought out, modular, allowing for future improvements (e.g. compatibility with improved multi-view models)
- The training-free nature and support for arbitrary camera poses and fields of view are practical
- The pose-aware stitching algorithm is a useful addition
- Quantitative and qualitative results on standard benchmarks (Laval Indoor, SUN360) are competitive

**Weaknesses:**

- The method does not address coverage near polar regions, which is a common and prominent source of artifact in panorama generation. While the authors address this, it remains a weakness.
- The approach is heavily dependent on SEVA’s consistency; performance degrades with extreme viewpoint changes, and the method inherits SEVA’s weaknesses with reflective and transparent surfaces.
- Despite being designed for multi-view input, the evaluation is almost entirely in the single-image-to-panorama regime due to the lack of multi-view benchmarks. Even a small, manually curated dataset could greatly improve the work. Claimed advantages for multi-view scenarios are not rigorously validated. (Datasets like UDIS [1] could be adapted to this task, e.g. stitching using conventional algorithms to get a panorama, and then sampling various views)
- The need for post-processing (Clarity AI Upscaler) to address diffusion model artifacts suggests that the raw output quality is not always sufficient, and introduces a large computational overhead.

[1] Nie, Lang, et al. "Parallax-tolerant unsupervised deep image stitching." Proceedings of the IEEE/CVF international conference on computer vision. 2023.

**Questions:**

- How robust is the method to inaccurate or noisy camera pose estimates, which are common in real-world multi-camera setups?
- Can the approach be extended to support vertical camera trajectories, or is this fundamentally limited by the current model and pipeline?
- Are there plans to establish or contribute to a multi-view panorama benchmark to better validate the method’s intended use case?

---

> ### Author Response · Authors · 2025-11-23
> **Responding to Polar region generations, SEVA multiview consistency, and computational overhead for postprocessing**
>
> W1: Polar region coverage
> We acknowledge this limitation. In the revised version, we will include:
> A quantitative analysis of artifact frequency near the polar regions. Qualitative examples of polar regions are already included in the supplementary: https://openreview.net/attachment?id=uYXHqNg87h&name=supplementary_material
> A discussion of our experiments with vertical trajectories and why they fail under the current SEVA model.
> Potential solutions framed as future work.
>
> W2: Dependence on SEVA consistency
> The revised submission will add:
> Distance vs. consistency plots (PSNR, SSIM, LPIPS).
> A quantitative breakdown of the main failure modes.
> An analysis explaining when and how the method degrades due to SEVA inconsistencies.
>
> W3: Missing multi-view evaluation
> The revised version will include Qualitative analysis for multi-view → panorama generation.
> A full multi-view panorama benchmark is planned as future work and lies outside the scope of this revision.
> Regarding the UDIS dataset: we appreciate the suggestion. UDIS is designed for pairwise stitching, and adapting it to equirectangular panorama generation would require non-trivial additional processing. We will point to UDIS as a potential baseline for future work.
>
> W4: Post-processing overhead
> The revised version will include:
> Computational cost breakdown of different components of the pipeline.
> A discussion of the quality–cost trade-off inherent to the upscaling and blending steps.
>
> Response to Questions
> Q1: Robustness to noisy poses
> Single-image case: Not applicable, since poses are analytically generated.
> Multi-view case: Since we don’t have access to the ground truth poses, it is hard to evaluate the pose error. However, a benchmark for this could be proposed in the future work.
>
> Q2: Vertical trajectories
> We will provide the analysis of our vertical trajectory experiments and explain how SEVA’s horizontal sampling bias prevents such trajectories from producing reliable results. This is a fundamental limitation of current multi-view diffusion models. Please see supplementary for qualitative examples.
>
> Q3: Multi-view benchmark
> We agree that a multi-view → panorama benchmark would be valuable. We are exploring adapting datasets such as ScanNet++, and we include initial qualitative examples in the revised submission. A full benchmark is beyond the scope of this revision but planned as future work.

---

### Official Review · Reviewer_eab2 · 2025-11-02

**Soundness:** 3
**Presentation:** 2
**Contribution:** 2
**Rating:** 4
**Confidence:** 5

**Summary:**

This paper introduces MultiViewPano, a training-free method for generating panoramas from multiple input images with known camera poses and fields of view. The framework is built upon existing pretrained models, achieving reasonable results and demonstrating some design contributions. However, the overall novelty is limited, as most components are adaptations of prior work.

**Strengths:**

1.The method flexibly leverages existing pretrained models to achieve panoramic generation, and the overall architectural design contributes to the integration of multi-view information.
2.The proposed Pose-Aware Stitching algorithm is a notable addition, introducing a new approach to image alignment and blending.

**Weaknesses:**

1.Unfair comparison due to known camera poses:
oPrevious works such as PanoDiffusion and CubeDiff do not assume access to camera poses, whereas the proposed method explicitly uses them as priors (Line 196). This raises fairness concerns in comparisons.
oMoreover, if the input camera poses are unknown or inaccurate — which often happens in real-world scenarios — the proposed pipeline may not function properly. The authors should discuss the robustness of their method under pose estimation errors.
2.Strong dependence on SEVA outputs:
oThe method relies heavily on SEVA to produce the initial results. It is unclear how MultiViewPano performs when SEVA outputs poor-quality results. The authors are encouraged to provide visual examples or quantitative results in cases where SEVA fails, to demonstrate the robustness of their method.
3.Incomplete ablation study:
oThe ablation analysis is insufficient. The paper applies multiple quality enhancement steps, making it unclear which improvements stem from the proposed modules. The authors should visualize intermediate results at each stage to show the contribution of each component — e.g., whether seam removal is due to Pose-Aware Stitching or seamless tiling.
4.Potential distortions and artifacts introduced by enhancement steps:
oThe pipeline includes several enhancement stages, each of which may introduce geometric distortions or noise. The authors should explain how these issues are mitigated or controlled throughout the refinement process.
5.Limited performance improvement:
oIn Table 1, the proposed method does not outperform CubeDiff, particularly in indoor scenes. The authors should provide a detailed analysis of why the method fails to show consistent improvements across different settings.
6.Formatting issue:
oIn Table 1, the leading scores across metrics should be boldfaced for clarity and consistency.

**Questions:**

1.How fair is the comparison with prior works that do not assume access to camera poses? How does MultiViewPano perform when camera poses are inaccurate or unavailable?
2.How robust is the proposed method to poor-quality SEVA outputs? Can the authors provide visualizations of such cases?
3.Could the authors expand the ablation study and visualize intermediate outputs to clarify which modules contribute to which improvements?
4.How are geometric distortions and artifacts handled during repeated enhancement steps?
5.Why does the proposed method not outperform CubeDiff, especially for indoor scenes?
6.Could the authors ensure that all best-performing metrics in Table 1 are boldfaced for readability?

---

> ### Author Response · Authors · 2025-11-23
> **Clarifying the uses of Camera poses for stitching, and panorama's dependent on SEVA output. Also addressing strengths/weaknesses of upscaling.**
>
> W1: Unfair comparison: use of known camera poses
> We clarify the role of poses in our system:
> - Single-image setting: No input poses are available. We analytically generate a camera trajectory. These poses are outputs of our trajectory design, not inputs.
> - Multi-view setting: When poses are not provided, we follow the standard SFM pipeline and estimate them using DUSt3R.
> Our pose-aware stitcher then uses these generated or estimated poses. This comparison remains fair because:
> - Our stitcher outperforms traditional non–pose-aware stitchers (Table 2).
> - CubeDiff adopts a similar strategy, cubemap projection implicitly assumes the poses in specific spatial configuration and uses them to construct the panorama.
> In the multiview-to-panorama setting, we use standard SfM pose estimation tools such as COLMAP and DUSt3R.
>
> W2: Dependence on SEVA outputs
> We include qualitative failure cases in the supplementary materials. In the revised version, we also provide qualitative examples of Multiview -> Panorama generations in the supplementary (https://openreview.net/attachment?id=uYXHqNg87h&name=supplementary_material)
> W3: Incomplete ablation study
>
> The revised version will add:
> Step-by-step visualizations: raw SEVA outputs → stitched panorama → upscaled result (Also present in the supplementary).
> An analysis of seam artifacts before and after the upscaling stage.
> W4: Distortions introduced by enhancement steps
> The step-by-step visualizations in the revised version (see W3) will address this concern directly. The upscaling module uses ControlNet for structural preservation and multi-band blending to maintain geometric consistency.
> W5: Limited performance improvement over CubeDiff
> Our primary distinction is training-free operation, whereas CubeDiff requires fine-tuning on each benchmark dataset. This represents a different trade-off:
> Flexibility and applicability versus dataset-specific optimization.
> We outperform CubeDiff on SUN360, demonstrating stronger generalization.
> The performance gap on indoor scenes is small (FID 12.82 vs. 11.7).
> Training-free methods enable use cases where panoramic datasets are unavailable.
> W6: Formatting
> This will be fixed in the revision.
>
> Q1–Q6:
> All reviewer questions correspond directly to the weaknesses above; please see our detailed responses in W1–W5.

---

### Official Review · Reviewer_S4vh · 2025-11-02

**Soundness:** 3
**Presentation:** 3
**Contribution:** 2
**Rating:** 2
**Confidence:** 3

**Summary:**

MultiViewPano presents a training-free framework for generating full 360° panoramas from one or more arbitrarily posed images with varying field-of-view (FoV). Unlike prior arts that require fixed 90° FoV, single-center capture, or task-specific fine-tuning, the method leverages a pre-trained multi-view diffusion model (SEVA) to synthesize overlapping novel views along a designed camera trajectory, and then fuses them via a pose-aware stitching algorithm that directly exploits known camera geometry instead of fragile feature matching.

Key Contributions

1） Training-free pipeline: Supports any number of images, any pose, any FoV without retraining or task-specific optimization.

2） Pose-aware stitching: A geometric fusion module that projects views to a common spherical canvas.

**Strengths:**

Overall, the key strength of this work lies in its practical applicability:

1）Training-free and practical framework.
The paper presents a panorama generation pipeline that requires no fine-tuning or re-training, greatly improving versatility and deployment efficiency—especially in real-world scenarios where large-scale panoramic datasets are unavailable.

2） Multi-view input support broadens applicability.
Unlike most prior methods that rely on a single image or fixed-view inputs, MultiViewPano accepts any number of images with arbitrary poses, making it far more adaptable to practical capture conditions.

**Weaknesses:**

My main concern regarding this work is its limited novelty.

1） Essentially, it combines the SEVA model with traditional stitching methods, offering marginal technical contribution.

2） The robustness of the approach remains unclear when SEVA fails to produce consistent novel views, especially in challenging scenarios with large viewpoint changes or repetitive textures.

3） The experimental comparison is insufficient. Notably, recent methods such as Diffusion360, PanFusion and HunyuanWorld are not included in the evaluation, which undermines the comprehensiveness of the benchmark.

**Questions:**

1.3D consistency failures in SEVA

In my understanding, SEVA does not always guarantee 3D-consistent multi-view outputs.
Under such conditions, would a reasonable panorama still be produced?

2.Incomplete baseline coverage

The experimental comparison omits several recent strong baselines.

---

> ### Author Response · Authors · 2025-11-23
> **Novelty lies in evaluating training free approach informed by video priors from SEVA for Panorama Generation**
>
> W1: Limited novelty: combining SEVA with traditional stitching:
>
> We acknowledge that our pipeline is built on SEVA, but we emphasise that the paper introduces capabilities that are not achievable with prior work. Specifically:
> - First training-free panorama generation method: All existing baselines require fine-tuning on limited panoramic datasets, whereas our system operates entirely without retraining.
> - First method supporting arbitrary poses and FoV: CubeDiff requires a fixed 90° FoV, and PanoDiffusion assumes a single capture centre. Our method supports unrestricted camera poses and FoVs.
>
> Although we use SEVA for novel-view synthesis, our trajectory design, pose-aware stitching module, and overall system integration collectively enable these new capabilities. These components are necessary to achieve training-free panoramic generation and do not exist in prior work.
>
> W2: Robustness when SEVA fails
> In the revised version, we will add:
> - A quantitative evaluation of SEVA’s multi-view consistency using metrics such as MET3R [1].
> - An analysis of how panorama quality varies with SEVA’s consistency scores.
> - Failure-cases examples in the supplementary material.
>
> W3: Incomplete baseline coverage:
>
> We follow CubeDiff’s (ICLR 2025) evaluation protocol exactly in selecting comparable baselines. Regarding the listed methods:
> Diffusion360: The paper provides no quantitative evaluation. An independent survey [2] indicate performance comparable to PanoDiffusion, which our method already surpasses. Regarding other approaches mentioned:
> - PanFusion: This is a text-to-panorama system requiring text prompts, and therefore solves a different problem from our image-to-panorama setting.
> - HunyuanWorld: Trained primarily on synthetic Unreal Engine data; its domain differs significantly from real-world panoramas used in our benchmark.
> Thus, these methods are not directly comparable to our task setting.
> 1. Asim, Mohammad, et al. "Met3r: Measuring multi-view consistency in generated images." Proceedings of the Computer Vision and Pattern Recognition Conference. 2025.
> 2. Wang, Hai, et al. "A survey on text-driven 360-degree panorama generation." arXiv preprint arXiv:2502.14799 (2025).
>
> Response to Questions
>
> Q1: 3D consistency failures in SEVA
> We will include the following analyses in the revised draft:
> A quantitative analysis between SEVA’s consistency (e.g., M3TER scores) and the quality of the final panorama.
> Distance-vs-PSNR (Distance from captured image(s) to generated images) plots that illustrate performance degradation when SEVA’s consistency drops.
>
> Q2: Incomplete baseline coverage
> Please refer to our response to W3 above.
>
> Please see the supplementary for detailed qualitative analysis: https://openreview.net/attachment?id=uYXHqNg87h&name=supplementary_material

---

### Note · Authors · 2025-12-03

I have read and agree with the venue's withdrawal policy on behalf of myself and my co-authors.